# Understanding Edge of Stability in Rank-1 Linear Models for Binary Classification

## Abstract

Recent research in deep learning optimization reveals that many neural network architectures trained using gradient descent with practical step sizes, $\eta$, exhibit an interesting phenomenon where the top eigenvalue of the Hessian of the loss function, $\lambda_1^H$ increases to and oscillates about the stability threshold, $\frac{2}{\eta}$. The two parts of the trajectory are referred to as progressive sharpening and edge of stability. The oscillation in $\lambda_1^H$ is accompanied by a non-monotonically decreasing training loss. In this work, we study the Edge of Stability phenomenon in a two-layer rank-1 linear model for the binary classification task with linearly separable data to minimize logistic loss. By capturing the core training dynamics of our model as a low-dimensional system, we rigorously prove that Edge of Stability behavior is not possible in the simplest one datapoint setting. We also empirically show that, with two datapoints, it is possible for Edge of Stability to occur and point out the source of the oscillation in $\lambda_1^H$ and non-monotonic training loss. We also give new approximations to $\lambda_1^H$ for such models. Lastly, we consider an asymptotic setting, in the limit as the margin converges to $0$, and provide empirical results that suggest the loss and sharpness trajectories may exhibit stable, perpetual oscillation.

## 1 Introduction

To optimize modern deep neural networks, algorithms such as stochastic gradient descent (SGD) and adaptive optimizers, such as Adam (Kingma & Ba, 2017), have become the go-to choice. However, the foundation for both algorithms, gradient descent, is not fully understood. In scenarios with a small learning rate $\eta$, gradient descent is well-understood via the descent lemma (Nesterov et al.):

**Lemma 1.1.** *For some function $f(\theta)$, if $\lambda_{max}\left(\nabla^2 f(\theta)\right) \leq \beta$ and $\theta_{t+1} = \theta_t - \eta \nabla f(\theta_t)$, then*

$$f\left(\theta_{t+1}\right) \leq f(\theta_t) - \eta\left(1 - \frac{\eta}{2}\beta\right)||\nabla f(\theta_t)||^2$$

The descent lemma suggests that one should choose learning rate $\eta$ to be close to $1/\beta$, where $\beta$ is an upperbound on the "sharpness" (largest eigenvalue of the Loss Hessian). However, recently efforts to understand neural network training with gradient descent with practical learning rates have revealed some interesting behavior. In particular, work by Cohen et al. (2021) reveals that many standard neural network architectures exhibit trajectories that can be broken down into two phases (see Figure 1(b)). Phase 1, denoted as Progressive Sharpening, is the regime where the sharpness increases until it reaches past the stability threshold $\frac{2}{\eta}$. Then, it enters phase 2, denoted as Edge of Stability (EOS), where the sharpness hovers around the stability threshold $\frac{2}{\eta}$ and the loss function in question decreases non-monotonically. This common training behavior has attracted much attention from the research community and has inspired theoretical research aimed at studying simple, low dimensional models with similar behavior as well as more general characterizations for this sort of training dynamics. However, since much of the analysis on these models is done using regression-type losses, our understanding of Edge of Stability in classification losses, such as logistic loss, is still limited.

### 1.1 Our Contributions

In this work, we build upon the contributions of Wu et al. (2023) and Kalra et al. (2025) by studying Edge of Stability behavior in a two-layer rank-1 linear neural network trained to minimize logistic

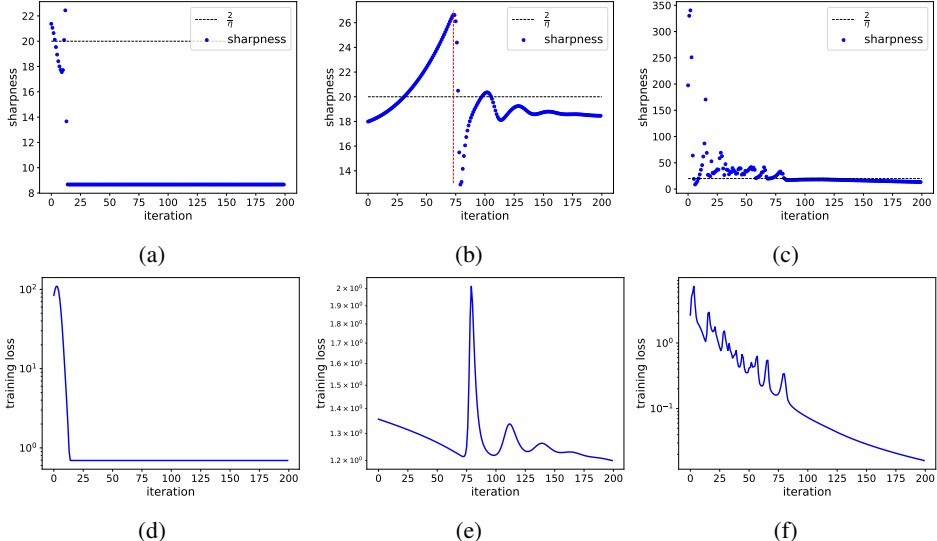

Figure 1: Sharpness and loss trajectories for 3 different models. Top row represents sharpness; bottom row represents training loss. The 3 models from left to right are our model with 1 datapoint (a)(d), 2 datapoints (b)(e) and a 4-layer feedforward ReLU network trained on a 1000 datapoint, linearly separable subset of the MNIST dataset (c)(f).

loss with gradient descent under some initialization assumptions. By distilling model dynamics to a low-dimensional system, we partition the system's phase space and prove that, under the simplest setting of one training datapoint, our model does not exhibit Edge of Stability.

However, by adding another datapoint to our training dataset, we empirically show that our model can exhibit Edge of Stability behavior. We provide an example of the loss and sharpness dynamics of our model in both dataset settings compared to more realistic neural network in Figure 1.

In our analysis of the two-datapoint setting, we determine the source of the Edge of Stability behavior and explain why it does not exist in the single datapoint setting. Furthermore, we investigate an asymptotic case in the two datapoint setting, where we consider the limit as the margin for the dataset approaches $0$. Based on simulations of our low-dimensional system, we believe it is possible for the system to exhibit perpetual stable oscillation in both the sharpness and loss.

We further provide a detailed comparison with the two-layer linear model studied in the regression setting (Kalra et al., 2025). Our comparison reveals that the condition under which Edge of Stability occurs in our model is more subtle than in the regression setting, since the parameter iterates of our model move between two nullclines of our low-dimensional system whereas the regression setting iterates move between an unstable axis. While the analysis in the regression setting (Kalra et al., 2025) relies on the trace of the Loss Hessian as a surrogate for the sharpness, we provide a closed-form formula of the sharpness in our single training datapoint setting and a low-error approximation of the sharpness in our two training datapoint setting.

## 2 RELATED WORK

Oscillations in overall convergent loss during training have been observed in a number of prior works [(Lewkowycz et al., 2020), Jastrzebski et al. (2020), (Wu et al., 2018), (Xing et al., 2018), (Arora et al., 2018), (Li et al., 2022)]. Cohen et al. (2021) formalized this observation in their empirical study of many neural network architectures and training objectives, denoting it as Edge of Stability.

We first note a number of works that analyze Edge of Stability for general functions. Ma et al. (2022) use subquadratic approximations of loss functions to explain the mechanism behind Edge of Stability training dynamics. Arora et al. (2022) prove Edge of Stability behavior for normalized gradient descent on a smooth loss function with time-dependent learning rate and gradient descent

on the square root of a smooth loss function with constant learning rate. Damian et al. (2023) identify the third-order term in the Taylor expansion of a loss function as a source of implicit bias for gradient descent that prevents divergence in the Edge of Stability regime. We also note research by Rosca et al. (2023) and Cohen et al. (2025) that develops continuous time tools to study the oscillatory dynamics in sharpness and loss observed with gradient descent training in the Edge of Stability regime. We note that Cohen et al. (2025) also show how their approach can be extended to adaptive optimizers, such as RMSProp.

Next, we discuss works that study Edge of Stability for particular models. We first note some works [(Wang et al., 2022), (Song & Yun, 2023), (Kalra et al., 2025)] that focus on studying two-layer linear networks in the regression setting. Wang et al. (2022) focus on studying sharpness dynamics by assuming direct correlation of the sharpness with the norm of the output layer. Song & Yun (2023) prove that the gradient descent trajectories observed to minimize a convex, 1-lipschitz, even loss align with the bifurcation diagram of a one-dimensional iterated map. Kalra et al. (2025) conduct fixed point analysis on the two-layer linear neural network and show that it can exhibit Edge of Stability behavior and period-doubling route to chaos. Besides two-layer linear networks, we also note that Zhu et al. (2023b) theoretically study the training dynamics of a 4-layer linear scalar network under a coupled initialization and prove the Edge of Stability behavior in a large local region.

For logistic loss, Wu et al. (2023) develop a new technique to study the implicit bias of logistic regression in the Edge of Stability regime and prove that it is guaranteed to converge, unlike exponential loss. Liu et al. (2023) prove the existence of the "catapult phase" in two-layer linear models on non-separable data with logistic loss. In comparison, we note that our work focuses on the separable data setting, points out the source of the Edge of Stability behavior, and studies an asymptotic case.

We defer further discussion of prior works to Appendix B.

## 3 PRELIMINARIES AND NOTATION

In this section, we introduce our two-layer linear model, some definitions, and the training dataset settings that we study.

### 3.1 TWO-LAYER RANK-1 LINEAR MODEL

A standard two-layer linear network would map $\mathbf{x}$ to $q^T M \mathbf{x}$. However, in such cases the only relevant component in $M$ would be a rank-1 component that is aligned with $q$. Motivated by the results in Ji & Telgarsky (2019), where they prove the asymptotic convergence of the weight matrices in a deep linear network (trained to minimize logistic loss) to rank-1 matrices with specific alignment properties, we break $M$ into a rank 1 matrix $uv^T$ to have a clearer understanding of its alignment behavior.

The model we study, $f : \mathbb{R}^d \to \mathbb{R}$, is parameterized by $u \in \mathbb{R}^h$, $q \in \mathbb{R}^h$, and $v \in \mathbb{R}^d$, where $f_{q,u,v}(\mathbf{x}) = q^T u v^T \mathbf{x}$. For clarity, we denote $d$ as the dimension of our input $\mathbf{x}$, where $\mathbf{x} \in \mathbb{R}^d$ is the feature vector and $h$ as the width of our model's hidden linear layer, $uv^T$. For a binary label $y \in \{\pm 1\}$, we consider the standard logistic loss $L : \mathbb{R}^{2h+d} \to \mathbb{R}$ on our training dataset $(\mathbf{x}_i, y_i)_{i=1}^N$ using gradient descent with constant learning rate $\eta$. We use a standard way to express the logistic loss for a datapoint $(\mathbf{x}_i, y_i)$ as $L = \log(1 + \exp(-f(y_i\mathbf{x}_i)))$.

At time-step $t$, the loss for our model is as follows:

$$L(q_t, u_t, v_t) = \sum_{i=1}^N \log\left(1 + \exp(-q_t^T u_t v_t^T y_i \mathbf{x}_i)\right)$$

The iterated map for our parameters $u$, $q$, and $v$ are also shown below.

$$q_{t+1} = q_t + \eta\, u_t \sum_{i=1}^N \left(1 + \exp(q_t^T u_t v_t^T y_i x_i)\right)^{-1} v_t^T y_i \mathbf{x}_i,$$

$$u_{t+1} = u_t + \eta \, q_t \sum_{i=1}^{N} \left(1 + \exp(q_t^T u_t v_t^T y_i x_i)\right)^{-1} v_t^T y_i \mathbf{x}_i,$$

$$v_{t+1} = v_t + \eta \, q_t^T u_t \sum_{i=1}^{N} \left(1 + \exp(q_t^T u_t v_t^T y_i x_i)\right)^{-1} y_i \mathbf{x}_i$$

To further simplify the model, we assume that model parameters $q$ and $u$ start from the same initialization, similar to the setup of Zhu et al. (2023b). With this initialization assumption, we make the following claim.

**Claim 3.1.** *If $q$ and $u$ start from the same initialization (i.e., $u_0 = q_0$), then, for any time $t > 0$, $u_t = q_t$.*

We defer the proof of the claim to Appendix C.

## 3.2 DEFINITIONS

Now we give the formal definition for sharpness:

**Definition 1** (Sharpness). Consider $\theta = \begin{bmatrix} q^T & u^T & v^T \end{bmatrix}^T$. We define sharpness, $\lambda_1^H$, as the largest algebraic eigenvalue of the Hessian of loss function $L$, $\lambda_{max}(\nabla_\theta^2 L)$.

In order to define Edge of Stability, we rely on the property that the loss value oscillates in the Edge of Stability regime. We start by defining the local maximum of the loss.

**Definition 2** (Local Maximum). For a discrete-time one-dimensional dynamical system $n$, we say the $n_t$ ($n$ a time $t \in \mathbb{N}, t > 0$) is a local maximum if $n_{t-1} < n_t$ and $n_{t+1} < n_t$.

With the local maximum, we argue a trajectory is in Edge of Stability regime if the loss has multiple local maxima.

**Definition 3** (Edge of Stability). Consider training a neural network using gradient descent with step size $\eta$ to minimize loss $L$. If there exists some local region during training (for discrete time-steps $k, l \in \mathbb{N}, l \geq k + 4$) where the loss $L$ trajectory has $n \geq 2$ local maxima at times $t_i$ for $i \in \{t_1, ..., t_n\}$ such that $k < t_1 < ... < t_n < l$, then we say the model exhibits Edge of Stability behavior.

We also note that the non-monotonic behavior of the loss function we describe in our Edge of Stability definition is often accompanied by the sharpness oscillating or hovering about the stability threshold $\frac{2}{\eta}$.

In our analysis, the main tool is to reduce the model to a lower dimensional equivalent system, and consider *nullclines* of system's components – boundaries where a component stays the same after a single iteration.

**Definition 4** (Nullcline). Consider an $r$-dimensional discrete-time dynamical system $n$, such that such that the iterated map for it is $n_{t+1} = f(n_t)$, where $f(n_t) = (f_1(n_t), ..., f_r(n_t))$ and $n_t = (n_{1,t}, ..., n_{r,t})$. For component $i \in \{1, .., r\}$, we define the nullcline of $n_{i,t}$ as the set of points $\{n_t\}$ such that $f_i(n_t) - n_{i,t} = 0$.

## 3.3 DATA SETUP

**One Training Datapoint** ($N = 1$)  In this setting, our training dataset $(\mathbf{x}_i, y_i)_{i=1}^N$ is simply $(\mathbf{x}, y)$, where label $y \in \{\pm 1\}$.

**Two Training Datapoints** ($N = 2$)  In this setting, our training dataset $(\mathbf{x}_i, y_i)_{i=1}^N$ is $(\mathbf{x}_1, 1), (\mathbf{x}_2, -1)$. We focus on the following range of parameters for this setting which often leads to Edge of Stability behavior in experiments.

1. $||\mathbf{x}_1||_2 = ||\mathbf{x}_2||_2 = 1$
2. $0.99 \leq \mathbf{x}_1^T \mathbf{x}_2 < 1$

3. there exists $w_* \in \mathbb{R}^d$ s.t. $w_*^T y_1 \mathbf{x}_1 > 0$ and $w_*^T y_2 \mathbf{x}_2 > 0$

4. $0.10 \le \eta \le 0.20$

## 3.4 FIXED POINT ANALYSIS

After constructing the low-dimensional systems for each dataset setting, we find and determine the stability of the fixed points. To find the fixed points, we first find the nullclines with respect to each component in the system. After finding the nullclines for each component, we find the points where the nulllclines for all the components of the system intersect. Those are our fixed points. We then determine the stability properties of the fixed points by forming the Jacobian of the system $J$ and analyzing its eigenvalues $\lambda_i^J$. If all the eigenvalues at a particular fixed point satisfy $|\lambda_i^J| < 1$, then we say that fixed point is stable. If all the eigenvalues at a particular fixed point satisfy $|\lambda_i^J| > 1$, then we say that fixed point is unstable. Lastly, if all the eigenvalues at a particular fixed point satisfy $|\lambda_i^J| = 1$, then we say that fixed point is marginally stable.

# 4 ANALYSIS OF GRADIENT DESCENT DYNAMICS

In this section, we analyze the dynamics of our model in the one and two datapoint settings based on the low dimensional systems we derive. Using the fixed point stability analysis, we present our main theoretical result that, in the single datapoint setting, our model does not exhibit Edge of Stability and describe a sufficient trajectory that produces Edge of Stability behavior in the two datapoint case. We also present our findings in an asymptotic case that we study in the two datapoint setting.

## 4.1 SINGLE TRAINING DATAPOINT ($N = 1$)

To understand the dynamics of the single training datapoint setting, we first observe that the dynamics can be completely captured by two quantities, $\|q_t\|_2^2$ and $v_t^T y\mathbf{x}$. This is because, under our initialization assumption, $u_t = q_t$, which we know from Claim 3.1. So, we get that $q_t^T u_t = \|q_t\|_2^2$. Since we are also dealing with only one datapoint, the component of $v_t$ that is orthogonal to $\mathbf{x}$ does not affect the updates of the parameters in the model. This allows us to study the training dynamics of the model as a 2-dimensional system with variables $\|q_t\|_2^2$ and $v_t^T y\mathbf{x}$.

In our analysis, we focus on the part of the phase space where $\|q_t\|_2^2 > 0$ since $\|q_t\|_2^2 = 0$ is a fixed point. We first show that $v_t^T y\mathbf{x}$ is strictly increasing in Lemma 4.1.

**Lemma 4.1.** *For any $k, l \in \mathbb{N}$ such that $l > k$ and $\|q_k\|_2^2 > 0$, $v_l^T y\mathbf{x} > v_k^T y\mathbf{x}$.*

For $\|q_t\|_2^2$, our analysis shows two important nullclines, which are $v_t^T y\mathbf{x} = 0$ and $\left(1 + \exp\left(\|q_t\|_2^2 v_t^T y\mathbf{x}\right)\right)^{-1} v_t^T y\mathbf{x} = -\frac{2}{\eta}$. The derivation of the nullclines and fixed points can be found in Appendix C.1.1. We also provide a visual representation of these nullclines in Figure 2. Using these nullclines, we partition the phase space into 3 disjoint regions, which we denote as regions 1, 2, and 3. Since $\|q_t\|_2^2 v_t^T y\mathbf{x}$ is the primary component in the loss for this setting, and we know that $\|q_t\|_2^2 v_t^T y\mathbf{x}$ grows (i.e., the loss shrinks) in the next time step if the system is in regions 1 and 2 according to corollaries 4.1.1 and 4.1.2, we further divide region 3 into regions 3.1 and 3.2. We define region 3.1 and region 3.2 such that $\|q_t\|_2^2 v_t^T y\mathbf{x}$ grows and shrinks in the next time step, respectively.

**Corollary 4.1.1.** *For any $k, l \in \mathbb{N}$ such that $l > k$, if $\|q_k\|_2^2 > 0, v_k^T y\mathbf{x} \ge 0$, then $\|q_l\|_2^2 v_l^T y\mathbf{x} > \|q_k\|_2^2 v_k^T y\mathbf{x}$.*

**Corollary 4.1.2.** *For any $k \in \mathbb{N}$, if $\|q_k\|_2^2 > 0, v_k^T y\mathbf{x}$ satisfies*

$$-\frac{2}{\eta} < \left(1 + \exp\left(\|q_k\|_2^2 v_k^T y\mathbf{x}\right)\right)^{-1} v_k^T y\mathbf{x} < 0, \text{ then } \|q_{k+1}\|_2^2 v_{k+1}^T y\mathbf{x} > \|q_k\|_2^2 v_k^T y\mathbf{x}.$$

By analyzing the dynamics of our system in each region, we prove the following theorem.

**Theorem 4.2** (No Edge of Stability in the One Datapoint Setting)**.** *Consider our two-layer rank-1 linear model trained to minimize logistic loss on a single datapoint $(\mathbf{x}, y)$, where $y = \pm 1$, using gradient descent with learning rate $\eta$. Under any initialization such that $\|q_0\|_2^2 > 0$, our loss trajectory will contain at most one local maximum.*

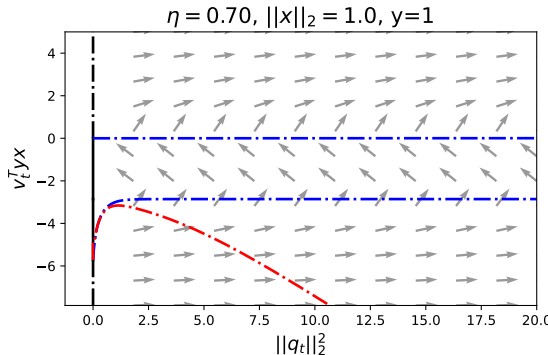

Figure 2: This is the phase space diagram for our one datapoint system for the case where $\eta = 0.70$, $\|\mathbf{x}\|_2^2 = 1$, and $y = 1$. The blue dashed curves represent the nullclines for $\|q_t\|_2^2$, where region 1 is above the top blue curve, region 2 is between both blue curves, and regions 3.1 and 3.2 are below the bottom blue curve. The red dashed curve represents the boundary where $\|q_t\|_2^2 v_t^T y\mathbf{x}$ shrinks inside the curve (region 3.2) and grows outside the curve (region 3.1). Since $\|q_t\|_2^2 = 0$ is also a fixed point of the system, we denote the solid black line portion as $\|q_t\|_2^2 = 0$ being stable, and the dashed black line portion as $\|q_t\|_2^2 = 0$ being unstable. We also include normalized vectors of the gradient field, shown as grey arrows, to provide some intuition for possible trajectories.

We defer the proof of the lemma, corollaries, and theorem in this section to Appendix C.1.3.

### 4.2 TWO TRAINING DATAPOINTS ($N = 2$)

In the two datapoint setting, we note that the dynamics can be captured by three quantities, $\|q_t\|_2^2$, $v_t^T(\mathbf{x}_1 - \mathbf{x}_2)$, and $v_t^T(\mathbf{x}_1 + \mathbf{x}_2)$. As in the single datapoint case, with our initialization, we know that $u_t = q_t$ from Claim 3.1. So, $q_t^T u_t = \|q_t\|_2^2$. However, since we have two datapoints, we observe the updates to the parameters of our model are affected by $v_t^T \mathbf{x}_1$ and $v_t^T \mathbf{x}_2$. However, we find $v_t^T(\mathbf{x}_1 - \mathbf{x}_2)$ and $v_t^T(\mathbf{x}_1 + \mathbf{x}_2)$ are more meaningful than $v_t^T \mathbf{x}_1$ and $v_t^T \mathbf{x}_2$, since $v_t^T(\mathbf{x}_1 - \mathbf{x}_2)$ and $v_t^T(\mathbf{x}_1 + \mathbf{x}_2)$ represent the alignment of parameter $v_t$ in the max-margin $(\mathbf{x}_1 - \mathbf{x}_2)$ and max-margin complement $(\mathbf{x}_1 + \mathbf{x}_2)$ directions, respectively. Furthermore, we can reconstruct $v_t^T \mathbf{x}_1$ and $v_t^T \mathbf{x}_2$ from $v_t^T(\mathbf{x}_1 - \mathbf{x}_2)$ and $v_t^T(\mathbf{x}_1 + \mathbf{x}_2)$. For conciseness, we define $m_t = v_t^T(\mathbf{x}_1 - \mathbf{x}_2)$ and $c_t = v_t^T(\mathbf{x}_1 + \mathbf{x}_2)$. So, we can study our model's training dynamics as the 3-dimensional system $\left(\|q_t\|_2^2, c_t, m_t\right)$. Similar to the single datapoint case, we find that $\|q_t\|_2^2 = 0$ is a fixed point of our system, so we focus on the system's behavior for $\|q_t\|_2^2 > 0$. And, similar to $v_t^T y\mathbf{x}$ in the single datapoint setting, we prove that $m_t$ is a strictly increasing component in Claim 4.3.

**Claim 4.3.** *For any $k, l \in \mathbb{N}$ such that $l > k$ and $\|q_k\|_2^2 > 0$, $m_l > m_k$.*

We defer the proof of this claim to Appendix C.2.

From our analysis of $\|q\|_2^2$, we define two of its nullclines below,

$$\left(1 + \exp\left(\frac{1}{2}\|q_t\|_2^2 (m_t + c_t)\right)\right)^{-1}(m_t + c_t) + \left(1 + \exp\left(\frac{1}{2}\|q_t\|_2^2 (m_t - c_t)\right)\right)^{-1}(m_t - c_t) = b,$$

for $b \in \{0, -\frac{4}{\eta}\}$. For convenience, we denote $\|q_t\|_2^2$ nullcline 1 for $b = 0$ and $\|q_t\|_2^2$ nullcline 2 for $b = -\frac{4}{\eta}$. However, unlike the single datapoint setting, we have the additional component, $c_t$. In our analysis, we find that the system can move away from the $c_t = 0$ plane for $\|q_t\|_2^2 > \sqrt{\frac{8}{\eta(1 + \mathbf{x}_1^T \mathbf{x}_2)}}$ and move towards otherwise. To understand how "far" this repelling behavior reaches, we derive the following closed-form boundary, where, if the system is in the interior of the boundary, $|c_t|$ grows.

$$\left(1 + \exp\left(\frac{1}{2}\|q_t\|_2^2 (m_t - |c_t|)\right)\right)^{-1} - \left(1 + \exp\left(\frac{1}{2}\|q_t\|_2^2 (m_t + |c_t|)\right)\right)^{-1} = \frac{2|c_t|}{\eta\|q_t\|_2^2 (1 + \mathbf{x}_1^T \mathbf{x}_2)}$$

We provide visuals for the $\|q_t\|_2^2$ nullclines and $|c_t|$ growth boundary in Figure 3. The derivation for nullclines, $|c_t|$ growth boundary, and fixed points can be found in Appendix C.2.1.

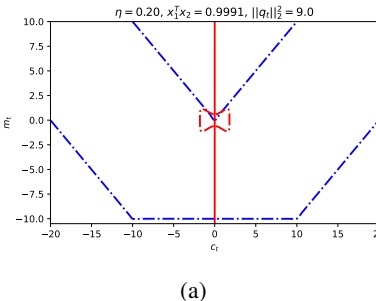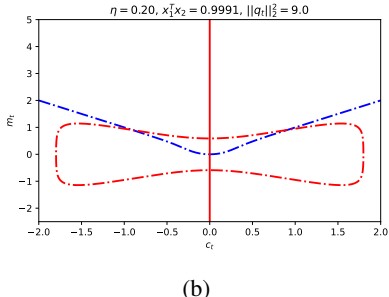

(a)                                    (b)

Figure 3: For a slice of of the 3-dimensional phase at $\|q_t\|_2^2 = 9.0$ for the case where $\eta = 0.20$ and $\mathbf{x}_1^T\mathbf{x}_2 = 0.9991$, we show the nullclines of $\|q_t\|$ and the $|c_t|$ growth boundary. In (a), the blue dashed lines represent the nullclines of $\|q_t\|_2^2$, where the bottom blue curve is nullcline 2 and the blue curve above it is nullcline 1. In (b), we provide zoomed-in view for the red dashed curve shown in a) that represents the $|c_t|$ growth boundary.

Compared to the single datapoint case, where we proved Edge of Stability was not possible, we empirically observe Edge of Stability behavior in the two datapoint setting, as shown in the Figure 4.

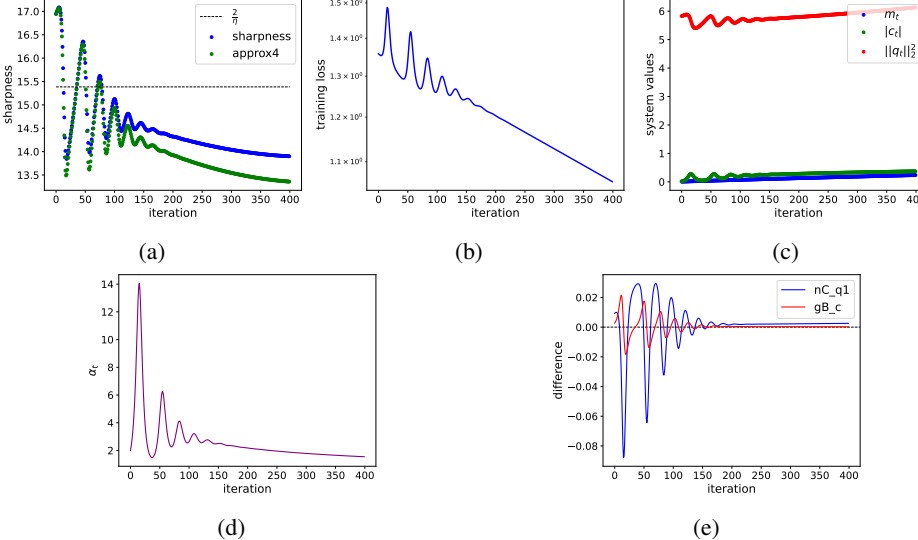

Figure 4: We show the sharpness (a), training loss (b), system values (c), $\alpha$ (i.e., $\frac{|c_t|}{m_t}$) (d), and "location" of the system relative to $\|q_t\|_2^2$ to nullcline 1 ($nC\_q1 = 0$) and $|c_t|$ the growth boundary ($gB\_c = 0$) (e) for $\eta = 0.13$, $\mathbf{x}_1^T\mathbf{x}_2 = 0.9991$, $\|q_0\|_2^2 = 1.05\sqrt{\frac{8}{0.13\cdot1.9991}}$, $c_0 = 0.02$, $m_0 = 0.01$. We also show the similarity between our sharpness approximation and the true sharpness.

Now, we explain how the trajectory in Figure 4 produces Edge of Stability behavior. We do so by explaining the first oscillation in loss. We defer our explanation of the sharpness dynamics in Figure 4 to Appendix C.2.3 using the sharpness approximation we derived in Appendix C.2.2. To explain the first oscillation in loss, we first split the initial trajectory into four phases based on the sign of $nC\_q1$ and $gB\_c$ in Figure 4(e).

In the initial part of the trajectory, we see that the system is inside $\|q_t\|_2^2$ nullcline 1 and the $|c_t|$ growth boundary ($nC\_q1 > 0, gB\_c > 0$), which we denote as phase 1. Then, the system is outside $\|q_t\|_2^2$ nullcline 1, but still inside the $|c_t|$ growth boundary ($nC\_q1 < 0, gB\_c > 0$), which we denote as phase 2. In phase 3, we see that the system is outside both $\|q_t\|_2^2$ nullcline 1 and the $|c_t|$ growth

boundary ($nC\_q1 < 0, gB\_c < 0$). Lastly, in phase 4, the system is inside $\|q_t\|_2^2$ nullcline 1, but still outside the $|c_t|$ growth boundary ($nC\_q1 > 0, gB\_c < 0$).

To explain the loss trajectory, we first rewrite our loss using $\alpha_t = \frac{|c_t|}{m_t}$ (see Figure 4(d)) as shown below.

$$L = \log\left(1 + \exp\left(-\frac{1}{2}\left(1 - \alpha_t\right)\|q_t\|_2^2 m_t\right)\right) + \log\left(1 + \exp\left(-\frac{1}{2}\|q_t\|_2^2 m_t\left(1 + \alpha_t\right)\right)\right)$$

In the first phase, we observe $\|q_t\|_2^2$ and $\alpha_t$ grow. During the first few iterations of this phase, we observe a decrease in loss. From our rewritten loss above, we see that for small $\alpha_t > 1$, $\|q_t\|_2^2 m_t$ needs to be larger in order for the loss to increase. We observe that $\|q_t\|_2^2 m_t$ is not large enough for the first few steps even as $\alpha_t$ increases. Once $\alpha_t$ and $\|q_t\|_2^2 m_t$ grow large enough, then the loss increases for the remainder of phase 1. In the second phase of the trajectory, we observe $\alpha_t$ increase and $\|q_t\|_2^2$ shrink. The loss begins to decrease, since $\|q_t\|_2^2 m_t$ is shrinking due to $\|q_t\|_2^2$. After this, in the third phase of the system's trajectory, we observe $\|q_t\|_2^2$ and $\alpha_t$ shrink, and this causes the loss to continue to decrease. Once the system enters the fourth phase, $\|q_t\|_2^2$ begins to grow and $\alpha_t$ continues to shrink. We continue to see decreases in loss due to the shrinkage of $\alpha_t$.

Following this last phase, we observe that $\|q_t\|_2^2$ continues to grow and $\alpha_t$ continues to shrink until the system is inside the $|c_t|$ growth boundary ($gB\_c > 0$), where we enter phase 1 again. Once this happens, the cycle repeats. We note that the dampening in the oscillations for both loss and sharpness is likely the result of the growth of $m_t$, which constrains the magnitude of $\alpha_t$ (see Figure 4(d)).

We attribute the Edge of Stability behavior observed above to $c_t$. Compared to our single data-point setting, the $c_t$ component in the two-datapoint setting provides a means for the system to oscillate between $\|q_t\|_2^2$ nullcline 1 (i.e., $nC\_q1$ changes signs), causing oscillations in $\|q_t\|_2^2$, which causes oscillations in the loss. We provide additional experiments for the two datapoint setting in Appendix C.2.4.

### 4.2.1 ASYMPTOTIC CASE ($\mathbf{x}_1^T \mathbf{x}_2 \rightarrow 1$)

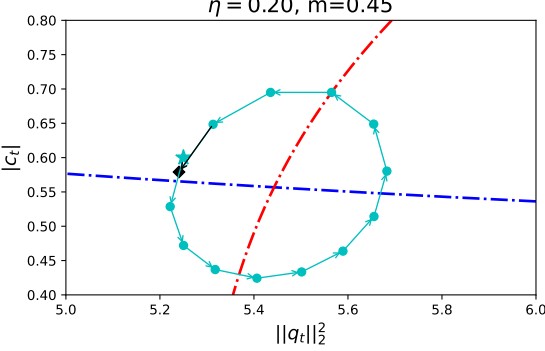

Figure 5: For $m = 0.45$ and $\eta = 0.20$, we show the first 14 steps of a trajectory starting close the intersection point between $\|q_t\|_2^2$ nullcline 1 and the $|c_t|$ growth boundary. We denote the starting point as the cyan star the 14th step as the black diamond, with cyan arrows pointing to subsequent iterates and a black arrow from iterate 13 to 14.

We further study an asymptotic case, where we consider the limit as $\mathbf{x}_1^T \mathbf{x}_2 \rightarrow 1$. This is meant to consider the behavior of our model as the margin for our dataset approaches 0. In this limit, our system reduces to 2 dimensions in terms of $\|q_t\|_2^2, c_t$, where $m_t$ becomes a constant based on initialization. Results from the simulations conducted for this case can be found in Appendix C.2.5. From the simulations, we find cases where our asymptotic system remains localized, jumping between a specific set of values producing a trajectory "band". From these cases, we focus on a particular set of instances with $m = 0.45$ and $\eta = 0.2$, where, in the trajectory tail, the system is localized appears around the point where the $\|q_t\|_2^2$ nullcline 1 intersects with the $|c_t|$ growth boundary. Upon further investigation, we find that, under a similar hyperparameter setting, initializing close to the

intersection point seems to generate a stable elliptical orbit, as shown in Figure 5. Based on the figure, we formulate a conjecture below.

**Conjecture 4.3.1.** *There exists some $\eta, m > 0$ where, if the system $(\|q_t\|_2^2, |c_t|)$ is initialized at some $\epsilon > 0$ distance from the intersection point of $\|q_t\|_2^2$ nullcline 1 and the $|c_t|$ growth boundary, then it will enter and remain in a stable orbit around the intersection point.*

## 5 DIFFERENCE BETWEEN SQUARED LOSS AND LOGISTIC LOSS

Based on our analysis, we highlight some of the differences between our two layer rank-1 linear model to minimize logistic loss and the two-layer linear model studied by Kalra et al. (2025) to minimize mean squared loss.

**Training Dataset Size** Kalra et al. (2025) highlight that Edge of Stability can occur in a two-layer linear model even with a single training datapoint, as long as the corresponding label $y \neq 0$. However, in our work, we find that Edge of Stability does not occur with only a single training datapoint. The simplest setting we found for Edge of Stability in our model was with two training datapoints.

**Existence of critical learning rate** $\eta_c$ Furthermore, Kalra et al. (2025) state that Edge of Stability occurs in their setting once the learning rate $\eta$ crosses some critical learning rate $\eta_c$. In our work, we claim that the occurrence of the Edge of Stability relies on more than just crossing some $\eta_c$. We find that it is necessary for $\mathbf{x}_t^T \mathbf{x}_2$ to be above some critical threshold, $(\mathbf{x}_t^T \mathbf{x}_2)_c$. While we know that for any choice of $\eta$ and $\mathbf{x}_t^T \mathbf{x}_2$, there exists a region of the phase space where the system moves away from the $c_t = 0$ plane, our model is not guaranteed to exhibit Edge of Stability behavior for $\mathbf{x}_t^T \mathbf{x}_2$ that is not large enough. Intuitively, we know that a large $\mathbf{x}_t^T \mathbf{x}_2$ corresponds to a smaller optimal margin of separation, since the datapoints are "harder" to distinguish. The small margin would then constrain the growth of $m_t$, requiring more iterations to escape the region where the $c_t = 0$ is unstable (assuming initialization in this region). This allows the system to movement across the $\|q_t\|_2^2$ nullclines for sufficient iterations until $m_t$ becomes large enough. So, the faster growth rate of $m_t$ with smaller $\mathbf{x}_1^T \mathbf{x}_2$ may cause the system to "escape" the $c_t$ growth region quite early during training, resulting in no sufficient oscillations in $|c_t|$ or $\|q_t\|_2^2$ to produce to oscillations in the loss or sharpness.

**Edge of Stability Source** In the regression setting, Kalra et al. (2025) claim that Edge of Stability occurs when a fixed point, defined as the line where the residual of their model is 0 for appropriate values of the Loss Hessian trace (zero-loss line), becomes unstable. The instability of the zero-loss line causes oscillations in the residual that cause the parameters to move towards the Edge of Stability manifold. In our classification setting, we find an analogous condition, where Edge of Stability can occur when our 3-dimensional system enters the region where the $c_t = 0$ plane is unstable and remains there for a sufficient number of training steps. However, we find that oscillations in $|c_t|$ that produce movement of the system in and out of $\|q_t\|_2^2$ nullcline 1 causes Edge of Stability behavior. This is because oscillations of the system about $\|q_t\|_2^2$ nullcline 1 causes $\|q_t\|_2^2$ to oscillate, and we find that $\|q_t\|_2^2$ is a significant component in the sharpness based on our interpretable approximation in Appendix C.2.2.

## 6 CONCLUSION AND FUTURE DIRECTIONS

In this work, we study the behavior of a two-layer rank-1 linear neural network to minimize logistic loss on linearly separable data. In this setting, we prove that with one datapoint the model does not display Edge of Stability behavior. We also show that with two datapoints, our model exhibits Edge of Stability. Furthermore, we extend our two datapoint setting to an asymptotic case where the margin approaches 0 and provide evidence that supports the possibility for perpetual stable oscillation in the loss and sharpness.

The main open problem is to prove the Edge of Stability behavior for the two datapoint setting rigorously and generalize it to more general settings, such as multiple linearly separable datapoints. We hope our analysis, together with the approximation of sharpness in Appendix C.2.2, provides a starting point for understanding Edge of Stability behavior in logistic losses for more complicated models.

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

## A    USE OF LARGE LANGUAGE MODELS

In this work, we used a Large Language Model to assist our sharpness derivations. Specifically, we used the Large Language Model to realize how similarity transforms could be applied to the Loss Hessian and the impact of the rank-2 perturbation in the Loss Hessian for the two datapoint setting. We also used the Large Language Model to look for resources related to eigenvalue approximation under perturbation, where the model recommended a book (Stewart & Sun, 1990) from which we learned about Cauchy Interlacing Theorem.

## B    ADDITIONAL RELATED WORKS

We continue our discussion of related works in this section. First, we discuss some additional works on general functions. Kong & Tao (2020) show that, under large learning rates and certain loss functions, gradient descent exhibits chaotic behavior leading it to converge to a distribution rather than a local minimizer. Lyu et al. (2022) show that a model with some form of normalization (batch normalization, layer normalization, etc.) and weight decay can enter the Edge of Stability regime, where the implicit bias induced drives the model to reduce sharpness. Ahn et al. (2022) discuss the causes and main features of unstable convergence (Edge of Stability) based on the evolution of gradient descent iterates, loss, and sharpness. Kreisler et al. (2023) identify the gradient-flow sharpness (GFS) as a monotonically decreasing quantity in gradient descent training and study how, in scalar neural networks, the decrease of the GFS below the stability threshold causes the loss to decrease at an exponential rate. Wang et al. (2023) attribute the occurrence of non-trivial training dynamics, such as Edge of Stability, to loss functions that satisfy certain regularity conditions along with gradient descent and sufficiently large learning rates. Bartlett et al. (2023) study sharpness-aware minimization (SAM) for convex quadratic and non-quadratic objectives. In the latter case, they show that the oscillations observed during training are essentially gradient descent steps on the spectral norm of the Hessian, where SAM represents the derivative of the sharpness. Cohen et al. show that the Edge of Stability phenomenon can carry over to adaptive optimizers, such as Adam, by considering the maximum eigenvalue of a preconditioned Loss Hessian. Long & Bartlett (2024) derive the Edge of Stability threshold for sharpness-aware minimization (SAM) and experimentally observe SAM operating in the Edge of Stability regime. Chen et al. (2024a) prove that, for twice-differentiable loss functions, gradient descent is forward-invariant. The authors also show that for initialization outside the forward-invariant set will oscillate for several iterations until entering the forward invariant set containing a local minimum.

Next, we continue the discussion of works focused on specific models. Pedregosa et al. (2022) show that quadratic regression models trained to minimize a quartic loss function exhibit both Progressive Sharpening and Edge of Stability. Chen & Bruna (2023) use two-step gradient updates to analyze matrix factorization and single-neuron ReLU networks for a range of learning rates where gradient descent dynamics hover around minimizers. Ahn et al. (2023) show that a 2-layer ReLU network can exhibit Edge of Stability behavior and prove that, with learning rates not sufficiently large enough, the network can fail to learn a nonzero first-layer bias, which is essential for useful implicit biases that can lead to better generalization. Even et al. (2023) study the implicit bias of 2-layer diagonal linear neural networks trained with gradient descent (GD) and stochastic gradient descent (SGD) using large step sizes. They show that the implicit bias discrepancy between SGD and GD is amplified with learning rates that induce Edge of Stability behavior. Noci et al. (2024) study sharpness dynamics in the context of learning rate transfer for neural networks in the rich regime and in the Neural Tangent Kernel to indicate feature learning. Zhu et al. (2023a) show that quadratic models used the approximate shallow feedforward ReLU networks exhibits a "catapult phase" when trained with sufficiently large learning rates and demonstrate improved generalization following the "catapult phase". Chen et al. (2024b) show that phase retrieval and a cubic iterated map representing a two-layer neural network with quadratic activations, fixed output layer, and orthogonal data exhibit five phases of training, including Edge of Stability.

## C    MODEL ANALYSIS

In this section, we provide additional information for our study of the single and two datapoint settings including sharpness derivations, fixed point analysis, and additional experiment results.

First, we prove Claim 3.1, which is essential for our analysis in both dataset settings.

*Proof of Claim 3.1.* Assume $u_0 = q_0$. We prove the claim by induction. First, we prove the base case for $u_1$ and $q_1$.

$$u_1 - q_1 = u_0 + \eta \, q_0 \sum_{i=1}^{N} \left(1 + \exp(q_0^T u_0 v_0^T y_i \mathbf{x}_i)\right)^{-1} v_0^T y_i \mathbf{x}_i$$

$$- \left(q_0 + \eta \, u_0 \sum_{i=1}^{N} \left(1 + \exp(q_0^T u_0 v_0^T y_i \mathbf{x}_i)\right)^{-1} v_0^T y_i \mathbf{x}_i\right)$$

$$= u_0 - q_0 - \eta \, (u_0 - q_0) \sum_{i=1}^{N} \left(1 + \exp(q_0^T u_0 v_0^T y_i \mathbf{x}_i)\right)^{-1} v_0^T y_i \mathbf{x}_i$$

$$= 0 \qquad \qquad \text{since } u_0 = q_0$$

Next, we prove the inductive step. Assume $u_k = q_k$. Then, for $u_{k+1}$ and $q_{k+1}$, we see the following.

$$u_{k+1} - q_{k+1} = u_k + \eta \, q_k \sum_{i=1}^{N} \left(1 + \exp(q_k^T u_k v_k^T y_i \mathbf{x}_i)\right)^{-1} v_k^T y_i \mathbf{x}_i$$

$$- \left(q_k + \eta \, u_k \sum_{i=1}^{N} \left(1 + \exp(q_k^T u_k v_k^T y_i \mathbf{x}_i)\right)^{-1} v_k^T y_i \mathbf{x}_i\right)$$

$$= u_k - q_k - \eta \, (u_k - q_k) \sum_{i=1}^{N} \left(1 + \exp(q_k^T u_k v_k^T y_i \mathbf{x}_i)\right)^{-1} v_k^T y_i \mathbf{x}_i$$

$$= 0 \qquad \qquad \text{since } u_k = q_k$$

$\square$

## C.1 SINGLE DATAPOINT SETTING ($N = 1$)

### C.1.1 FIXED POINT ANALYSIS

Recall from Section 4 that can study the dynamics of our model in the single datapoint setting as the system, $(\|q_t\|_2^2, v_t^T y\mathbf{x})$. In this section, we derive the fixed points for our system, $(\|q_t\|_2^2, v_t^T y\mathbf{x})$. We make the following claims.

**Claim C.1** (Nullclines of $\|q_t\|_2^2$). *The nullclines of $\|q_t\|_2^2$ are as follows:* $\|q_t\|_2^2 = 0$, $v_t^T y\mathbf{x} = 0$, $\left(1 + \exp\left(\|q_t\|_2^2 v_t^T y\mathbf{x}\right)\right)^{-1} v_t^T y\mathbf{x} = -\frac{2}{\eta}$, *and* $\|q_t\|_2^2 v_t^T y\mathbf{x} \to \infty$.

**Claim C.2** (Fixed points for $(\|q_t\|_2^2, v_t^T y\mathbf{x})$ system). *The fixed points for the system of two variables,* $\|q_t\|_2^2$ *and* $v_t^T y\mathbf{x}$ *are* $\|q_t\|_2^2 v_t^T y\mathbf{x} \to \infty$ *and* $\|q_t\|_2^2 = 0$, *where* $\|q_t\|_2^2 v_t^T y\mathbf{x} \to \infty$ *is marginally stable and* $\|q_t\|_2^2 = 0$ *is stable only if* $\frac{-4}{\eta} < v_t^T y\mathbf{x} < 0$.

First, we prove Claim C.1.

*Proof of Claim C.1.* First, we define the iterated map of $\|q_t\|_2^2$.

$$\|q_{t+1}\|_2^2 = \|q_t\|_2^2 + 2\eta \left(1 + \exp\left(\|q_t\|_2^2 v_t^T y\mathbf{x}\right)\right)^{-1} \|q_t\|_2^2 v_t^T y\mathbf{x} + \eta^2 \left(1 + \exp\left(\|q_t\|_2^2 v_t^T y\mathbf{x}\right)\right)^{-2} \|q_t\|_2^2 \left(v_t^T y\mathbf{x}\right)^2$$

$$= \|q_t\|_2^2 \left(1 + \eta \left(1 + \exp\left(\|q_t\|_2^2 v_t^T y\mathbf{x}\right)\right)^{-1} v_t^T y\mathbf{x}\right)^2$$

To solve for the nullclines, we solve for $\|q_{t+1}\|_2^2 - \|q_t\|_2^2 = 0$, which is equivalent to solving $\left(1 + \exp\left(\|q_t\|_2^2 v_t^T y\mathbf{x}\right)\right)^{-1} \left(2 + \eta \left(1 + \exp\left(\|q_t\|_2^2 v_t^T y\mathbf{x}\right)\right)^{-1} v_t^T y\mathbf{x}\right) \|q_t\|_2^2 v_t^T y\mathbf{x} = 0$. From the

factors, we see that the nullclines are $\|q_t\|_2^2 = 0$, $v_t^T y\mathbf{x} = 0$, $\left(1 + \exp\left(\|q_t\|_2^2 v_t^T y\mathbf{x}\right)\right)^{-1} v_t^T y\mathbf{x} = -\frac{2}{\eta}$, and $\|q_t\|_2^2 v_t^T y\mathbf{x} \to \infty$ to get $\left(1 + \exp\left(\|q_t\|_2^2 v_t^T y\mathbf{x}\right)\right)^{-1} = 0$. $\qquad\square$

Next, we proceed with the fixed point stability analysis.

*Proof of Claim C.2.* From Claim C.1, we already have the nullclines for $\|q_t\|_2^2$. So, we find the nullclines of $v_t^T y\mathbf{x}$. First, we define the iterated map for $v_t^T y\mathbf{x}$.

$$v_{t+1}^T y\mathbf{x} = v_t^T y\mathbf{x} + \eta \left(1 + \exp\left(\|q_t\|_2^2 v_t^T y\mathbf{x}\right)\right)^{-1} \|q_t\|_2^2 \|y\mathbf{x}\|_2^2$$

We solve for the nullclines as follows.

$$0 = v_{t+1}^T y\mathbf{x} - v_t^T y\mathbf{x} = \eta \left(1 + \exp\left(\|q_t\|_2^2 v_t^T y\mathbf{x}\right)\right)^{-1} \|q_t\|_2^2 \|y\mathbf{x}\|_2^2$$

$$0 = \left(1 + \exp\left(\|q_t\|_2^2 v_t^T y\mathbf{x}\right)\right)^{-1} \|q_t\|_2^2 \qquad\qquad \text{since } \|y\mathbf{x}\|_2^2 \text{ is a constant}$$

So, we get that the nullclines are $\|q_t\|_2^2 = 0$ and $\|q_t\|_2^2 v_t^T y\mathbf{x} \to \infty$. Then, we see that the intersection of the nullclines for $\|q_t\|_2^2$ and $v_t^T y\mathbf{x}$ shows that the fixed points are $\|q_t\|_2^2 = 0$ and $\|q_t\|_2^2 v_t^T y\mathbf{x} \to \infty$.

To analyze the stability properties of the fixed points, we find the eigenvalues for the Jacobian of the system $(\|q_t\|_2^2, v_t^T y\mathbf{x})$. First, we define the Jacobian as follows.

$$J(\|q_{t+1}\|_2^2, v_{t+1}^T y\mathbf{x}) = \begin{bmatrix} \frac{\partial \|q_{t+1}\|_2^2}{\partial \|q_t\|_2^2} & \frac{\partial \|q_{t+1}\|_2^2}{\partial v_t^T y\mathbf{x}} \\ \frac{\partial v_{t+1}^T y\mathbf{x}}{\partial \|q_t\|_2^2} & \frac{\partial v_{t+1}^T y\mathbf{x}}{\partial v_t^T y\mathbf{x}} \end{bmatrix}$$

Then, we solve for the partial derivatives.

$$\frac{\partial v_{t+1}^T y\mathbf{x}}{\partial v_t^T y\mathbf{x}} = 1 - \eta \left(1 + \exp\left(\|q_t\|_2^2 v_t^T y\mathbf{x}\right)\right)^{-2} \exp\left(\|q_t\|_2^2 v_t^T y\mathbf{x}\right) \|q_t\|_2^4 \|y\mathbf{x}\|_2^2$$

$$\frac{\partial v_{t+1}^T y\mathbf{x}}{\partial \|q_t\|_2^2} = \eta \left(1 + \exp\left(\|q_t\|_2^2 v_t^T y\mathbf{x}\right)\right)^{-1} \|y\mathbf{x}\|_2^2 - \eta \left(1 + \exp\left(\|q_t\|_2^2 v_t^T y\mathbf{x}\right)\right)^{-2} \exp\left(\|q_t\|_2^2 v_t^T y\mathbf{x}\right) \|q_t\|_2^2 v_t^T y\mathbf{x} \|y\mathbf{x}\|_2^2$$

$$\frac{\partial \|q_{t+1}\|_2^2}{\partial v_t^T y\mathbf{x}} = 2\eta \left(1 + \exp\left(\|q_t\|_2^2 v_t^T y\mathbf{x}\right)\right)^{-1} \|q_t\|_2^2 + 2\eta^2 \left(1 + \exp\left(\|q_t\|_2^2 v_t^T y\mathbf{x}\right)\right)^{-2} \|q_t\|_2^2 v_t^T y\mathbf{x}$$

$$- 2\eta \left(1 + \exp\left(\|q_t\|_2^2 v_t^T y\mathbf{x}\right)\right)^{-2} \exp\left(\|q_t\|_2^2 v_t^T y\mathbf{x}\right) \|q_t\|_2^4 v_t^T y\mathbf{x}$$

$$- 2\eta^2 \left(1 + \exp\left(\|q_t\|_2^2 v_t^T y\mathbf{x}\right)\right)^{-3} \exp(\|q_t\|_2^2 v_t^T y\mathbf{x}) \|q_t\|_2^4 \left(v_t^T y\mathbf{x}\right)^2$$

$$\frac{\partial \|q_{t+1}\|_2^2}{\partial \|q_t\|_2^2} = 1 + 2\eta \left(1 + \exp\left(\|q_t\|_2^2 v_t^T y\mathbf{x}\right)\right)^{-1} v_t^T y\mathbf{x} + \eta^2 \left(1 + \exp\left(\|q_t\|_2^2 v_t^T y\mathbf{x}\right)\right)^{-2} \left(v_t^T y\mathbf{x}\right)^2$$

$$- 2\eta \left(1 + \exp\left(\|q_t\|_2^2 v_t^T y\mathbf{x}\right)\right)^{-2} \exp\left(\|q_t\|_2^2 v_t^T y\mathbf{x}\right) \|q_t\|_2^2 \left(v_t^T y\mathbf{x}\right)^2$$

$$- 2\eta^2 \left(1 + \exp\left(\|q_t\|_2^2 v_t^T y\mathbf{x}\right)\right)^{-3} \exp\left(\|q_t\|_2^2 v_t^T y\mathbf{x}\right) \|q_t\|_2^2 \left(v_t^T y\mathbf{x}\right)^3$$

We first consider the fixed point $\|q_t\|_2^2 v_t^T y\mathbf{x} \to \infty$. We see that the Jacobian reduces to the following.

$$J(\|q_{t+1}\|_2^2, v_{t+1}^T y\mathbf{x}) = \begin{bmatrix} 1 & 0 \\ 0 & 1 \end{bmatrix}$$

Since the both eigenvalues for the Jacobian are 1, $\|q_t\|_2^2 v_t^T y\mathbf{x} \to \infty$ is marginally stable.

Next, we consider the fixed point $\|q_t\|_2^2 = 0$. The Jacobian reduces to the following.

$$J(\|q_{t+1}\|_2^2, v_{t+1}^T y\mathbf{x}) = \begin{bmatrix} 1 + \eta v_t^T y\mathbf{x} + \frac{\eta^2 \left(v_t^T y\mathbf{x}\right)^2}{4} & 0 \\ \frac{\eta}{2}\|y\mathbf{x}\|_2^2 & 1 \end{bmatrix}$$

The eigenvalues for the Jacobian are 1 and $1 + \eta v_t^T y\mathbf{x} + \frac{\eta^2 (v_t^T y\mathbf{x})^2}{4}$. This means that a segment of the $v_t^T y\mathbf{x}$ axis is stable. We solve for the segment below.

$$-1 < 1 + \eta v_t^T y\mathbf{x} + \frac{\eta^2 (v_t^T y\mathbf{x})^2}{4} < 1$$

$$\frac{-8}{\eta^2} < \left(\frac{4}{\eta} + v_t^T y\mathbf{x}\right) v_t^T y\mathbf{x} < 0$$

Since the $\frac{-8}{\eta^2}$ lower bound is not tight, we get that the condition is satisfied for $\frac{-4}{\eta} < v_t^T y\mathbf{x} < 0$. $\quad\square$

### C.1.2 SHARPNESS DERIVATION

In this section, we derive a closed-form formula for the sharpness and the other eigenvalues of the Loss Hessian. We first make the following claim about the sharpness.

**Claim C.3** (Closed-Form for Sharpness in One Datapoint Setting). *The sharpness for the model in the one datapoint setting is*

$$\frac{1}{2}\psi_t \|q_t\|_2^4 \|y\mathbf{x}\|_2^2 - \frac{1}{2}\phi_t v_t^T y\mathbf{x} + \psi_t \|q_t\|_2^2 \left(v_t^T y\mathbf{x}\right)^2$$

$$+ \sqrt{\begin{array}{l} \frac{1}{4}\psi_t^2 \|q_t\|_2^8 \|y\mathbf{x}\|_2^4 - \frac{7}{2}\phi_t \psi_t \|q_t\|_2^4 v_t^T y\mathbf{x} \|y\mathbf{x}\|_2^2 + \psi_t^2 \|q_t\|_2^6 \left(v_t^T y\mathbf{x}\right)^2 \|y\mathbf{x}\|_2^2 \\ + \frac{1}{4}\phi_t^2 \left(v_t^T y\mathbf{x}\right)^2 - \phi_t \psi_t \|q_t\|_2^2 \left(v_t^T y\mathbf{x}\right)^3 + \psi_t^2 \|q_t\|_2^4 \left(v_t^T y\mathbf{x}\right)^4 + 2\phi_t^2 \|q_t\|_2^2 \|y\mathbf{x}\|_2^2 \end{array}}$$

*where* $\phi_t = \left(1 + \exp\left(\|q_t\|_2^2 v_t^T y\mathbf{x}\right)\right)^{-1}$ *and* $\psi_t = \left(1 + \exp\left(\|q_t\|_2^2 v_t^T y\mathbf{x}\right)\right)^{-2} \exp\left(\|q_t\|_2^2 v_t^T y\mathbf{x}\right)$.

*Proof.* To solve for the sharpness, we first define the Loss Hessian. For $\theta_t = (q_t, u_t, v_t)$, we have

$$\nabla_{\theta_t} L(q_t, u_t, v_t) = \begin{bmatrix} \frac{\partial^2 L(q_t, u_t, v_t)}{\partial q_t \partial q_t^T} & \frac{\partial^2 L(q_t, u_t, v_t)}{\partial q_t \partial u_t^T} & \frac{\partial^2 L(q_t, u_t, v_t)}{\partial q_t \partial v_t^T} \\ \frac{\partial^2 L(q_t, u_t, v_t)}{\partial u_t \partial q_t^T} & \frac{\partial^2 L(q_t, u_t, v_t)}{\partial u_t \partial u_t^T} & \frac{\partial^2 L(q_t, u_t, v_t)}{\partial u_t \partial v_t^T} \\ \frac{\partial^2 L(q_t, u_t, v_t)}{\partial v_t \partial q_t^T} & \frac{\partial^2 L(q_t, u_t, v_t)}{\partial v_t \partial u_t^T} & \frac{\partial^2 L(q_t, u_t, v_t)}{\partial v_t \partial v_t^T} \end{bmatrix}$$

We solve for the eigenvalues using the technique developed by Singh & Hofmann (2024). First, we solve for $\det\left(\nabla_{\theta_t} L(q_t, u_t, v_t) - \lambda I_{2h+d}\right) = 0$. We form the matrix $\nabla_{\theta_t} L(q_t, u_t, v_t) - \lambda I_{2h+d}$ below.

$$\nabla_{\theta_t} L(q_t, u_t, v_t) - \lambda I_{2h+d}$$
$$= \begin{bmatrix} \psi_t \left(v_t^T y\mathbf{x}\right)^2 u_t u_t^T - \lambda I_h & -\phi_t v_t^T y\mathbf{x} I_h + \psi_t \left(v_t^T y\mathbf{x}\right)^2 u_t q_t^T & \left(-\phi_t + \psi_t q_t^T u_t v_t^T y\mathbf{x}\right) u_t y\mathbf{x}^T \\ -\phi_t v_t^T y\mathbf{x} I_h + \psi_t \left(v_t^T y\mathbf{x}\right)^2 q_t u_t^T & \psi_t \left(v_t^T y\mathbf{x}\right)^2 q_t q_t^T - \lambda I_h & \left(-\phi_t + \psi_t q_t^T u_t v_t^T y\mathbf{x}\right) q_t y\mathbf{x}^T \\ \left(-\phi_t + \psi_t q_t^T u_t v_t^T y\mathbf{x}\right) y\mathbf{x} u_t^T & \left(-\phi_t + \psi_t q_t^T u_t v_t^T y\mathbf{x}\right) y\mathbf{x} q_t^T & \psi_t \left(q_t^T u_t\right)^2 xx^T - \lambda I_d \end{bmatrix}$$

With our initialization assumption (i.e., $u_0 = q_0$, then $u_t = q_t$ by Claim 3.1) $\nabla_{\theta_t} L(q_t, u_t, v_t) - \lambda I_{2h+d}$ reduces to

$$= \begin{bmatrix} \psi_t \left(v_t^T y\mathbf{x}\right)^2 q_t q_t^T - \lambda I_h & -\phi_t v_t^T y\mathbf{x} I_h + \psi_t \left(v_t^T y\mathbf{x}\right)^2 q_t q_t^T & \left(-\phi_t + \psi_t \|q_t\|_2^2 v_t^T y\mathbf{x}\right) q_t y\mathbf{x}^T \\ -\phi_t v_t^T y\mathbf{x} I_h + \psi_t \left(v_t^T y\mathbf{x}\right)^2 q_t q_t^T & \psi_t \left(v_t^T y\mathbf{x}\right)^2 q_t q_t^T - \lambda I_h & \left(-\phi_t + \psi_t \|q_t\|_2^2 v_t^T y\mathbf{x}\right) q_t y\mathbf{x}^T \\ \left(-\phi_t + \psi_t \|q_t\|_2^2 v_t^T y\mathbf{x}\right) y\mathbf{x} q_t^T & \left(-\phi_t + \psi_t \|q_t\|_2^2 v_t^T y\mathbf{x}\right) y\mathbf{x} q_t^T & \psi_t \|q_t\|_2^4 xx^T - \lambda I_d \end{bmatrix}$$

Based on the pattern introduced in the $\nabla_{\theta_t} L(q_t, u_t, v_t) - \lambda I_{2h+d}$ by the initialization assumption, we perform two similarity transforms. We first define $A = \psi_t \left(v_t^T y\mathbf{x}\right)^2 q_t q_t^T - \lambda I_h$, $B = -\phi_t v_t^T y\mathbf{x} I_h + \psi_t \left(v_t^T y\mathbf{x}\right)^2 q_t q_t^T$, $C = \left(-\phi_t + \psi_t \|q_t\|_2^2 v_t^T y\mathbf{x}\right) q_t y\mathbf{x}^T$, and $D = \psi_t \|q_t\|_2^4 xx^T - \lambda I_d$. Then, we can rewrite $\nabla_{\theta_t} L(q_t, u_t, v_t) - \lambda I_{2h+d}$ as

$$\begin{bmatrix} A & B & C \\ B & A & C \\ C^T & C^T & D \end{bmatrix}$$

We proceed with the following similarity transforms.

$$\frac{1}{2}\begin{bmatrix} I_h & I_h & 0 \\ -I_h & I_h & 0 \\ 0 & 0 & \sqrt{2}I_d \end{bmatrix}\begin{bmatrix} A & B & C \\ B & A & C \\ C^T & C^T & D \end{bmatrix}\begin{bmatrix} I_h & -I_h & 0 \\ I_h & I_h & 0 \\ 0 & 0 & \sqrt{2}I_d \end{bmatrix}$$

$$=\frac{1}{2}\begin{bmatrix} I_h & I_h & 0 \\ -I_h & I_h & 0 \\ 0 & 0 & \sqrt{2}I_d \end{bmatrix}\begin{bmatrix} A+B & B-A & \sqrt{2}C \\ A+B & A-B & \sqrt{2}C \\ 2C^T & 0 & \sqrt{2}D \end{bmatrix}$$

$$=\frac{1}{2}\begin{bmatrix} 2(A+B) & 0 & 2\sqrt{2}C \\ 0 & 2(A-B) & 0 \\ 2\sqrt{2}C^T & 0 & 2D \end{bmatrix}$$

$$=\begin{bmatrix} A+B & 0 & \sqrt{2}C \\ 0 & A-B & 0 \\ \sqrt{2}C^T & 0 & D \end{bmatrix}$$

$$\begin{bmatrix} I_h & 0 & 0 \\ 0 & 0 & I_d \\ 0 & I_h & 0 \end{bmatrix}\begin{bmatrix} A+B & 0 & \sqrt{2}C \\ 0 & A-B & 0 \\ \sqrt{2}C^T & 0 & D \end{bmatrix}\begin{bmatrix} I_h & 0 & 0 \\ 0 & 0 & I_h \\ 0 & I_d & 0 \end{bmatrix}$$

$$=\begin{bmatrix} I_h & 0 & 0 \\ 0 & 0 & I_d \\ 0 & I_h & 0 \end{bmatrix}\begin{bmatrix} A+B & \sqrt{2}C & 0 \\ 0 & 0 & A-B \\ \sqrt{2}C^T & D & 0 \end{bmatrix}$$

$$=\begin{bmatrix} A+B & \sqrt{2}C & 0 \\ \sqrt{2}C^T & D & 0 \\ 0 & 0 & A-B \end{bmatrix}$$

After applying the similarity transforms, we solve $\det\left(\begin{bmatrix} A+B & \sqrt{2}C \\ \sqrt{2}C^T & D \end{bmatrix}\right)=0$ to get the sharpness.
To do so, we assume that $D$ is invertible. This means that $\lambda \neq \psi_t\|q_t\|_2^4\|y\mathbf{x}\|_2^2, 0$.

Then, $\det\left(\begin{bmatrix} A+B & \sqrt{2}C \\ \sqrt{2}C^T & D \end{bmatrix}\right) = \det(D)\det\left(A+B-2CD^{-1}C^T\right)$. So, we solve for $\det\left(A+B-2CD^{-1}C^T\right)=0$.

First, we solve for $D^{-1}$ using the Sherman-Morrison Formula:

$$D^{-1}=-\frac{1}{\lambda}I_d-\frac{\psi_t\|q_t\|_2^4 xx^T}{\lambda\left(\lambda-\psi_t\|q_t\|_2^4\|y\mathbf{x}\|_2^2\right)}$$

Then, we find that

$$A+B-2CD^{-1}C^T=2\left[\psi_t\left(v_t^T y\mathbf{x}\right)^2+\frac{\psi_t^2\|q_t\|_2^4\left(v_t^T y\mathbf{x}\right)^2\|y\mathbf{x}\|_2^2}{\lambda}+\frac{\phi_t^2\|y\mathbf{x}\|_2^2}{\lambda}-\frac{2\phi_t\psi_t\|q_t\|_2^2 v_t^T y\mathbf{x}\|y\mathbf{x}\|_2^2}{\lambda}\right.$$

$$\left.+\frac{\psi_t^3\|q_t\|_2^8\|y\mathbf{x}\|_2^4\left(v_t^T y\mathbf{x}\right)^2}{\lambda\left(\lambda-\psi_t\|q_t\|_2^4\|y\mathbf{x}\|_2^2\right)}+\frac{\psi_t\phi_t^2\|q_t\|_2^4\|y\mathbf{x}\|_2^4}{\lambda\left(\lambda-\psi_t\|q_t\|_2^4\|y\mathbf{x}\|_2^2\right)}-\frac{2\psi_t^2\phi_t\|q_t\|_2^6\|y\mathbf{x}\|_2^4 v_t^T y\mathbf{x}}{\lambda\left(\lambda-\psi_t\|q_t\|_2^4\|y\mathbf{x}\|_2^2\right)}\right]q_t q_t^T$$

$$-\left(\phi_t v_t^T y\mathbf{x}+\lambda\right)I_h$$

To solve for $\det\left(A + B - 2CD^{-1}C^T\right) = 0$, we first multiply by $\lambda\left(\lambda - \psi_t\|q_t\|_2^4\|y\mathbf{x}\|_2^2\right)$, which yields the following.

$$
\left(2\left[\psi_t\left(v_t^T y\mathbf{x}\right)^2\lambda\left(\lambda - \psi_t\|q_t\|_2^4\|y\mathbf{x}\|_2^2\right) + \psi_t^2\|q_t\|_2^4\left(v_t^T y\mathbf{x}\right)^2\|y\mathbf{x}\|_2^2\left(\lambda - \psi_t\|q_t\|_2^4\|y\mathbf{x}\|_2^2\right)\right.\right.
$$

$$
+ \phi_t^2\|y\mathbf{x}\|_2^2\left(\lambda - \psi_t\|q_t\|_2^4\|y\mathbf{x}\|_2^2\right) - 2\phi_t\psi_t\|q_t\|_2^2 v_t^T y\mathbf{x}\|y\mathbf{x}\|_2^2\left(\lambda - \psi_t\|q_t\|_2^4\|y\mathbf{x}\|_2^2\right) + \psi_t^3\|q_t\|_2^8\|y\mathbf{x}\|_2^4\left(v_t^T y\mathbf{x}\right)^2
$$

$$
\left.+ \psi_t\phi_t^2\|q_t\|_2^4\|y\mathbf{x}\|_2^4 - 2\psi_t^2\phi_t\|q_t\|_2^6\|y\mathbf{x}\|_2^4 v_t^T y\mathbf{x}\right]\|q_t\|_2^2 - \left(\phi_t v_t^T y\mathbf{x} + \lambda\right)\lambda\left(\lambda - \psi_t\|q_t\|_2^4\|y\mathbf{x}\|_2^2\right)\right)
$$

$$
\cdot\left(-\lambda\left(\phi_t v_t^T y\mathbf{x} + \lambda\right)\left(\lambda - \psi_t\|q_t\|_2^4\|y\mathbf{x}\|_2^2\right)\right)^{h-1} = 0
$$

We focus on the following factor from above.

$$
2\left[\psi_t\left(v_t^T y\mathbf{x}\right)^2\lambda\left(\lambda - \psi_t\|q_t\|_2^4\|y\mathbf{x}\|_2^2\right) + \psi_t^2\|q_t\|_2^4\left(v_t^T y\mathbf{x}\right)^2\|y\mathbf{x}\|_2^2\left(\lambda - \psi_t\|q_t\|_2^4\|y\mathbf{x}\|_2^2\right)\right.
$$

$$
+ \phi_t^2\|y\mathbf{x}\|_2^2\left(\lambda - \psi_t\|q_t\|_2^4\|y\mathbf{x}\|_2^2\right) - 2\phi_t\psi_t\|q_t\|_2^2 v_t^T y\mathbf{x}\|y\mathbf{x}\|_2^2\left(\lambda - \psi_t\|q_t\|_2^4\|y\mathbf{x}\|_2^2\right) + \psi_t^3\|q_t\|_2^8\|y\mathbf{x}\|_2^4\left(v_t^T y\mathbf{x}\right)^2
$$

$$
\left.+ \psi_t\phi_t^2\|q_t\|_2^4\|y\mathbf{x}\|_2^4 - 2\psi_t^2\phi_t\|q_t\|_2^6\|y\mathbf{x}\|_2^4 v_t^T y\mathbf{x}\right]\|q_t\|_2^2 - \lambda\left(\phi_t v_t^T y\mathbf{x} + \lambda\right)\left(\lambda - \psi_t\|q_t\|_2^4\|y\mathbf{x}\|_2^2\right) = 0
$$

We see that the above reduces down to the following.

$$
-\lambda\left(-2\lambda\psi_t\left(v_t^T y\mathbf{x}\right)^2\|q_t\|_2^2 - 2\phi_t^2\|y\mathbf{x}\|_2^2\|q_t\|_2^2 + 4\phi_t\psi_t\|q_t\|_2^4 v_t^T y\mathbf{x}\|y\mathbf{x}\|_2^2\right.
$$

$$
\left.+ \lambda\phi_t v_t^T y\mathbf{x} - \psi_t\phi_t\|q_t\|_2^4\|y\mathbf{x}\|_2^2 v_t^T y\mathbf{x} + \lambda^2 - \lambda\psi_t\|q_t\|_2^4\|y\mathbf{x}\|_2^2\right) = 0
$$

Since we assumed earlier that $\lambda \neq 0$, we solve the quadratic. From solving the quadratic, we get the following symmetric pair of eigenvalues.

$$
\lambda = \frac{1}{2}\psi_t\|q_t\|_2^4\|y\mathbf{x}\|_2^2 - \frac{1}{2}\phi_t v_t^T y\mathbf{x} + \psi_t\left(v_t^T y\mathbf{x}\right)^2\|q_t\|_2^2
$$

$$
\pm\sqrt{\begin{array}{l}\frac{1}{4}\psi_t^2\|q_t\|_2^8\|y\mathbf{x}\|_2^4 - \frac{7}{2}\phi_t\psi_t\|q_t\|_2^4 v_t^T y\mathbf{x}\|y\mathbf{x}\|_2^2 + \psi_t^2\|q_t\|_2^6\|y\mathbf{x}\|_2^2\left(v_t^T y\mathbf{x}\right)^2 \\ + \frac{1}{4}\phi_t^2\left(v_t^T y\mathbf{x}\right)^2 - \phi_t\psi_t\left(v_t^T y\mathbf{x}\right)^3\|q_t\|_2^2 + \psi_t^2\left(v_t^T y\mathbf{x}\right)^4\|q_t\|_2^4 + 2\phi_t^2\|y\mathbf{x}\|_2^2\|q_t\|_2^2\end{array}}
$$

where the plus direction is our sharpness. $\qquad\square$

In addition to the sharpness, we also have closed-forms for remaining eigenvalues and their multiplicity. We have $h$ copies of $\phi_t v_t^T y\mathbf{x}$, $h-1$ copies of $-\phi_t v_t^T y\mathbf{x}$, $d-1$ copies of $0$, and the following eigenvalues from our calculation above.

$$
\lambda = \frac{1}{2}\psi_t\|q_t\|_2^4\|y\mathbf{x}\|_2^2 - \frac{1}{2}\phi_t v_t^T y\mathbf{x} + \psi_t\left(v_t^T y\mathbf{x}\right)^2\|q_t\|_2^2
$$

$$
-\sqrt{\begin{array}{l}\frac{1}{4}\psi_t^2\|q_t\|_2^8\|y\mathbf{x}\|_2^4 - \frac{7}{2}\phi_t\psi_t\|q_t\|_2^4 v_t^T y\mathbf{x}\|y\mathbf{x}\|_2^2 + \psi_t^2\|q_t\|_2^6\|y\mathbf{x}\|_2^2\left(v_t^T y\mathbf{x}\right)^2 \\ + \frac{1}{4}\phi_t^2\left(v_t^T y\mathbf{x}\right)^2 - \phi_t\psi_t\left(v_t^T y\mathbf{x}\right)^3\|q_t\|_2^2 + \psi_t^2\left(v_t^T y\mathbf{x}\right)^4\|q_t\|_2^4 + 2\phi_t^2\|y\mathbf{x}\|_2^2\|q_t\|_2^2\end{array}}
$$

We note that $\phi_t v_t^T y\mathbf{x}$ comes from solving $\det(A - B)$, $-\phi_t v_t^T y\mathbf{x}$ comes from our approach above, and $0$ also comes from as similar approach as above, but by assuming the $A + B$ matrix is invertible instead of $D$ matrix.

### C.1.3 PROOF OF THEOREM 1

In this section we give the proof of Theorem 4.2. Recall that according to our setup, we always have $y = \pm 1$ is the label, $\|x\|_2^2 > 0$ (to avoid degenerate cases) and $\eta > 0$.

As we mentioned in Section 4, our problem can be simplified to a 2-dimensional system with parameters $\|q_t\|_2^2$ and $v_t^T y\mathbf{x}$. We partition the space for this two dimensional system based on two nullclines of $\|q_t\|_2^2$ and the boundary splits the space based on whether $\|q_t\|_2^2 v_t^T y\mathbf{x}$ grows or shrinks in the next time step. We also give supporting lemmas that illustrate the properties of the 4 regions, shown below in Figure 6.

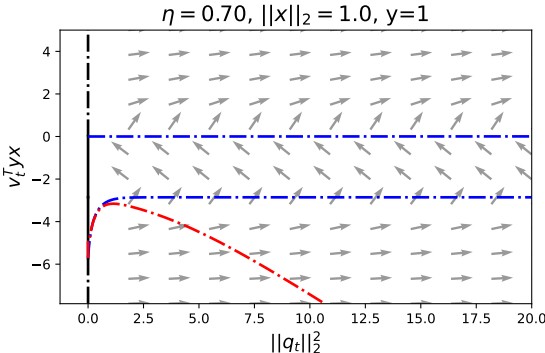

Figure 6: This is the phase space diagram for our one datapoint system for the case where $\eta = 0.70$, $\|\mathbf{x}\|_2^2 = 1$, and $y = 1$. The blue dashed curves represent the nullclines for $\|q_t\|_2^2$, where region 1 is above the top blue curve, region 2 is between both blue curves, and regions 3.1 and 3.2 are below the bottom blue curve. The red dashed curve represents the boundary where $\|q_t\|_2^2 v_t^T y\mathbf{x}$ shrinks inside the curve (region 3.2) and grows outside the curve (region 3.1). Since $\|q_t\|_2^2 = 0$ is also a fixed point of the system, we denote the solid black line portion as $\|q_t\|_2^2 = 0$ being stable, and the dashed black line portion as $\|q_t\|_2^2 = 0$ being unstable. We also include normalized vectors of the gradient field, shown as grey arrows, to provide some intuition for possible trajectories.

First, we provide the proof for Theorem 4.2. Then, we give the supporting lemmas.

For Theorem 4.2, our proof mainly focuses on the behavior of the loss in each region, since our definition for Edge of Stability focuses on specific attributes of the loss (i.e., it must have at least 2 local maxima). The intuition behind the proof is based on our understanding that the loss can only increase in region 3.2 (see Figure 6) and could potentially contribute to the Edge of Stability if we had some form of movement in and out of region 3.2. However, as seen in Figure 6, region 3.2 is upper bounded in the $v_t^T y\mathbf{x}$ direction. So, if for some $\eta$, the system "escapes" region 3.2 to regions 2 or 1, we cannot move back into region 3.2, since $v_t^T y\mathbf{x}$ is strictly increasing according to Lemma 4.1. So, our only potential candidate could be movement between regions 3.2 and 3.1. However, this is also not possible either, since region 3.2 does not appear to have an holes or significant indentations, and the strict growth of $\|q_t\|_2^2$ in regions 3.1 and 3.2 makes it that return into region 3.2 from region 3.1 is not possible. So, there would be no significant non-monotonic behavior in the loss. We now show the formal proof below.

*Proof of Theorem 4.2.* To prove the theorem, we first consider the phase space of our 2-dimensional system for this setting, consisting of $\|q_t\|_2^2$ and $v_t^T y\mathbf{x}$.

Based on our fixed point analysis, we divide up our phase space into four regions. Region 1 is defined as the area where $\|q_t\|_2^2 > 0$ and $v_t^T y\mathbf{x} \geq 0$. Region 2 is the area where $\|q_t\|_2^2 > 0, v_t^T y\mathbf{x}$ satisfies $-\frac{2}{\eta} < \left(1 + \exp\left(\|q_t\|_2^2 v_t^T y\mathbf{x}\right)\right)^{-1} v_t^T y\mathbf{x} < 0$. Region 3.1 is the area where $\|q_t\|_2^2 > 0, v_t^T y\mathbf{x}$ satisfies $\left(1 + \exp\left(\|q_t\|_2^2 v_t^T y\mathbf{x}\right)\right)^{-1} v_t^T y\mathbf{x} < -\frac{2}{\eta}$ and

$$\left(v_t^T y\mathbf{x}\right)^2 \left(2 + \eta \left(1 + \exp\left(\|q_t\|_2^2 v_t^T y\mathbf{x}\right)\right)^{-1} v_t^T y\mathbf{x}\right) + \|q_t\|_2^2 \|y\mathbf{x}\|_2^2 \left(1 + \eta \left(1 + \exp\left(\|q_t\|_2^2 v_t^T y\mathbf{x}\right)\right)^{-1} v_t^T y\mathbf{x}\right)^2 \geq 0.$$

Region 3.2 is the area where $\|q_t\|_2^2 > 0, v_t^T y\mathbf{x}$ satisfies $\left(1 + \exp\left(\|q_t\|_2^2 v_t^T y\mathbf{x}\right)\right)^{-1} v_t^T y\mathbf{x} < -\frac{2}{\eta}$ and

$$\left(v_t^T y\mathbf{x}\right)^2 \left(2 + \eta \left(1 + \exp\left(\|q_t\|_2^2 v_t^T y\mathbf{x}\right)\right)^{-1} v_t^T y\mathbf{x}\right) + \|q_t\|_2^2 \|y\mathbf{x}\|_2^2 \left(1 + \eta \left(1 + \exp\left(\|q_t\|_2^2 v_t^T y\mathbf{x}\right)\right)^{-1} v_t^T y\mathbf{x}\right)^2 < 0.$$

We proceed by proving that every possible training loss trajectory in the single datapoint setting does not meet the EOS conditions in our definition by considering the initialization $||q_0||_2^2 > 0, v_0^T y\mathbf{x}$ in each region. We don't consider any initialization with $||q_0||_2^2 = 0$, since that is a fixed point (see Claim C.2).

**Case** 1: $||q_0||_2^2 > 0, v_0^T y\mathbf{x}$ in region 1. We proceed by induction. For the base case, we know, by corollary 4.1.1, that $||q_1||_2^2 v_1^T y\mathbf{x} > ||q_0||_2^2 v_0^T y\mathbf{x}$. This means that $\log\left(1 + \left(-||q_1||_2^2 v_1^T y\mathbf{x}\right)\right) < \log\left(1 + \exp\left(||q_0||_2^2 v_0^T y\mathbf{x}\right)\right)$.

For the inductive case, we assume $||q_k||_2^2 > 0, v_k^T y\mathbf{x}$ in region 1. By corollary 4.1.1, we know that $||q_{k+1}||_2^2 v_{k+1}^T y\mathbf{x} > ||q_k||_2^2 v_k^T y\mathbf{x}$, meaning $\log\left(1 + \left(-||q_{k+1}||_2^2 v_{k+1}^T y\mathbf{x}\right)\right) < \log\left(1 + \exp\left(||q_k||_2^2 v_k^T y\mathbf{x}\right)\right)$

Since our loss is strictly decreasing in this case, it does not meet the EOS condition.

**Case** 2: $||q_0||_2^2 > 0, v_0^T y\mathbf{x}$ in region 2. We consider two possible trajectories based on this initialization. We know that these trajectories are exhaustive based on lemmas 4.1, C.5, and C.11.

Trajectory 1: there exists some $k \geq 1$, where $||q_k||_2^2 = 0$. Since we know that $||q_k||_2^2 = 0$ is a fixed point, we know that for any $l > k$, $||q_l||_2^2 = 0$. So, $\log\left(1 + \exp\left(-||q_l||_2^2 v_l^T y\mathbf{x}\right)\right) = \log\left(1 + \exp\left(-||q_k||_2^2 v_k^T y\mathbf{x}\right)\right) = \log\left(2\right)$.

Trajectory 2: there exists some $k \geq 1$, where $||q_k||_2^2 > 0$ and $v_k^T y\mathbf{x} \geq 0$. We know that our loss trajectory will be strictly decreasing for subsequent iterations by considering $||q_k||_2^2, v_k^T y\mathbf{x}$ as some initialization in region 1.

Now, for all iterations prior to $k$, we will show that our loss is strictly decreasing by induction. First, we consider the base case. By corollary 4.1.2, we know that $||q_1||_2^2 v_1^T y\mathbf{x} > ||q_0||_2^2 v_0^T y\mathbf{x}$. This means that $\log\left(1 + \left(-||q_1||_2^2 v_1^T y\mathbf{x}\right)\right) < \log\left(1 + \exp\left(||q_0||_2^2 v_0^T y\mathbf{x}\right)\right)$.

Next, we consider the inductive case. Assume $||q_j||_2^2 > 0, v_j^T y\mathbf{x}$ is in region 2, for $1 \leq j \leq k$. By corollary 4.1.2, we know that $||q_{j+1}||_2^2 v_{j+1}^T y\mathbf{x} > ||q_j||_2^2 v_j^T y\mathbf{x}$, meaning $\log\left(1 + \left(-||q_{j+1}||_2^2 v_{j+1}^T y\mathbf{x}\right)\right) < \log\left(1 + \exp\left(||q_j||_2^2 v_j^T y\mathbf{x}\right)\right)$

Since our loss can only decrease monotonically in this case, it does not meet the EOS condition.

**Case** 3: $||q_0||_2^2 > 0, v_0^T y\mathbf{x}$ in region 3.1. We consider two possible trajectories based on this initialization. We know that these trajectories are exhaustive based on lemmas 4.1, C.6, and C.11.

Trajectory 1: there exists some $k \geq 1$, where $||q_k||_2^2, v_k^T y\mathbf{x}$ jumps directly from region 3.1 to 1. We know that our loss trajectory will be strictly decreasing for subsequent iterations by considering $||q_k||_2^2, v_k^T y\mathbf{x}$ as some initialization in region 1.

Trajectory 2: there exists some $k \geq 1$, where $||q_k||_2^2, v_k^T y\mathbf{x}$ is in region 2. We know that our loss trajectory will be monotonically decreasing for subsequent iterations by considering $||q_k||_2^2, v_k^T y\mathbf{x}$ as some initialization in region 2.

Now, for all iterations prior to $k$, we will show that our loss is monotonically decreasing by induction. For the base case, we know by lemma C.7 that $||q_1||_2^2 v_1^T y\mathbf{x} \geq ||q_0||_2^2 v_0^T y\mathbf{x}$, meaning $\log\left(1 + \left(-||q_1||_2^2 v_1^T y\mathbf{x}\right)\right) \leq \log\left(1 + \exp\left(||q_0||_2^2 v_0^T y\mathbf{x}\right)\right)$.

Next, we consider the inductive case. Assume $||q_j||_2^2 > 0$, $v_j^T y\mathbf{x}$ is in region 3.1, for $1 \leq j \leq k$. We know by lemma C.7 that $||q_{j+1}||_2^2 v_{j+1}^T y\mathbf{x} \geq ||q_j||_2^2 v_j^T y\mathbf{x}$, meaning $\log\left(1 + \left(-||q_{j+1}||_2^2 v_{j+1}^T y\mathbf{x}\right)\right) \leq \log\left(1 + \exp\left(||q_j||_2^2 v_j^T y\mathbf{x}\right)\right)$.

Since our loss can only decrease monotonically in this case, it does not meet the EOS condition.

**Case** 4: $||q_0||_2^2 > 0$, $v_0^T y\mathbf{x}$ in region 3.2. We consider three possible trajectories based on this initialization. We know these trajectories are exhaustive based on lemmas 4.1, C.6, C.11, and C.12.

Trajectory 1: there exists some $k \geq 1$, where $||q_k||_2^2, v_k^T y\mathbf{x}$ jumps directly from region 3.2 to 1. We know that our loss trajectory will be strictly decreasing for subsequent iterations by considering $||q_k||_2^2, v_k^T y\mathbf{x}$ as some initialization in region 1.

Trajectory 2: there exists some $k \geq 1$, where $||q_k||_2^2, v_k^T y\mathbf{x}$ jumps directly from region 3.2 to 2. We know that our loss trajectory will be monotonically decreasing for subsequent iterations by considering $||q_k||_2^2, v_k^T y\mathbf{x}$ as some initialization in region 2.

Trajectory 3: there exists some $k \geq 1$, where $||q_k||_2^2, v_k^T y\mathbf{x}$ is in region 3.1. We know that our loss trajectory will be monotonically decreasing for subsequent iterations by considering $||q_k||_2^2, v_k^T y\mathbf{x}$ as some initialization in region 3.1.

Now, for all iterations prior to $k$, we will show that our loss is strictly increasing by induction. For the base case, we know by lemma C.7 that $||q_1||_2^2 v_1^T y\mathbf{x} < ||q_0||_2^2 v_0^T y\mathbf{x}$, meaning $\log\left(1 + \left(-||q_1||_2^2 v_1^T y\mathbf{x}\right)\right) > \log\left(1 + \exp\left(||q_0||_2^2 v_0^T y\mathbf{x}\right)\right)$.

Next, we consider the inductive case. Assume $||q_j||_2^2 > 0$, $v_j^T y\mathbf{x}$ is in region 3.2, for $1 \leq j \leq k$. We know by lemma C.7 that $||q_{j+1}||_2^2 v_{j+1}^T y\mathbf{x} < ||q_j||_2^2 v_j^T y\mathbf{x}$, meaning $\log\left(1 + \left(-||q_{j+1}||_2^2 v_{j+1}^T y\mathbf{x}\right)\right) > \log\left(1 + \exp\left(||q_j||_2^2 v_j^T y\mathbf{x}\right)\right)$.

Since our loss has only one local maximum in this case, it does not meet the EOS condition.

Since we have shown that our training loss can have at most one local maximum for every possible trajectory, it therefore does not exhibit EOS behavior according to our definition.

$\square$

Now, we show the supporting lemmas for Theorem 4.2. First, we prove lemma 4.1 that $v_t^T y\mathbf{x}$ is always monotonically increasing. In Lemma C.4, we prove that $||q_t||_2^2$ grows monotonically in region 1 (i.e., $v_t^T y\mathbf{x} \geq 0$). In Corollary 4.1.1, we prove that the loss is always decreasing in region 1. In Lemma C.5, we prove that $||q_t||_2^2$ shrinks in region 2 (see Figure 6). In Corollary 4.1.2, we prove that, despite $||q_t||_2^2$ shrinking (see lemma C.5), the loss is still decreasing in region 2. In Lemma C.6, we prove that for regions 3.1 and 3.2, $||q_t||_2^2$ is increasing. In Lemma C.7, we show how the boundary separating region 3.1 and 3.2 (see the red curve in Figure 6) causes the loss to worsen in region 3.2 but not in region 3.1. In Lemma C.8, we define a necessary condition for the system to be in region 3.2, which we use in later lemmas. In Lemma C.9, we prove that the separating boundary for regions 3.1 and region 3.2 is upper bounded in the $v_t^T y\mathbf{x}$ direction by the one of the $||q_t||_2^2$ nullclines (see Figure 6). In Lemma C.10, we prove an upper bound in the $v_t^T y\mathbf{x}$ direction for one of the nullclines of $||q_t||_2^2$, which we use in Lemma C.12. In Lemma C.11, we show that once the system leaves region 3.2, it is impossible to return back. Lastly, in Lemma C.12, we show that the system can escape region 3.2 in finite time.

First we show that $v_t^T y\mathbf{x}$ is always monotonically increasing, therefore it does not have a finite nullcline, and we don't partition the space according to this variable.

*Proof of Lemma 4.1.* Assume some $k, l \in \mathbb{N}$, $l > k$ and $\|q_k\|_2^2 > 0$. We consider the iterated map for $v_l^T y\mathbf{x}$ below.

$$
\begin{aligned}
v_l^T y\mathbf{x} &= v_{l-1}^T y\mathbf{x} + \eta \left(1 + \exp\left(\|q_{l-1}\|_2^2 v_{l-1}^T y\mathbf{x}\right)\right)^{-1} \|q_{l-1}\|_2^2 \|y\mathbf{x}\|_2^2 \\
&= v_k^T y\mathbf{x} + \eta \|y\mathbf{x}\|_2^2 \left(1 + \exp\left(\|q_k\|_2^2 v_k^T y\mathbf{x}\right)\right)^{-1} \|q_k\|_2^2 \\
&\quad + \eta \|y\mathbf{x}\|_2^2 \sum_{i=1}^{l-k-1} \left(1 + exp(\|q_{k+i}\|_2^2 v_{k+i}^T y\mathbf{x})\right)^{-1} \|q_{k+i}\|_2^2 \\
&\geq v_k^T y\mathbf{x} + \eta \|y\mathbf{x}\|_2^2 \left(1 + \exp\left(\|q_k\|_2^2 v_k^T y\mathbf{x}\right)\right)^{-1} \|q_k\|_2^2 \qquad \text{since } \forall i \in \{1, ..., l-k-1\}. \|q_{k+i}\|_2^2 \geq 0 \\
&> v_k^T y\mathbf{x} \qquad \text{since } \|q_k\|_2^2 > 0
\end{aligned}
$$

$\square$

Next we show that whenever $v_k^T y\mathbf{x} \geq 0$ the norm of $q_t$ is nondecreasing, this is the main property of region 1 (see Figure 6)

**Lemma C.4.** *For any $k, l \in \mathbb{N}$ such that $l > k$, if $\|q_k\|_2^2 > 0$, $v_k^T y\mathbf{x} \geq 0$, then $\|q_l\|_2^2 \geq \|q_k\|_2^2$.*

*Proof.* Assume some $k, l \in \mathbb{N}$, $l > k$, $\eta > 0$, $\|q_k\|_2^2 > 0$ and $v_k^T y\mathbf{x} \geq 0$.

We consider the iterated map for $\|q_l\|_2^2$ below.

$$
\begin{aligned}
\|q_l\|_2^2 &= \|q_{l-1}\|_2^2 \left(1 + \eta \left(1 + \exp\left(\|q_{l-1}\|_2^2 v_{l-1}^T y\mathbf{x}\right)\right)^{-1} v_{l-1}^T y\mathbf{x}\right)^2 \\
&= \|q_k\|_2^2 \prod_{i=0}^{l-k-1} \left(1 + \eta \left(1 + \exp\left(\|q_{k+i}\|_2^2 v_{k+i}^T y\mathbf{x}\right)\right)^{-1} v_{k+i}^T y\mathbf{x}\right)^2 \\
&\geq \|q_k\|_2^2 \qquad \text{by lemma 4.1}
\end{aligned}
$$

$\square$

As a simple corollary of the first two Lemmas, it's clear that in region 1 the output of the model is always improving.

*Proof of Corollary 4.1.1.* Assume some $k, l \in \mathbb{N}$, $l > k$, $\|q_k\|_2^2 > 0$ and $v_k^T y\mathbf{x} \geq 0$. We derive the following.

$$
\begin{aligned}
v_l^T y\mathbf{x} &> v_k^T y\mathbf{x} \qquad \text{by lemma 4.1} \\
\|q_l\|_2^2 &\geq \|q_k\|_2^2 \qquad \text{by lemma C.4} \\
\|q_l\|_2^2 v_l^T y\mathbf{x} &> \|q_k\|_2^2 v_k^T y\mathbf{x}
\end{aligned}
$$

$\square$

Now we consider region 2 of Figure 6. This region is sandwiched between two nullclines of $\|q_t\|_2^2$, and we show that norm of $q_t$ decreases in this region.

**Lemma C.5.** *For any $k \in \mathbb{N}$, if $\|q_k\|_2^2 > 0$, $v_k^T y\mathbf{x}$ satisfies*

$-\frac{2}{\eta} < \left(1 + \exp\left(\|q_k\|_2^2 v_k^T y\mathbf{x}\right)\right)^{-1} v_k^T y\mathbf{x} < 0$, *then* $\|q_{k+1}\|_2^2 < \|q_k\|_2^2$.

*Proof.* Assume some $k \in \mathbb{N}$ and $\|q_k\|_2^2 > 0$, $v_k^T y\mathbf{x}$ that satisfies

$-\frac{2}{\eta} < \left(1 + \exp\left(\|q_k\|_2^2 v_k^T y\mathbf{x}\right)\right)^{-1} v_k^T y\mathbf{x} < 0$. Consider the iterated map for $\|q_{k+1}\|_2^2$ below.

$$\|q_{k+1}\|_2^2 = \|q_k\|_2^2 \left(1 + \eta \left(1 + \exp\left(\|q_k\|_2^2 v_k^T y\mathbf{x}\right)\right)^{-1} v_k^T y\mathbf{x}\right)^2$$
$$< \|q_k\|_2^2$$

We get to the final step by our assumption that $-\frac{2}{\eta} < \left(1 + \exp\left(\|q_k\|_2^2 v_k^T y\mathbf{x}\right)\right)^{-1} v_k^T y\mathbf{x} < 0$. □

Note that even though in region 2, the norm of $q_t$ is decreasing, the loss function is actually still improving as Corollary 4.1.2 states, which we prove below.

*Proof of Corollary 4.1.2.* Assume some $k \in \mathbb{N}$ and $\|q_k\|_2^2 > 0, v_k^T y\mathbf{x}$ that satisfies $-\frac{2}{\eta} < \left(1 + \exp\left(\|q_k\|_2^2 v_k^T y\mathbf{x}\right)\right)^{-1} v_k^T y\mathbf{x} < 0$. We derive the following.

$$v_{k+1}^T y\mathbf{x} > v_k^T y\mathbf{x} \qquad \text{by lemma 4.1}$$
$$\|q_{k+1}\|_2^2 < \|q_k\|_2^2 \qquad \text{by lemma C.5}$$
$$\|q_k\|_2^2 v_k^T y\mathbf{x} < \|q_{k+1}\|_2^2 v_{k+1}^T y\mathbf{x}$$

The final step comes from our assumption that $-\frac{2}{\eta} < \left(1 + \exp\left(\|q_k\|_2^2 v_k^T y\mathbf{x}\right)\right)^{-1} v_k^T y\mathbf{x} < 0$. □

Finally, we give lemmas related to region 3.1 and 3.2 in Figure 6. In these regions, the norm of $q_t$ is once again increasing.

**Lemma C.6.** *For any $k \in \mathbb{N}$, if $\|q_k\|_2^2 > 0, v_k^T y\boldsymbol{x}$ satisfies $\left(1 + \exp\left(\|q_k\|_2^2 v_k^T y\boldsymbol{x}\right)\right)^{-1} v_k^T y\boldsymbol{x} < -\frac{2}{\eta}$, then $\|q_{k+1}\|_2^2 > \|q_k\|_2^2$.*

*Proof.* Assume some $k \in \mathbb{N}$ and $\|q_k\|_2^2 > 0, v_k^T y\boldsymbol{x}$ that satisfies $\left(1 + \exp\left(\|q_k\|_2^2 v_k^T y\mathbf{x}\right)\right)^{-1} v_k^T y\mathbf{x} < -\frac{2}{\eta}$. We derive the following.

$$\|q_{k+1}\|_2^2 = \|q_k\|_2^2 \left(1 + \eta \left(1 + \exp\left(\|q_k\|_2^2 v_k^T y\mathbf{x}\right)\right)^{-1} v_k^T y\mathbf{x}\right)^2$$
$$> \|q_k\|_2^2$$

The final step is due to our assumption $\left(1 + \exp\left(\|q_k\|_2^2 v_k^T y\mathbf{x}\right)\right)^{-1} v_k^T y\mathbf{x} < -\frac{2}{\eta}$.

□

In region 3, the loss function may move in different directions, so we further partition the region into 3.1 and 3.2 according to whether the loss function is improving. In region 3.2, we show that the loss worsens by the following lemma.

**Lemma C.7.** *For any $k \in \mathbb{N}$ such that $\|q_k\|_2^2 > 0, v_k^T y\boldsymbol{x}$,*

$$\left(v_k^T y\boldsymbol{x}\right)^2 \left(2 + \eta \left(1 + exp(\|q_k\|_2^2 v_k^T y\boldsymbol{x})\right)^{-1} v_k^T y\boldsymbol{x}\right) + \|q_k\|_2^2 \|y\boldsymbol{x}\|_2^2 \left(1 + \eta \left(1 + \exp\left(\|q_k\|_2^2 v_k^T y\boldsymbol{x}\right)\right)^{-1} v_k^T y\boldsymbol{x}\right)^2 \geq 0$$

*if and only if $\|q_{k+1}\|_2^2 v_{k+1}^T y\boldsymbol{x} \geq \|q_k\|_2^2 v_k^T y\boldsymbol{x}$.*

*Proof.* First, we prove the forward direction of the biconditional. Assume some $k \in \mathbb{N}$ and $\|q_k\|_2^2 > 0, v_k^T y\mathbf{x}$ that satisfies

$$\left(v_k^T y\mathbf{x}\right)^2 \left(2 + \eta \left(1 + exp(\|q_k\|_2^2 v_k^T y\mathbf{x})\right)^{-1} v_k^T y\mathbf{x}\right) + \|q_k\|_2^2 \|y\mathbf{x}\|_2^2 \left(1 + \eta \left(1 + \exp\left(\|q_k\|_2^2 v_k^T y\mathbf{x}\right)\right)^{-1} v_k^T y\mathbf{x}\right)^2 \geq 0.$$

We derive the following.

$$\|q_{k+1}\|_2^2 v_{k+1}^T y\mathbf{x} - \|q_k\|_2^2 v_k^T y\mathbf{x}$$

$$= v_k^T y\mathbf{x} \left( \|q_k\|_2^2 + 2\eta \left(1 + \exp\left(\|q_k\|_2^2 v_k^T y\mathbf{x}\right)\right)^{-1} \|q_k\|_2^2 v_k^T y\mathbf{x} + \eta^2 \left(1 + \exp\left(\|q_k\|_2^2 v_k^T y\mathbf{x}\right)\right)^{-2} \|q_k\|_2^2 \left(v_k^T y\mathbf{x}\right)^2 \right)$$

$$+ \eta \left(1 + \exp\left(\|q_k\|_2^2 v_k^T y\mathbf{x}\right)\right)^{-1} \|q_k\|_2^2 \|y\mathbf{x}\|_2^2 \left( \|q_k\|_2^2 + 2\eta \left(1 + \exp\left(\|q_k\|_2^2 v_k^T y\mathbf{x}\right)\right)^{-1} \|q_k\|_2^2 v_k^T y\mathbf{x} \right.$$

$$+ \eta^2 \left(1 + \exp\left(\|q_k\|_2^2 v_k^T y\mathbf{x}\right)\right)^{-2} \|q_k\|_2^2 \left(v_k^T y\mathbf{x}\right)^2 \right) - \|q_k\|_2^2 v_k^T y\mathbf{x}$$

$$= 2\eta \left(1 + \exp\left(\|q_k\|_2^2 v_k^T y\mathbf{x}\right)\right)^{-1} \|q_k\|_2^2 \left(v_k^T y\mathbf{x}\right)^2 + \eta^2 \left(1 + \exp\left(\|q_k\|_2^2 v_k^T y\mathbf{x}\right)\right)^{-2} \|q_k\|_2^2 \left(v_k^T y\mathbf{x}\right)^3$$

$$+ \eta \|q_k\|_2^4 \|y\mathbf{x}\|_2^2 \left(1 + \exp\left(\|q_k\|_2^2 v_k^T y\mathbf{x}\right)\right)^{-1} + 2\eta^2 \left(1 + \exp\left(\|q_k\|_2^2 v_k^T y\mathbf{x}\right)\right)^{-2} \|q_k\|_2^4 v_k^T y\mathbf{x} \|y\mathbf{x}\|_2^2$$

$$+ \eta^3 \left(1 + \exp\left(\|q_k\|_2^2 v_k^T y\mathbf{x}\right)\right)^{-3} \|q_k\|_2^4 \left(v_k^T y\mathbf{x}\right)^2 \|y\mathbf{x}\|_2^2$$

$$= \eta \|q_k\|_2^2 \left(1 + \exp\left(\|q_k\|_2^2 v_k^T y\mathbf{x}\right)\right)^{-1} \left( \left(v_k^T y\mathbf{x}\right)^2 \left(2 + \eta \left(1 + \exp\left(\|q_k\|_2^2 v_k^T y\mathbf{x}\right)\right)^{-1} v_k^T y\mathbf{x}\right) \right.$$

$$+ \|q_k\|_2^2 \|y\mathbf{x}\|_2^2 \left(1 + \eta \left(1 + \exp\left(\|q_k\|_2^2 v_k^T y\mathbf{x}\right)\right)^{-1} v_k^T y\mathbf{x}\right)^2 \right)$$

$$\geq 0$$

Now, we consider the reverse direction of the biconditional, which we will prove by contrapositive. From our above derivation, we already know that $\|q_{k+1}\|_2^2 v_{k+1}^T y\mathbf{x} - \|q_k\|_2^2 v_k^T y\mathbf{x} < 0$, since we assumed that $\|q_k\|_2^2 > 0$ and

$$\left(v_k^T y\mathbf{x}\right)^2 \left(2 + \eta \left(1 + exp(\|q_k\|_2^2 v_k^T y\mathbf{x})\right)^{-1} v_k^T y\mathbf{x}\right) + \|q_k\|_2^2 \|y\mathbf{x}\|_2^2 \left(1 + \eta \left(1 + \exp\left(\|q_k\|_2^2 v_k^T y\mathbf{x}\right)\right)^{-1} v_k^T y\mathbf{x}\right)^2 < 0.$$

$$\square$$

The boundary separating regions 3.1 and 3.2 is not easy to express explicitly, but we do show an upperbound for the boundary in the following lemmas. In the lemma below, we first show that the separating boundary for regions 3.1 and 3.2 is upper bounded in the $v_t^T y\mathbf{x}$ direction by one of the nullclines of $\|q_t\|$ (see the red and blue curves in Figure 6).

**Lemma C.8.** *A necessary condition for* $\|q\|_2^2 > 0, v^T y\boldsymbol{x}$ *to satisfy*

$$\left(v^T y\boldsymbol{x}\right)^2 \left(2 + \eta \left(1 + \exp(\|q\|_2^2 v_k^T y\boldsymbol{x})\right)^{-1} v^T y\boldsymbol{x}\right) + \|q\|_2^2 \|y\boldsymbol{x}\|_2^2 \left(1 + \eta \left(1 + \exp\left(\|q\|_2^2 v^T y\boldsymbol{x}\right)\right)^{-1} v^T y\boldsymbol{x}\right)^2 < 0,$$

*is that* $\left(1 + \exp\left(\|q\|_2^2 v^T y\boldsymbol{x}\right)\right)^{-1} v^T y\boldsymbol{x} < -\frac{2}{\eta}$.

*Proof.* Assume that some $\|q\|_2^2 > 0$.

We will show that if $\left(1 + \exp\left(\|q\|_2^2 v^T y\mathbf{x}\right)\right)^{-1} v^T y\mathbf{x} \geq -\frac{2}{\eta}$, then

$$\left(v_k^T y\mathbf{x}\right)^2 \left(2 + \eta \left(1 + exp(\|q_k\|_2^2 v_k^T y\mathbf{x})\right)^{-1} v_k^T y\mathbf{x}\right) + \|q_k\|_2^2 \|y\mathbf{x}\|_2^2 \left(1 + \eta \left(1 + \exp\left(\|q_k\|_2^2 v_k^T y\mathbf{x}\right)\right)^{-1} v_k^T y\mathbf{x}\right)^2 \geq 0.$$

We derive the following.

$$\left(v^T y\mathbf{x}\right)^2 \left(2 + \eta \left(1 + \exp\left(\|q\|_2^2 v^T y\mathbf{x}\right)\right)^{-1} v^T y\mathbf{x}\right) \geq 0 \qquad\qquad \text{by our assumption}$$

$$\|q\|_2^2 \|y\mathbf{x}\|_2^2 \left(1 + \eta \left(1 + \exp\left(\|q\|_2^2 v^T y\mathbf{x}\right)\right)^{-1} v^T y\mathbf{x}\right)^2 \geq 0$$

$$\left(v^T y\mathbf{x}\right)^2 \left(2 + \eta \left(1 + \exp\left(\|q\|_2^2 v^T y\mathbf{x}\right)\right)^{-1} v^T y\mathbf{x}\right)$$

$$+ \|q\|_2^2 \|y\mathbf{x}\|_2^2 \left(1 + \eta \left(1 + \exp\left(\|q\|_2^2 v^T y\mathbf{x}\right)\right)^{-1} v^T y\mathbf{x}\right)^2 \geq 0.$$

$\square$

In the following lemma, we show an upper bound in the $v_t^T y\mathbf{x}$ direction one of the nullclines of $\|q_t\|_2^2$, which is also an upper bound for the boundary separating regions 3.1 and 3.2.

**Lemma C.9.** *For any $\|q\|_2^2 > 0$, the $v_1^T y\boldsymbol{x}$ that satisfies*

$$\left(v_1^T y\boldsymbol{x}\right)^2 \left(2 + \eta \left(1 + \exp\left(\|q\|_2^2 v_1^T y\boldsymbol{x}\right)\right)^{-1} v_1^T y\boldsymbol{x}\right) + \|q\|_2^2 \|y\boldsymbol{x}\|_2^2 \left(1 + \eta \left(1 + \exp\left(\|q\|_2^2 v_1^T y\boldsymbol{x}\right)\right)^{-1} v_1^T y\boldsymbol{x}\right)^2 < 0$$

*is less than the $v_2^T y\boldsymbol{x}$ that satisfies $\left(1 + \exp\left(\|q\|_2^2 v_2^T y\boldsymbol{x}\right)\right)^{-1} v_2^T y\boldsymbol{x} = -\frac{2}{\eta}$.*

*Proof.* Consider some $\|q\|_2^2 > 0$, $v_1^T y\mathbf{x}$ that satisfies

$$\left(v_1^T y\mathbf{x}\right)^2 \left(2 + \eta \left(1 + \exp\left(\|q\|_2^2 v_1^T y\mathbf{x}\right)\right)^{-1} v_1^T y\mathbf{x}\right) + \|q\|_2^2 \|y\mathbf{x}\|_2^2 \left(1 + \eta \left(1 + \exp\left(\|q\|_2^2 v_1^T y\mathbf{x}\right)\right)^{-1} v_1^T y\mathbf{x}\right)^2 < 0,$$

and $v_2^T y\mathbf{x}$ that satisfies $\left(1 + \exp\left(\|q\|_2^2 v_2^T y\mathbf{x}\right)\right)^{-1} v_2^T y\mathbf{x} = -\frac{2}{\eta}$. Assume for the sake of contradiction that $v_2^T y\mathbf{x} \le v_1^T y\mathbf{x}$. Then,

$$\left(1 + \exp\left(\|q\|_2^2 v_2^T y\mathbf{x}\right)\right)^{-1} v_2^T y\mathbf{x} \le \left(1 + \exp\left(\|q\|_2^2 v_1^T y\mathbf{x}\right)\right)^{-1} v_2^T y\mathbf{x} \qquad \text{by our assumption}$$

$$\left(1 + \exp\left(\|q\|_2^2 v_1^T y\mathbf{x}\right)\right)^{-1} v_1^T y\mathbf{x} \ge \left(1 + \exp\left(\|q\|_2^2 v_2^T y\mathbf{x}\right)\right)^{-1} v_2^T y\mathbf{x} = -\frac{2}{\eta} \quad \text{since } v_2^T y\mathbf{x} \le v_1^T y\mathbf{x}$$

This contradicts the necessary condition we determined in lemma C.8. $\square$

In the following lemma, we show an upper bound in the $v_t^T y\mathbf{x}$ direction for a nullcline of $\|q_t\|_2^2$ (see the bottom blue curve in Figure 6), which is also an upper bound for the boundary separating regions 3.1 and 3.2.

**Lemma C.10.** *For any $\|q\|_2^2 > 0$, the $v_t^T y\boldsymbol{x}$ that satisfies $\left(1 + \exp\left(\|q\|_2^2 v^T y\boldsymbol{x}\right)\right)^{-1} v^T y\boldsymbol{x} = -\frac{2}{\eta}$ is less than $-\frac{2}{\eta}$.*

*Proof.* First, we will show that for $0 < \|q_1\|_2^2 < \|q_2\|_2^2$ and $v_1^T y\mathbf{x}, v_2^T y\mathbf{x}$ which satisfy $\left(1 + \exp\left(\|q_1\|_2^2 v_1^T y\mathbf{x}\right)\right)^{-1} v_1^T y\mathbf{x} = \left(1 + \exp\left(\|q\|_2^2 v_1^T y\mathbf{x}\right)\right)^{-1} v_1^T y\mathbf{x} = -\frac{2}{\eta}$, we get that $v_1^T y\mathbf{x} < v_2^T y\mathbf{x}$. Assume for the sake of contradiction that $v_1^T y\mathbf{x} \ge v_2^T y\mathbf{x}$. Then,

$$\|q_1\|_2^2 v_1^T y\mathbf{x} > \|q_2\|_2^2 v_2^T y\mathbf{x} \qquad \text{since } v_1^T y\mathbf{x} \ge v_2^T y\mathbf{x} \text{ and } \|q_1\|_2^2 < \|q_2\|_2^2$$

$$\left(1 + \exp\left(\|q_1\|_2^2 v_1^T y\mathbf{x}\right)\right)^{-1} v_1^T y\mathbf{x} > \left(1 + \exp\left(\|q_2\|_2^2 v_2^T y\mathbf{x}\right)\right)^{-1} v_2^T y\mathbf{x},$$

where the last step comes from our assumption that $v_1^T y\mathbf{x} \ge v_2^T y\mathbf{x}$ and $\left(1 + \exp\left(\|q_1\|_2^2 v_1^T y\mathbf{x}\right)\right)^{-1} v_1^T y\mathbf{x} = -\frac{2}{\eta}$. Now that we have proven the above, we consider $\|q\|_2^2 \to \infty$. Then, we see that $\left(1 + \exp\left(\|q\|_2^2 v^T y\mathbf{x}\right)\right)^{-1} \to 1$. So, $\left(1 + \exp\left(\|q\|_2^2 v^T y\mathbf{x}\right)\right)^{-1} v^T y\mathbf{x} = -\frac{2}{\eta}$ becomes $v^T y\mathbf{x} = -\frac{2}{\eta}$. $\square$

We are now ready to show that once the system leaves region 3.2, it cannot go back.

**Lemma C.11.** *For any $k \in \mathbb{N}$, if $\|q_k\|_2^2 > 0$, $v_k^T y\boldsymbol{x}$ satisfies*

$$\left(v_k^T y\boldsymbol{x}\right)^2 \left(2 + \eta \left(1 + \exp\left(\|q_k\|_2^2 v_k^T y\boldsymbol{x}\right)\right)^{-1} v_k^T y\boldsymbol{x}\right) + \|q_k\|_2^2 \|y\boldsymbol{x}\|_2^2 \left(1 + \eta \left(1 + \exp\left(\|q_k\|_2^2 v_k^T y\boldsymbol{x}\right)\right)^{-1} v_k^T y\boldsymbol{x}\right)^2 \ge 0,$$

*then $\left(\|q_{k+1}\|_2^2 > 0, v_{k+1}^T y\boldsymbol{x}\right)$ satisfies*

$$\left(v_{k+1}^T y\boldsymbol{x}\right)^2 \left(2 + \eta \left(1 + \exp\left(\|q_{k+1}\|_2^2 v_{k+1}^T y\boldsymbol{x}\right)\right)^{-1} v_{k+1}^T y\boldsymbol{x}\right)$$

$$+ \|q_{k+1}\|_2^2 \|y\boldsymbol{x}\|_2^2 \left(1 + \eta \left(1 + \exp\left(\|q_{k+1}\|_2^2 v_{k+1}^T y\boldsymbol{x}\right)\right)^{-1} v_{k+1}^T y\boldsymbol{x}\right)^2 \ge 0.$$

*Proof.* Assume some $k \in \mathbb{N}$ and $\|q_k\|_2^2 > 0, v_k^T y\mathbf{x}$ that satisfies

$$\left(v_k^T y\mathbf{x}\right)^2 \left(2 + \eta \left(1 + \exp\left(\|q_k\|_2^2 v_k^T y\mathbf{x}\right)\right)^{-1} v_k^T y\mathbf{x}\right) + \|q_k\|_2^2 \|y\mathbf{x}\|_2^2 \left(1 + \eta \left(1 + \exp\left(\|q_k\|_2^2 v_k^T y\mathbf{x}\right)\right)^{-1} v_k^T y\mathbf{x}\right)^2 \geq 0.$$

We consider 3 cases for $\|q_k\|_2^2 > 0, v_k^T y\mathbf{x}$.

Case 1: $v_k^T y\mathbf{x} \geq 0$. Then, by lemma 4.1, we know that $v_{k+1}^T y\mathbf{x} > 0$. So, $\left(1 + \exp\left(\|q_{k+1}\|_2^2 v_{k+1}^T y\mathbf{x}\right)\right)^{-1} v_{k+1}^T y\mathbf{x} > 0 > -\frac{2}{\eta}$. This violates the necessary condition that we derived in lemma C.8.

Case 2: $-\frac{2}{\eta} \leq \left(1 + \exp\left(\|q_k\|_2^2 v_k^T y\mathbf{x}\right)\right)^{-1} v_k^T y\mathbf{x} < 0$.

We derive the following.

$$\|q_{k+1}\|_2^2 v_{k+1}^T y\mathbf{x} \geq \|q_k\|_2^2 v_k^T y\mathbf{x} \qquad \text{by lemma C.7}$$

$$v_k^T y\mathbf{x} \left(1 + \exp\left(\|q_{k+1}\|_2^2 v_{k+1}^T y\mathbf{x}\right)\right)^{-1} \geq v_k^T y\mathbf{x} \left(1 + \exp\left(\|q_k\|_2^2 v_k^T y\mathbf{x}\right)\right)^{-1} \qquad \text{since } v_k^T y\mathbf{x} < 0$$

$$v_{k+1}^T y\mathbf{x} \left(1 + \exp\left(\|q_{k+1}\|_2^2 v_{k+1}^T y\mathbf{x}\right)\right)^{-1} > \left(1 + \exp\left(\|q_{k+1}\|_2^2 v_{k+1}^T y\mathbf{x}\right)\right)^{-1} v_k^T y\mathbf{x} \qquad \text{by lemma 4.1}$$

$$-\frac{2}{\eta} \leq v_k^T y\mathbf{x} \left(1 + \exp\left(\|q_k\|_2^2 v_k^T y\mathbf{x}\right)\right)^{-1} < v_{k+1}^T y\mathbf{x} \left(1 + \exp\left(\|q_{k+1}\|_2^2 v_{k+1}^T y\mathbf{x}\right)\right)^{-1}.$$

This also violates the necessary condition that we derived in lemma C.8.

Case 3: $\left(1 + \exp\left(\|q_k\|_2^2 v_k^T y\mathbf{x}\right)\right)^{-1} v_k^T y\mathbf{x} < -\frac{2}{\eta}$.

If $-\frac{2}{\eta} \leq \left(1 + \exp\left(\|q_{k+1}\|_2^2 v_{k+1}^T y\mathbf{x}\right)\right)^{-1} v_{k+1}^T y\mathbf{x}$, then we use the violation of the necessary condition derived in lemma C.8 to conclude. Now, we assume $\left(1 + \exp\left(\|q_{k+1}\|_2^2 v_{k+1}^T y\mathbf{x}\right)\right)^{-1} v_{k+1}^T y\mathbf{x} < -\frac{2}{\eta}$. We start by deriving a relation between

$\left(v_k^T y\mathbf{x}\right)^2 \left(2 + \eta \left(1 + \exp\left(\|q_k\|_2^2 v_k^T y\mathbf{x}\right)\right)^{-1} v_k^T y\mathbf{x}\right)$ and $\left(v_{k+1}^T y\mathbf{x}\right)^2 \left(2 + \eta \left(1 + \left(\|q_{k+1}\|_2^2 v_{k+1}^T y\mathbf{x}\right)\right)^{-1} v_{k+1}^T y\mathbf{x}\right)$.

$$\|q_{k+1}\|_2^2 v_{k+1}^T y\mathbf{x} \geq \|q_k\|_2^2 v_k^T y\mathbf{x} \qquad \text{by lemma C.7}$$

$$v_{k+1}^T y\mathbf{x} \left(1 + \exp\left(\|q_{k+1}\|_2^2 v_{k+1}^T y\mathbf{x}\right)\right)^{-1}$$
$$> v_k^T y\mathbf{x} \left(1 + \exp\left(\|q_k\|_2^2 v_k^T y\mathbf{x}\right)\right)^{-1} \qquad \text{since } v_k^T y\mathbf{x} < 0 \text{ and lemma 4.1}$$

$$\left(v_{k+1}^T y\mathbf{x}\right)^2 \left(2 + \eta \left(1 + \exp\left(\|q_{k+1}\|_2^2 v_{k+1}^T y\mathbf{x}\right)\right)^{-1} v_{k+1}^T y\mathbf{x}\right)$$
$$> \left(v_{k+1}^T y\mathbf{x}\right)^2 \left(2 + \eta \left(1 + \exp\left(\|q_k\|_2^2 v_k^T y\mathbf{x}\right)\right)^{-1} v_k^T y\mathbf{x}\right) \qquad \text{by lemma 4.1}$$

$$\left(v_{k+1}^T y\mathbf{x}\right)^2 \left(2 + \eta \left(1 + \exp\left(\|q_k\|_2^2 v_k^T y\mathbf{x}\right)\right)^{-1} v_k^T y\mathbf{x}\right)$$
$$> \left(v_k^T y\mathbf{x}\right)^2 \left(2 + \eta \left(1 + \exp\left(\|q_k\|_2^2 v_k^T y\mathbf{x}\right)\right)^{-1} v_k^T y\mathbf{x}\right) \qquad \text{since } v_k^T y\mathbf{x}, v_{k+1}^T y\mathbf{x} < 0 \text{ and lemma 4.1}$$

$$\left(v_k^T y\mathbf{x}\right)^2 \left(2 + \eta \left(1 + \exp\left(\|q_k\|_2^2 v_k^T y\mathbf{x}\right)\right)^{-1} v_k^T y\mathbf{x}\right)$$
$$< \left(v_{k+1}^T y\mathbf{x}\right)^2 \left(2 + \eta \left(1 + \exp\left(\|q_{k+1}\|_2^2 v_{k+1}^T y\mathbf{x}\right)\right)^{-1} v_{k+1}^T y\mathbf{x}\right).$$

Now, we derive a relation between $\|q_k\|_2^2\|y\mathbf{x}\|_2^2 \left(1 + \eta \left(1 + \exp\left(\|q_k\|_2^2 v_k^T y\mathbf{x}\right)\right)^{-1} v_k^T y\mathbf{x}\right)^2$

and $\|q_{k+1}\|_2^2\|y\mathbf{x}\|_2^2 \left(1 + \eta \left(1 + \exp\left(\|q_{k+1}\|_2^2 v_{k+1}^T y\mathbf{x}\right)\right)^{-1} v_{k+1}^T y\mathbf{x}\right)^2$. With our assumption

$\left(1 + \exp\left(\|q_{k+1}\|_2^2 v_{k+1}^T y\mathbf{x}\right)\right)^{-1} v_{k+1}^T y\mathbf{x} < -\frac{2}{\eta}$, we know that

$$\|q_{k+1}\|_2^2\|y\mathbf{x}\|_2^2 \left(1 + \eta \left(1 + \exp\left(\|q_{k+1}\|_2^2 v_{k+1}^T y\mathbf{x}\right)\right)^{-1} v_{k+1}^T y\mathbf{x}\right)^2$$

$$= \left(\|q_k\|_2^2\|y\mathbf{x}\|_2^2 \left(1 + \eta \left(1 + \exp\left(\|q_k\|_2^2 v_k^T y\mathbf{x}\right)\right)^{-1} v_k^T y\mathbf{x}\right)^2\right) \left(1 + \eta \left(1 + \exp\left(\|q_{k+1}\|_2^2 v_{k+1}^T y\mathbf{x}\right)\right)^{-1} v_{k+1}^T y\mathbf{x}\right)^2$$

$$> \|q_k\|_2^2\|y\mathbf{x}\|_2^2 \left(1 + \eta \left(1 + \exp\left(\|q_k\|_2^2 v_k^T y\mathbf{x}\right)\right)^{-1} v_k^T y\mathbf{x}\right)^2.$$

Then we get that

$$\left(v_{k+1}^T y\mathbf{x}\right)^2 \left(2 + \eta \left(1 + \exp\left(\|q_{k+1}\|_2^2 v_{k+1}^T y\mathbf{x}\right)\right)^{-1} v_{k+1}^T y\mathbf{x}\right)$$

$$+ \|q_{k+1}\|_2^2\|y\mathbf{x}\|_2^2 \left(1 + \eta \left(1 + \exp\left(\|q_{k+1}\|_2^2 v_{k+1}^T y\mathbf{x}\right)\right)^{-1} v_{k+1}^T y\mathbf{x}\right)^2$$

$$> \left(v_k^T y\mathbf{x}\right)^2 \left(2 + \eta \left(1 + \exp\left(\|q_k\|_2^2 v_k^T y\mathbf{x}\right)\right)^{-1} v_k^T y\mathbf{x}\right)$$

$$+ \|q_k\|_2^2\|y\mathbf{x}\|_2^2 \left(1 + \eta \left(1 + \exp\left(\|q_k\|_2^2 v_k^T y\mathbf{x}\right)\right)^{-1} v_k^T y\mathbf{x}\right)^2$$

$$\geq 0.$$

$\square$

Finally, we show that for any starting point in region 3.2, the trajectory will eventually "escape" 3.2 and reach other regions.

**Lemma C.12.** *For any $k \in \mathbb{N}$, if $\|q_k\|_2^2 > 0$, $v_k^T y\boldsymbol{x}$ satisfies*

$$\left(v_k^T y\boldsymbol{x}\right)^2 \left(2 + \eta \left(1 + \exp\left(\|q_k\|_2^2 v_k^T y\boldsymbol{x}\right)\right)^{-1} v_k^T y\boldsymbol{x}\right) + \|q_k\|_2^2\|y\boldsymbol{x}\|_2^2 \left(1 + \eta \left(1 + \exp\left(\|q_k\|_2^2 v_k^T y\boldsymbol{x}\right)\right)^{-1} v_k^T y\boldsymbol{x}\right)^2 < 0,$$

*then there exists some $l \in \mathbb{N}$, where $k < l \leq \lceil \frac{-\left(\frac{2}{\eta} + v_k^T y\boldsymbol{x}\right)\left(1 + \exp\left(\|q_k\|_2^2 v_k^T y\boldsymbol{x}\right)\right)}{\eta\|q_k\|_2^2\|y\boldsymbol{x}\|_2^2} \rceil + k + 1$, such that $\|q_l\|_2^2, v_l^T y\boldsymbol{x}$ satisfies*

$$\left(v_l^T y\boldsymbol{x}\right)^2 \left(2 + \eta \left(1 + \exp\left(\|q_l\|_2^2 v_l^T y\boldsymbol{x}\right)\right)^{-1} v_l^T y\boldsymbol{x}\right) + \|q_l\|_2^2\|y\boldsymbol{x}\|_2^2 \left(1 + \eta \left(1 + \exp\left(\|q_l\|_2^2 v_l^T y\boldsymbol{x}\right)\right)^{-1} v_l^T y\boldsymbol{x}\right)^2 \geq 0.$$

*Proof.* Consider the set $\{v^T y\mathbf{x}\}$ that satisfies

$$\left(v^T y\mathbf{x}\right)^2 \left(2 + \eta \left(1 + \exp\left(\|q\|_2^2 v^T y\mathbf{x}\right)\right)^{-1} v^T y\mathbf{x}\right) + \|q\|_2^2\|y\mathbf{x}\|_2^2 \left(1 + \eta \left(1 + \exp\left(\|q\|_2^2 v^T y\mathbf{x}\right)\right)^{-1} v^T y\mathbf{x}\right)^2 < 0$$

for all $\|q\|_2^2 > 0$. By lemmas 9 and 10, we know that the set is upper bounded by $-\frac{2}{\eta}$.

Assume for the sake of contradiction that there exists some $\|q_j\|_2^2, v_j^T y\mathbf{x}$ that satisfies

$$\left(v_j^T y\mathbf{x}\right)^2 \left(2 + \eta \left(1 + \exp\left(\|q_j\|_2^2 v_j^T y\mathbf{x}\right)\right)^{-1} v_j^T y\mathbf{x}\right) + \|q_j\|_2^2\|y\mathbf{x}\|_2^2 \left(1 + \eta \left(1 + \exp\left(\|q_j\|_2^2 v_j^T y\mathbf{x}\right)\right)^{-1} v_j^T y\mathbf{x}\right)^2 < 0.$$

where for all $j < l \leq \lceil \frac{-\left(\frac{2}{\eta} + v_j^T y\mathbf{x}\right)\left(1 + \exp\left(\|q_j\|_2^2 v_j^T y\mathbf{x}\right)\right)}{\eta\|q_j\|_2^2\|y\mathbf{x}\|_2^2} \rceil + j + 1$, $\|q_l\|_2^2, v_l^T y\mathbf{x}$ satisfies

$$\left(v_l^T y\mathbf{x}\right)^2 \left(2 + \eta \left(1 + \exp\left(\|q_l\|_2^2 v_l^T y\mathbf{x}\right)\right)^{-1} v_l^T y\mathbf{x}\right) + \|q_l\|_2^2\|y\mathbf{x}\|_2^2 \left(1 + \eta \left(1 + \exp\left(\|q_l\|_2^2 v_l^T y\mathbf{x}\right)\right)^{-1} v_l^T y\mathbf{x}\right)^2 < 0.$$

For conciseness, let's set $\xi = \lceil \frac{-\left(\frac{2}{\eta}+v_j^T y\mathbf{x}\right)\left(1+\exp\left(\|q_j\|_2^2 v_j^T y\mathbf{x}\right)\right)}{\eta\|q_j\|_2^2\|y\mathbf{x}\|_2^2} \rceil + j + 1$. We derive the following.

$$-\frac{2}{\eta} \geq v_\xi^T y\mathbf{x}$$

$$= v_j^T y\mathbf{x} + \eta\|y\mathbf{x}\|_2^2 \sum_{i=j}^{\xi-1} \left(1 + \exp\left(\|q_i\|_2^2 v_i^T y\mathbf{x}\right)\right)^{-1} \|q_i\|_2^2$$

$$> v_j^T y\mathbf{x} + \eta\|y\mathbf{x}\|_2^2\|q_j\|_2^2 \sum_{i=j}^{\xi-1} \left(1 + \exp\left(\|q_i\|_2^2 v_i^T y\mathbf{x}\right)\right)^{-1} \qquad \text{by lemmas 6 and 8}$$

$$> v_j^T y\mathbf{x} + \eta\|y\mathbf{x}\|_2^2\|q_j\|_2^2 \left(1 + \exp\left(\|q_j\|_2^2 v_j^T y\mathbf{x}\right)\right)^{-1} (\xi - j),$$

where the last step comes from lemma C.7 and our assumption for all $j < l \leq \lceil \frac{-\left(\frac{2}{\eta}+v_j^T y\mathbf{x}\right)\left(1+\exp\left(\|q_j\|_2^2 v_j^T y\mathbf{x}\right)\right)}{\eta\|q_j\|_2^2\|y\mathbf{x}\|_2^2} \rceil + j + 1$.

Then, we get

$$\frac{-\left(\frac{2}{\eta}+v_j^T y\mathbf{x}\right)\left(1+\exp\left(\|q_j\|_2^2 v_j^T y\mathbf{x}\right)\right)}{\eta\|y\mathbf{x}\|_2^2\|q_j\|_2^2} > \xi - j = \lceil \frac{-\left(\frac{2}{\eta}+v_j^T y\mathbf{x}\right)\left(1+\exp\left(\|q_j\|_2^2 v_j^T y\mathbf{x}\right)\right)}{\eta\|q_j\|_2^2\|y\mathbf{x}\|_2^2} \rceil + 1,$$

which is a contradiction.

$\square$

## C.2 TWO DATAPOINT SETTING ($N = 2$)

In the following subsection, we provide our fixed analysis, sharpness approximations and analysis, and additional experiments.

Prior to that discussion, we first prove Claim 4.3 below about the strictly increasing behavior of $m_t$. Recall in Section 4 that we defined $m_t$ as $v_t^T (\mathbf{x}_1 - \mathbf{x}_2)$, the component of $v_t$ in the max-margin direction.

*Proof of Claim 4.3.* Consider some $m_k$ and $\|q_k\|_2^2 > 0$. Based on the iterated map of $m_t$, we see the following.

$$m_l = m_{l-1} + \eta\|q_{l-1}\|_2^2 \left(1 - \mathbf{x}_1^T \mathbf{x}_2\right)\left(\left(1 + \exp\left(\frac{1}{2}\|q_{l-1}\|_2^2 (m_{l-1} + c_{l-1})\right)\right)^{-1} \right.$$
$$\left. + \left(1 + \exp\left(\frac{1}{2}\|q_{l-1}\|_2^2 (m_{l-1} - c_{l-1})\right)\right)^{-1}\right)$$

$$= m_k + \eta\|q_k\|_2^2 \left(1 - \mathbf{x}_1^T \mathbf{x}_2\right)\left(\left(1 + \exp\left(\frac{1}{2}\|q_k\|_2^2 (m_k + c_k)\right)\right)^{-1} \right.$$
$$\left. + \left(1 + \exp\left(\frac{1}{2}\|q_k\|_2^2 (m_k - c_k)\right)\right)^{-1}\right)$$

$$+ \eta \left(1 - \mathbf{x}_1^T \mathbf{x}_2\right) \sum_{i=k+1}^{l-1} \|q_i\|_2^2 \left(\left(1 + \exp\left(\frac{1}{2}\|q_i\|_2^2 (m_i + c_i)\right)\right)^{-1} \right.$$
$$\left. + \left(1 + \exp\left(\frac{1}{2}\|q_i\|_2^2 (m_i - c_i)\right)\right)^{-1}\right)$$

$$\geq m_k + \eta \|q_k\|_2^2 \left(1 - \mathbf{x}_1^T \mathbf{x}_2\right) \left( \left(1 + \exp\left(\frac{1}{2}\|q_k\|_2^2 (m_k + c_k)\right)\right)^{-1} \right.$$

$$\left. + \left(1 + \exp\left(\frac{1}{2}\|q_k\|_2^2 (m_k - c_k)\right)\right)^{-1} \right) \quad \text{since } \|q_k\|_2^2 > 0$$

$$> m_k$$

$\square$

### C.2.1 FIXED POINT ANALYSIS

Recall in Section 4, we showed that the dynamics of our model in the two datapoint setting can be captured by the following system $(\|q_t\|_2^2, c_t, m_t)$, where $c_t = v_t^T (\mathbf{x}_1 + \mathbf{x}_2)$ (projection of model parameter $v$ in max-margin complement direction) and $m_t = v_t^T (\mathbf{x}_1 - \mathbf{x}_2)$ (projection of model parameter $v$ in max-margin direction). We proceed to analyze the fixed points for the two datapoint system below. We make the following claims about the system.

**Claim C.13** (Nullclines of $\|q_t\|_2^2$). *The nullclines for $\|q_t\|_2^2$ are.*

$$\left(1 + \exp\left(\frac{1}{2}\|q_t\|_2^2 (m_t + c_t)\right)\right)^{-1} (m_t + c_t) + \left(1 + \exp\left(\frac{1}{2}\|q_t\|_2^2 (m_t - c_t)\right)\right)^{-1} (m_t - c_t) = b$$

*for $b \in \{0, -\frac{4}{\eta}\}$ and $\|q_t\|_2^2 = 0$.*

**Claim C.14** (Nullclines of $c_t$). *The nullclines for $c_t$ are $\|q_t\|_2^2 = 0$ and $c_t = 0$.*

**Claim C.15** (Fixed Points for $(\|q_t\|_2^2, c_t, m_t)$ system). *The fixed points for the system $(\|q_t\|_2^2, c_t, m_t)$ are $\|q_t\|_2^2 = 0$ and $c_t = 0, \|q_t\|_2^2 m_t \to \infty$, where $c_t = 0, \|q_t\|_2^2 m_t \to \infty$ is marginally stable and $\|q_t\|_2^2 = 0$ is stable for $-\frac{4}{\eta} < m_t < 0$.*

**Claim C.16** (Stable $c_t = 0$). *If our system satisfies the following, then the $c_t = 0$ plane is stable.*

$$\|q_t\|_2^4 \left(1 + \exp\left(\frac{1}{2}\|q_t\|_2^2 m_t\right)\right)^{-2} \exp\left(\frac{1}{2}\|q_t\|_2^2 m_t\right) < \frac{2}{\eta\left(1 + \mathbf{x}_1^T \mathbf{x}_2\right)}$$

**Claim C.17** ($|c_t|$ Growth Boundary). *If our system satisfies the following, then $c_t$ will grow in absolute value.*

$$\|q_t\|_2^2 \left(1 + \exp\left(\frac{1}{2}\|q_t\|_2^2 (m_t - |c_t|)\right)\right)^{-1} - \left(1 + \exp\left(\frac{1}{2}\|q_t\|_2^2 (m_t + |c_t|)\right)\right)^{-1} > \frac{2|c_t|}{\eta\left(1 + \mathbf{x}_1^T \mathbf{x}_2\right)}$$

We prove the claims below, starting with Claim C.13 about the nullclines of $\|q_t\|_2^2$.

*Proof of Claim C.13.* To solve for the nullclines of $\|q_t\|_2^2$, we first specify its iterated map.

$$\|q_{t+1}\|_2^2 = \|q_t\|_2^2$$

$$+ \eta\|q_t\|_2^2 \left[ \left(1 + \exp\left(\frac{1}{2}\|q_t\|_2^2 (m_t + c_t)\right)\right)^{-1} (m_t + c_t) + \left(1 + \exp\left(\frac{1}{2}\|q_t\|_2^2 (m_t - c_t)\right)\right)^{-1} (m_t - c_t) \right]$$

$$+ \frac{\eta^2}{4}\|q_t\|_2^2 \left[ \left(1 + \exp\left(\frac{1}{2}\|q_t\|_2^2 (m_t + c_t)\right)\right)^{-1} (m_t + c_t) + \left(1 + \exp\left(\frac{1}{2}\|q_t\|_2^2 (m_t - c_t)\right)\right)^{-1} (m_t - c_t) \right]^2$$

$$= \|q_t\|_2^2 \left[ 1 + \frac{\eta}{2} \left( \left(1 + \exp\left(\frac{1}{2}\|q_t\|_2^2 (m_t + c_t)\right)\right)^{-1} (m_t + c_t) + \left(1 + \exp\left(\frac{1}{2}\|q_t\|_2^2 (m_t - c_t)\right)\right)^{-1} (m_t - c_t) \right) \right]^2$$

With the iterated map above, we find the nullclines by solving $\|q_{t+1}\|_2^2 - \|q_t\|_2^2 = 0$. With some calculation we find get the following.

$$0 = \|q_t\|_2^2 \left[ \left(1 + \exp\left(\frac{1}{2}\|q_t\|_2^2 (m_t + c_t)\right)\right)^{-1} (m_t + c_t) + \left(1 + \exp\left(\frac{1}{2}\|q_t\|_2^2 (m_t - c_t)\right)\right)^{-1} (m_t - c_t) \right]$$

$$\cdot \left[ 1 + \frac{\eta}{4} \left[ \left(1 + \exp\left(\frac{1}{2}\|q_t\|_2^2 (m_t + c_t)\right)\right)^{-1} (m_t + c_t) + \left(1 + \exp\left(\frac{1}{2}\|q_t\|_2^2 (m_t - c_t)\right)\right)^{-1} (m_t - c_t) \right] \right]$$

From the above, we see that the nullclines are $\|q_t\|_2^2 = 0$ and the following:

$$\left[\left(1 + \exp\left(\frac{1}{2}\|q_t\|_2^2(m_t + c_t)\right)\right)^{-1}(m_t + c_t) + \left(1 + \exp\left(\frac{1}{2}\|q_t\|_2^2(m_t - c_t)\right)\right)^{-1}(m_t - c_t)\right] = 0$$

$$\left[\left(1 + \exp\left(\frac{1}{2}\|q_t\|_2^2(m_t + c_t)\right)\right)^{-1}(m_t + c_t) + \left(1 + \exp\left(\frac{1}{2}\|q_t\|_2^2(m_t - c_t)\right)\right)^{-1}(m_t - c_t)\right] = -\frac{4}{\eta}.$$

$\square$

Now, we prove Claim C.14 for the nullclines of $c_t$.

*Proof of Claim C.14.* To calculate the nullclines of $c_t$, we first specify its iterated map below.

$$c_{t+1} = c_t + \eta\|q_t\|_2^2(1 + \mathbf{x}_1^T\mathbf{x}_2)\left(\left(1 + \exp\left(\frac{1}{2}\|q_t\|_2^2(m_t + c_t)\right)\right)^{-1} - \left(1 + \exp\left(\|q_t\|_2^2\frac{1}{2}(m_t - c_t)\right)\right)^{-1}\right)$$

With the iterated map above, we can find the nullclines by solving $c_{t+1} - c_t = 0$, which is equivalent to solving

$$\|q_t\|_2^2\left(\left(1 + \exp\left(\frac{1}{2}\|q_t\|_2^2(m_t + c_t)\right)\right)^{-1} - \left(1 + \exp\left(\|q_t\|_2^2\frac{1}{2}(m_t - c_t)\right)\right)^{-1}\right) = 0.$$

We see that solutions to the above are $\|q_t\|_2^2 = 0$ and $c_t = 0$. $\square$

Next, we prove Claim C.15 about the fixed points.

*Proof of Claim C.15.* Since we already have the nullclines for $\|q_t\|_2^2$ and $c_t$ from Claims C.13 and C.14, we find the fixed points by first calculating the nullclines for $m_t$. We first specify the iterated map of $m_t$.

$$m_{t+1} = m_t + \eta\|q_t\|_2^2(1 - \mathbf{x}_1^T\mathbf{x}_2)\left(\left(1 + \exp\left(\frac{1}{2}\|q_t\|_2^2(m_t + c_t)\right)\right)^{-1} + \left(1 + \exp\left(\frac{1}{2}\|q_t\|_2^2(m_t - c_t)\right)\right)^{-1}\right)$$

Then, we solve for the nullclines of $m_t$ by solving

$$\|q_t\|_2^2\left(\left(1 + \exp\left(\frac{1}{2}\|q_t\|_2^2(m_t + c_t)\right)\right)^{-1} + \left(1 + \exp\left(\frac{1}{2}\|q_t\|_2^2(m_t - c_t)\right)\right)^{-1}\right) = 0.$$

From our calculation, we find that the nullclines are $\|q_t\|_2^2 = 0$ and $\|q_t\|_2^2(m_t - c_t) \to \infty$, $\|q_t\|_2^2(m_t + c_t) \to \infty$. Considering the above with the nullclines of $c_t$ and $\|q_t\|_2^2$, we find that the fixed points of the system are $\|q_t\|_2^2 = 0$ and $c_t = 0$, $\|q_t\|_2^2 m_t \to \infty$.

With the fixed points calculated, we proceed to determine their the stability. We do so by solving for the eigenvalues of the Jacobian of the system at each fixed point. First, we form the Jacobian matrix below.

$$J(\|q_{t+1}\|_2^2, c_{t+1}, m_{t+1}) = \begin{bmatrix} \frac{\partial\|q_{t+1}\|_2^2}{\partial\|q_t\|_2^2} & \frac{\partial\|q_{t+1}\|_2^2}{\partial c_t} & \frac{\partial\|q_{t+1}\|_2^2}{\partial m_t} \\ \frac{\partial c_{t+1}}{\partial\|q_t\|_2^2} & \frac{\partial c_{t+1}}{\partial c_t} & \frac{\partial c_{t+1}}{\partial m_t} \\ \frac{\partial m_{t+1}}{\partial\|q_t\|_2^2} & \frac{\partial m_{t+1}}{\partial c_t} & \frac{\partial m_{t+1}}{\partial m_t} \end{bmatrix}$$

Next, we evaluate it at the fixed point $c_t = 0$, $\|q_t\|_2^2 m_t \to \infty$ as shown below.

$$J(\|q_{t+1}\|_2^2, c_{t+1}, m_{t+1}) = \begin{bmatrix} 1 & 0 & 0 \\ 0 & 1 & 0 \\ 0 & 0 & 1 \end{bmatrix}$$

Since all the eigenvalues are 1, we know that $c_t = 0$, $\|q_t\|_2^2 m_t \to \infty$ is marginally stable.

For fixed point $\|q_t\|_2^2 = 0$, the Jacobian reduces to the following.

$$J(\|q_{t+1}\|_2^2, c_{t+1}, m_{t+1}) = \begin{bmatrix} 1 + \eta m_t + \frac{\eta^2}{4} m_t^2 & 0 & 0 \\ 0 & 1 & 0 \\ \eta\left(1 - \mathbf{x}_1^T \mathbf{x}_2\right) & 0 & 1 \end{bmatrix}$$

where the eigenvalues are 1 (with algebraic multiplicity 2) and $1 + \eta m_t + \frac{\eta^2}{4} m_t^2$.

This means that there exists some part of the $m_t$ axis that is stable if the following condition is satisfied:

$$-\frac{8}{\eta^2} < \left(\frac{4}{\eta} + m_t\right) m_t < 0$$

Since $-\frac{8}{\eta^2}$ is not a tight lower bound of the function, we get that the condition is satisfied for $-\frac{4}{\eta} < m_t < 0$.

$\square$

In the following, we prove Claim C.16 about the stable region of $c_t = 0$ plane.

*Proof of Claim C.16.* To determine the stability of the $c_t = 0$ plane, we study the eigenvalues of the Jacobian of the system at $c_t = 0$. From our calculations, we get that

$$J(\|q_{t+1}\|_2^2, c_{t+1}, m_{t+1}) = \begin{bmatrix} \left.\frac{\partial \|q_{t+1}\|_2^2}{\partial \|q_t\|_2^2}\right|_{c_t=0} & 0 & \left.\frac{\partial \|q_{t+1}\|_2^2}{\partial m_t}\right|_{c_t=0} \\ 0 & \left.\frac{\partial c_{t+1}}{\partial c_t}\right|_{c_t=0} & 0 \\ \left.\frac{\partial m_{t+1}}{\partial \|q_t\|_2^2}\right|_{c_t=0} & 0 & \left.\frac{\partial m_{t+1}}{\partial m_t}\right|_{c_t=0} \end{bmatrix}$$

From the Jacobian, the relevant eigenvalue is

$$\lambda = \left.\frac{\partial c_{t+1}}{\partial c_t}\right|_{c_t=0} = 1 - \eta\|q_t\|_2^4\left(1 + \mathbf{x}_1^T \mathbf{x}_2\right)\left(1 + \exp\left(\frac{1}{2}\|q_t\|_2^2 m_t\right)\right)^{-2}\exp\left(\frac{1}{2}\|q_t\|_2^2 m_t\right)$$

, since the eigendirection is $\nu = \begin{bmatrix} 0 \\ 1 \\ 0 \end{bmatrix}$ (i.e., in the direction of $c_t$).

To determine where the $c_t = 0$ plane is stable, we consider the following condition:

$$-1 < 1 - \eta\|q_t\|_2^4\left(1 + \mathbf{x}_1^T \mathbf{x}_2\right)\left(1 + \exp\left(\frac{1}{2}\|q_t\|_2^2 m_t\right)\right)^{-2}\exp\left(\frac{1}{2}\|q_t\|_2^2 m_t\right) < 1$$

$$\|q_t\|_2^4\left(1 + \exp\left(\frac{1}{2}\|q_t\|_2^2 m_t\right)\right)^{-2}\exp\left(\frac{1}{2}\|q_t\|_2^2 m_t\right) < \frac{2}{\eta\left(1 + \mathbf{x}_1^T \mathbf{x}_2\right)}$$

$\square$

Lastly, we prove Claim C.17 about the $|c_t|$ growth boundary.

*Proof of Claim C.17.* To understand when $c_t$ will grow in absolute value, we look at the nullclines for $c_t^2$. First, we specify the iterated map for $c_t^2$ below.

$$c_{t+1}^2 = c_t^2$$

$$- 2\eta\left(1 + \mathbf{x}_1^T \mathbf{x}_2\right)\|q_t\|_2^2\sqrt{c_t^2}\left[\left(1 + \exp\left(\frac{1}{2}\|q_t\|_2^2\left(m_t - \sqrt{c_t^2}\right)\right)\right)^{-1} - \left(1 + \exp\left(\frac{1}{2}\|q_t\|_2^2\left(m_t + \sqrt{c_t^2}\right)\right)\right)^{-1}\right]$$

$$+ \eta^2\left(1 + \mathbf{x}_1^T \mathbf{x}_2\right)^2\|q_t\|_2^4\left[\left(1 + \exp\left(\frac{1}{2}\|q_t\|_2^2\left(m_t - \sqrt{c_t^2}\right)\right)\right)^{-1} - \left(1 + \exp\left(\frac{1}{2}\|q_t\|_2^2\left(m_t + \sqrt{c_t}\right)\right)\right)^{-1}\right]^2$$

Now, to solve for the nullclines (i.e., $c_{t+1}^2 - c_t^2 = 0$), we solve the following.

$$0 = \|q_t\|_2^2 \left[ \left( 1 + \exp\left( \frac{1}{2} \|q_t\|_2^2 \left( m_t - \sqrt{c_t^2} \right) \right) \right)^{-1} - \left( 1 + \exp\left( \frac{1}{2} \|q_t\|_2^2 \left( m_t + \sqrt{c_t^2} \right) \right) \right)^{-1} \right]$$

$$\cdot \left[ -2\sqrt{c_t^2} + \eta \left( 1 + \mathbf{x}_1^T \mathbf{x}_2 \right) \|q_t\|_2^2 \left[ \left( 1 + \exp\left( \frac{1}{2} \|q_t\|_2^2 \left( m_t - \sqrt{c_t^2} \right) \right) \right)^{-1} - \left( 1 + \exp\left( \frac{1}{2} \|q_t\|_2^2 \left( m_t + \sqrt{c_t^2} \right) \right) \right)^{-1} \right] \right]$$

As we are already familiar with the nullclines $\|q_t\|_2^2 = 0$ and $c_t = 0$ from Claim C.14, we see that $c_t^2$ grows if our system satisfies the following:

$$\|q_t\|_2^2 \left( 1 + \exp\left( \frac{1}{2} \|q_t\|_2^2 \left( m_t - \sqrt{c_t^2} \right) \right) \right)^{-1} - \left( 1 + \exp\left( \frac{1}{2} \|q_t\|_2^2 \left( m_t + \sqrt{c_t^2} \right) \right) \right)^{-1} > \frac{2\sqrt{c_t^2}}{\eta \left( 1 + \mathbf{x}_1^T \mathbf{x}_2 \right)}$$

This is equivalent to saying the system should satisfy

$$\|q_t\|_2^2 \left( 1 + \exp\left( \frac{1}{2} \|q_t\|_2^2 \left( m_t - |c_t| \right) \right) \right)^{-1} - \left( 1 + \exp\left( \frac{1}{2} \|q_t\|_2^2 \left( m_t + |c_t^|| \right) \right) \right)^{-1} > \frac{2|c_t|}{\eta \left( 1 + \mathbf{x}_1^T \mathbf{x}_2 \right)}$$

$\square$

### C.2.2 SHARPNESS APPROXIMATIONS

As mentioned in Section 4, we derived some approximations for the sharpness in our two datapoint setting. In this section, we derive the four sharpness approximations and their error bounds, which we show below.

First, we define the following notation.

$$m_t = v_t^T \left( \mathbf{x}_1 - \mathbf{x}_2 \right) \qquad \text{(projection onto max-margin)}$$

$$c_t = v_t^T \left( \mathbf{x}_1 + \mathbf{x}_2 \right) \qquad \text{(projection onto max-margin complement)}$$

$$\psi_{1,t} = \left( 1 + \exp\left( \frac{1}{2} \|q_t\|_2^2 \left( m_t + c_t \right) \right) \right)^{-2} \exp\left( \frac{1}{2} \|q_t\|_2^2 \left( m_t + c_t \right) \right)$$

$$\psi_{2,t} = \left( 1 + \exp\left( \frac{1}{2} \|q_t\|_2^2 \left( m_t - c_t \right) \right) \right)^{-2} \exp\left( \frac{1}{2} \|q_t\|_2^2 \left( m_t - c_t \right) \right)$$

$$\phi_{1,t} = \left( 1 + \exp\left( \frac{1}{2} \|q_t\|_2^2 \left( m_t + c_t \right) \right) \right)^{-1}$$

$$\phi_{2,t} = \left( 1 + \exp\left( \frac{1}{2} \|q_t\|_2^2 \left( m_t - c_t \right) \right) \right)^{-1}$$

Now, we introduce our approximations.

**Approximation 1** Our first sharpness approximation is

$$\frac{1}{2} \|q_t\|_2^4 \left( \left( \psi_{1,t} + \psi_{2,t} \right) \pm \sqrt{\left( \psi_{1,t} - \psi_{2,t} \right)^2 + 4\psi_{1,t}\psi_{2,t} \left( \mathbf{x}_1^T \mathbf{x}_2 \right)^2} \right)$$

where the error with the true sharpness is upper bounded by

$$\frac{1}{4} \left( \psi_{1,t} \|q_t\|_2^2 \left( m_t + c_t \right)^2 - \phi_{1,t} \left( m_t + c_t \right) + \psi_{2,t} \|q_t\|_2^2 \left( m_t - c_t \right)^2 - \phi_{2,t} \left( m_t - c_t \right) \right)$$

$$+ \sqrt{ \begin{array}{l} \frac{1}{16} \left( \psi_{1,t} \|q_t\|_2^2 \left( m_t + c_t \right)^2 - \phi_{1,t} \left( m_t + c_t \right) + \psi_{2,t} \|q_t\|_2^2 \left( m_t - c_t \right)^2 - \phi_{2,t} \left( m_t - c_t \right) \right)^2 \\[2mm] + 2\|q_t\|_2^2 \left( \left( \frac{1}{2} \psi_{1,t} \|q_t\|_2^2 \left( m_t + c_t \right) - \phi_{1,t} \right)^2 + \left( -\frac{1}{2} \psi_{2,t} \|q_t\|_2^2 \left( m_t - c_t \right) + \phi_{2,t} \right)^2 \right) \\[2mm] + 4\|q_t\|_2^2 \mathbf{x}_1^T \mathbf{x}_2 \left( \frac{1}{2} \psi_{1,t} \|q_t\|_2^2 \left( m_t + c_t \right) - \phi_{1,t} \right) \left( -\frac{1}{2} \psi_{2,t} \|q_t\|_2^2 \left( m_t - c_t \right) + \phi_{2,t} \right) \end{array} }$$

**Approximation 2** Our second sharpness approximation is

$$\frac{1}{4}\left(\psi_{1,t}\|q_t\|_2^2\left(m_t+c_t\right)^2+\psi_{2,t}\|q_t\|_2^2\left(m_t-c_t\right)^2-\phi_{1,t}\left(m_t+c_t\right)-\phi_{2,t}\left(m_t-c_t\right)+(\psi_{1,t}+\psi_{2,t})\|q_t\|_2^4\left(1-\mathbf{x}_1^T\mathbf{x}_2\right)\right)$$

$$+\frac{1}{2}\sqrt{\begin{array}{l}\frac{1}{4}\left(\psi_{1,t}\|q_t\|_2^2\left(m_t+c_t\right)^2+\psi_{2,t}\|q_t\|_2^2\left(m_t-c_t\right)^2-\phi_{1,t}\left(m_t+c_t\right)-\phi_{2,t}\left(m_t-c_t\right)\right.\\\left.+(\psi_{1,t}+\psi_{2,t})\|q_t\|_2^4\left(1-\mathbf{x}_1^T\mathbf{x}_2\right)\right)^2-\left(4\psi_{1,t}\psi_{2,t}\|q_t\|_2^6c_t^2+2\left(\phi_{2,t}\psi_{1,t}-\phi_{1,t}\psi_{2,t}\right)\|q_t\|_2^4c_t\right.\\\left.+3\|q_t\|_2^4\left(\psi_{1,t}\left(m_t+c_t\right)+\psi_{2,t}\left(m_t-c_t\right)\right)\left(\phi_{1,t}+\phi_{2,t}\right)-4\left(\phi_{1,t}+\phi_{2,t}\right)^2\|q_t\|_2^2\right)\left(1-\mathbf{x}_1^T\mathbf{x}_2\right)\end{array}}$$

where the error with the true sharpness is upper bounded by

$$\frac{1}{4}(\psi_{1,t}+\psi_{2,t})\|q_t\|_2^4\left(1+\mathbf{x}_1^T\mathbf{x}_2\right)$$

$$+\frac{1}{2}\|q_t\|_2\sqrt{1+\mathbf{x}_1^T\mathbf{x}_2}\sqrt{\begin{array}{l}\frac{1}{4}\left(\psi_{1,t}+\psi_{2,t}\right)^2\|q_t\|_2^6\left(1+\mathbf{x}_1^T\mathbf{x}_2\right)+\left(\psi_{1,t}-\psi_{2,t}\right)^2\|q_t\|_2^6\left(1-\mathbf{x}_1^T\mathbf{x}_2\right)\\+4\left(\frac{1}{2}\psi_{1,t}\|q_t\|_2^2\left(m_t+c_t\right)-\frac{1}{2}\psi_{2,t}\|q_t\|_2^2\left(m_t-c_t\right)-\phi_{1,t}+\phi_{2,t}\right)^2\end{array}}\,\Big)$$

**Approximation 3** (Low-Error Approximation) Our third sharpness approximation is

$$\frac{1}{4}\left(\psi_{1,t}\|q_t\|_2^2\left(m_t+c_t\right)^2+\psi_{2,t}\|q_t\|_2^2\left(m_t-c_t\right)^2-\phi_{1,t}\left(m_t+c_t\right)-\phi_{2,t}\left(m_t-c_t\right)+(\psi_{1,t}+\psi_{2,t})\|q_t\|_2^4\left(1+\mathbf{x}_1^T\mathbf{x}_2\right)\right)$$

$$+\frac{1}{2}\sqrt{\begin{array}{l}\frac{1}{4}\left(\psi_{1,t}\|q_t\|_2^2\left(m_t+c_t\right)^2+\psi_{2,t}\|q_t\|_2^2\left(m_t-c_t\right)^2-\phi_{1,t}\left(m_t+c_t\right)-\phi_{2,t}\left(m_t-c_t\right)\right.\\\left.+(\psi_{1,t}+\psi_{2,t})\|q_t\|_2^4\left(1+\mathbf{x}_1^T\mathbf{x}_2\right)\right)^2-\left(4\psi_{1,t}\psi_{2,t}\|q_t\|_2^6m_t^2-2\|q_t\|_2^4\left(\phi_{1,t}\psi_{2,t}+\phi_{2,t}\psi_{1,t}\right)m_t\right.\\\left.+3\|q_t\|_2^4\left(\psi_{1,t}\left(m_t+c_t\right)-\psi_{2,t}\left(m_t-c_t\right)\right)\left(\phi_{1,t}-\phi_{2,t}\right)-4\left(\phi_{1,t}-\phi_{2,t}\right)^2\|q_t\|_2^2\right)\left(1+\mathbf{x}_1^T\mathbf{x}_2\right)\end{array}}$$

where the error with the true sharpness is upper bounded by

$$\frac{1}{4}\left(\psi_{1,t}+\psi_{2,t}\right)\|q_t\|_2^4\left(1-\mathbf{x}_1^T\mathbf{x}_2\right)$$

$$+\frac{1}{2}\|q_t\|_2\sqrt{1-\mathbf{x}_1^T\mathbf{x}_2}\sqrt{\begin{array}{l}\frac{1}{4}\left(\psi_{1,t}+\psi_{2,t}\right)^2\|q_t\|_2^6\left(1-\mathbf{x}_1^T\mathbf{x}_2\right)+\left(\psi_{1,t}-\psi_{2,t}\right)^2\|q_t\|_2^6\left(1+\mathbf{x}_1^T\mathbf{x}_2\right)\\+4\left(\frac{1}{2}\psi_{1,t}\|q_t\|_2^2\left(m_t+c_t\right)-\phi_{1,t}+\frac{1}{2}\psi_{2,t}\|q_t\|_2^2\left(m_t-c_t\right)-\phi_{2,t}\right)^2\end{array}}$$

**Approximation 4** (Interpretable Approximation) Our fourth sharpness approximation is

$$\frac{1}{2}(\psi_{1,t}+\psi_{2,t})\|q_t\|_2^4(1+\mathbf{x}_1^T\mathbf{x}_2)$$

where the error with the true sharpness is upper bounded by

$$\sqrt{\begin{array}{l}\frac{1}{4}\left(\psi_{1,t}\|q_t\|_2^2\left(m_t+c_t\right))^2+\psi_{2,t}\|q_t\|_2^2\left(m_t-c_t\right)^2-\phi_{1,t}\left(m_t+c_t\right)-\phi_{2,t}\left(m_t-c_t\right)\right)^2\\+2\|q_t\|_2^2\left(1+\mathbf{x}_1^T\mathbf{x}_2\right)\left(\frac{1}{2}\psi_{1,t}\|q_t\|_2^2\left(m_t+c_t\right)-\frac{1}{2}\psi_{2,t}\|q_t\|_2^2\left(m_t-c_t\right)-\phi_{1,t}+\phi_{2,t}\right)^2\\+2\|q_t\|_2^2\left(1-\mathbf{x}_1^T\mathbf{x}_2\right)\left(\frac{1}{2}\psi_{1,t}\|q_t\|_2^2\left(m_t+c_t\right)+\frac{1}{2}\psi_{2,t}\|q_t\|_2^2\left(m_t-c_t\right)-\phi_{1,t}-\phi_{2,t}\right)^2\\+\frac{1}{2}\left(\psi_{1,t}-\psi_{2,t}\right)^2\|q_t\|_2^8\left(1-(\mathbf{x}_1^T\mathbf{x}_2)^2\right)+\frac{1}{4}\left(\psi_{1,t}+\psi_{2,t}\right)^2\|q_t\|_2^8\left(1-\mathbf{x}_1^T\mathbf{x}_2\right)^2\end{array}}$$

From our approximations, we claim that approximation 3 is a low-error approximation of the sharpness in our setting. We also denote approximation 4 as our interpretable sharpness approximation based on its simplicity in form.

In the remainder of the section, we perform some preliminary calculations necessary for the approximations, show our derivations of the four sharpness approximations and their error bounds, and provide some empirical results and error bound comparisons to support for our claim about approximation 3.

**Preliminary Calculations**   Before getting into the details of the approximations, we perform some preliminarily calculations on the Loss Hessian. With $\theta = (q_t, u_t, v_t)$, we start with

$$
\nabla_{\theta_t} L(q_t, u_t, v_t) - \lambda I_{2h+d}
$$
$$
= \begin{bmatrix} A & B & C \\ B & A & C \\ C^T & C^T & D \end{bmatrix}
$$

where

$$
A = \left[ \psi_{1,t} \left( v_t^T \mathbf{x}_1 \right)^2 + \psi_{2,t} \left( v_t^T \mathbf{x}_2 \right)^2 \right] q_t q_t^T - \lambda I_h
$$
$$
B = - \left[ \phi_{1,t} v_t^T \mathbf{x}_1 - \phi_{2,t} v_t^T \mathbf{x}_2 \right] I_h + \left[ \psi_{1,t} \left( v_t^T \mathbf{x}_1 \right)^2 + \psi_{2,t} \left( v_t^T \mathbf{x}_2 \right)^2 \right] q_t q_t^T
$$
$$
C = -q_t \left[ \phi_{1,t} \mathbf{x}_1^T - \phi_{2,t} \mathbf{x}_2^T \right] + \|q_t\|_2^2 q_t \left[ \psi_{1,t} v_t^T \mathbf{x}_1 \mathbf{x}_1^T + \psi_{2,t} v_t^T \mathbf{x}_2 \mathbf{x}_2^T \right]
$$
$$
D = \|q_t\|_2^4 \left[ \psi_{1,t} \mathbf{x}_1 \mathbf{x}_1^T + \psi_{2,t} \mathbf{x}_2 \mathbf{x}_2^T \right] - \lambda I_d.
$$

After performing the same similarity transforms as we discussed in the single datapoint setting (see Appendix C.1.2), we look at the following matrix decomposition:

$$
\begin{bmatrix} 0 & \sqrt{2}C \\ \sqrt{2}C^T & 0 \end{bmatrix} + \begin{bmatrix} A+B & 0 \\ 0 & D \end{bmatrix} \text{ in which we overload notation a little here by denoting}
$$

$$
A + B = 2 \left[ \psi_{1,t} \left( v_t^T \mathbf{x}_1 \right)^2 + \psi_{2,t} \left( v_t^T \mathbf{x}_2 \right)^2 \right] q_t q_t^T - \left[ \phi_{1,t} v_t^T \mathbf{x}_1 - \phi_{2,t} v_t^T \mathbf{x}_2 \right] I_h
$$
$$
D = \|q_t\|_2^4 \left[ \psi_{1,t} \mathbf{x}_1 \mathbf{x}_1^T + \psi_{2,t} \mathbf{x}_2 \mathbf{x}_2^T \right].
$$

In the above decomposition, we treat $\begin{bmatrix} 0 & \sqrt{2}C \\ \sqrt{2}C^T & 0 \end{bmatrix}$ as a perturbation on the matrix $\begin{bmatrix} A+B & 0 \\ 0 & D \end{bmatrix}$.

Based on our original definition for $C$, we know that $\begin{bmatrix} 0 & \sqrt{2}C \\ \sqrt{2}C^T & 0 \end{bmatrix}$ is rank-2 with eigenvalues

$$
\pm\sqrt{2}\|q\|_2 \sqrt{ \begin{array}{l} \left( \psi_{1,t}\|q\|_2^2 v_t^T \mathbf{x}_1 - \phi_{1,t} \right)^2 + \left( \psi_{2,t}\|q\|_2^2 v_t^T \mathbf{x}_2 + \phi_{2,t} \right)^2 \\ + 2 \left( \psi_{1,t}\|q\|_2^2 v_t^T \mathbf{x}_1 - \phi_{1,t} \right) \left( \psi_{2,t}\|q\|_2^2 v_t^T \mathbf{x}_2 + \phi_{2,t} \right) \mathbf{x}_1^T \mathbf{x}_2 \end{array} }.
$$

Since the perturbation occurs in a 3-dimensional subspsace spanned by $\begin{bmatrix} q \\ 0 \end{bmatrix}$, $\begin{bmatrix} 0 \\ \mathbf{x}_1 \end{bmatrix}$, and $\begin{bmatrix} 0 \\ \mathbf{x}_2 \end{bmatrix}$, we believe the 2 nonzero eigenvalues from the D block matrix and 1 nonzero eigenvalue from the $A+B$ block matrix have been perturbed.

We now focus on studying this perturbation in the 3-dimensional subspace through the following derivation.

$$U = \begin{bmatrix} q & 0 & 0 \\ 0 & \mathbf{x}_1 & \mathbf{x}_2 \end{bmatrix}$$

$$U^T \begin{bmatrix} A+B & \sqrt{2}C \\ \sqrt{2}C^T & D \end{bmatrix} U = \begin{bmatrix} q_t^T(A+B)q_t & \sqrt{2}q_t^T C\mathbf{x}_1 & \sqrt{2}q_t^T C\mathbf{x}_2 \\ \sqrt{2}q_t^T C\mathbf{x}_1 & \mathbf{x}_1^T D\mathbf{x}_1 & \mathbf{x}_1^T D\mathbf{x}_2 \\ \sqrt{2}q_t^T C\mathbf{x}_2 & \mathbf{x}_1^T Dfirst\mathbf{x}_2 & \mathbf{x}_2^T D\mathbf{x}_2 \end{bmatrix} = K$$

$$U^T U = \begin{bmatrix} \|q_t\|_2^2 & 0 & 0 \\ 0 & 1 & \mathbf{x}_1^T\mathbf{x}_2 \\ 0 & \mathbf{x}_1^T\mathbf{x}_2 & 1 \end{bmatrix} = \begin{bmatrix} 1 & 0 & 0 \\ 0 & \frac{1}{\sqrt{2}} & \frac{1}{\sqrt{2}} \\ 0 & \frac{1}{\sqrt{2}} & -\frac{1}{\sqrt{2}} \end{bmatrix} \begin{bmatrix} \|q_t\|_2^2 & 0 & 0 \\ 0 & 1+\mathbf{x}_1^T\mathbf{x}_2 & 0 \\ 0 & 0 & 1-\mathbf{x}_1^T\mathbf{x}_2 \end{bmatrix} \begin{bmatrix} 1 & 0 & 0 \\ 0 & \frac{1}{\sqrt{2}} & \frac{1}{\sqrt{2}} \\ 0 & \frac{1}{\sqrt{2}} & -\frac{1}{\sqrt{2}} \end{bmatrix}$$

$$= Q\Lambda Q = G$$

$$(U^T U)^{-1} = Q\Lambda^{-1}Q = \begin{bmatrix} 1 & 0 & 0 \\ 0 & \frac{1}{\sqrt{2}} & \frac{1}{\sqrt{2}} \\ 0 & \frac{1}{\sqrt{2}} & -\frac{1}{\sqrt{2}} \end{bmatrix} \begin{bmatrix} \frac{1}{\|q_t\|_2^2} & 0 & 0 \\ 0 & \frac{1}{1+\mathbf{x}_1^T\mathbf{x}_2} & 0 \\ 0 & 0 & \frac{1}{1-\mathbf{x}_1^T\mathbf{x}_2} \end{bmatrix} \begin{bmatrix} 1 & 0 & 0 \\ 0 & \frac{1}{\sqrt{2}} & \frac{1}{\sqrt{2}} \\ 0 & \frac{1}{\sqrt{2}} & -\frac{1}{\sqrt{2}} \end{bmatrix} \quad \text{(Assuming } \|q_t\|_2^2 \neq 0)$$

This means the sharpness is one of the solutions to the general eigenvalue problem $K\nu = \lambda G\nu$. We can also rewrite this to get that the solutions to the general eigenvalue problem are also the eigenvalues that we get from the matrices $G^{-1}K$, $\Lambda^{-\frac{1}{2}}QKQ\Lambda^{-\frac{1}{2}}$, and $\Lambda^{-1}QKQ$. Since $\Lambda^{-\frac{1}{2}}QKQ\Lambda^{-\frac{1}{2}}$ is real and symmetric, we know that its eigenvalues are real. We focus on the deriving the matrix $\Lambda^{-\frac{1}{2}}QKQ\Lambda^{-\frac{1}{2}}$.

$$\Lambda^{-\frac{1}{2}}QKQ\Lambda^{-\frac{1}{2}}$$

$$= \begin{bmatrix} \frac{1}{\|q_t\|_2} & 0 & 0 \\ 0 & \frac{1}{\sqrt{1+\mathbf{x}_1^T\mathbf{x}_2}} & 0 \\ 0 & 0 & \frac{1}{\sqrt{1-\mathbf{x}_1^T\mathbf{x}_2}} \end{bmatrix}$$

$$\cdot \begin{bmatrix} q_t^T(A+B)q_t & q_t^T C(\mathbf{x}_1+\mathbf{x}_2) & q_t^T C(\mathbf{x}_1-\mathbf{x}_2) \\ q_t^T C(\mathbf{x}_1+\mathbf{x}_2) & \frac{1}{2}(\mathbf{x}_1+\mathbf{x}_2)^T D(\mathbf{x}_1+\mathbf{x}_2) & \frac{1}{2}(\mathbf{x}_1+\mathbf{x}_2)^T D(\mathbf{x}_1-\mathbf{x}_2) \\ q_t^T C(\mathbf{x}_1-\mathbf{x}_2) & \frac{1}{2}(\mathbf{x}_1+\mathbf{x}_2)^T D(\mathbf{x}_1-\mathbf{x}_2) & \frac{1}{2}(\mathbf{x}_1-\mathbf{x}_2)^T D(\mathbf{x}_1-\mathbf{x}_2) \end{bmatrix}$$

$$\cdot \begin{bmatrix} \frac{1}{\|q_t\|_2} & 0 & 0 \\ 0 & \frac{1}{\sqrt{1+\mathbf{x}_1^T\mathbf{x}_2}} & 0 \\ 0 & 0 & \frac{1}{\sqrt{1-\mathbf{x}_1^T\mathbf{x}_2}} \end{bmatrix}$$

$$= \begin{bmatrix} \frac{1}{\|q_t\|_2^2}q_t^T(A+B)q_t & \frac{1}{\|q_t\|_2\sqrt{1+\mathbf{x}_1^T\mathbf{x}_2}}q_t^T C(\mathbf{x}_1+\mathbf{x}_2) & \frac{1}{\|q_t\|_2\sqrt{1-\mathbf{x}_1^T\mathbf{x}_2}}q_t^T C(\mathbf{x}_1-\mathbf{x}_2) \\ \frac{1}{\|q_t\|_2\sqrt{1+\mathbf{x}_1^T\mathbf{x}_2}}q_t^T C(\mathbf{x}_1+\mathbf{x}_2) & \frac{1}{2(1+\mathbf{x}_1^T\mathbf{x}_2)}(\mathbf{x}_1+\mathbf{x}_2)^T D(\mathbf{x}_1+\mathbf{x}_2) & \frac{1}{2\sqrt{1-(\mathbf{x}_1^T\mathbf{x}_2)^2}}(\mathbf{x}_1+\mathbf{x}_2)^T D(\mathbf{x}_1-\mathbf{x}_2) \\ \frac{1}{\|q_t\|_2\sqrt{1-\mathbf{x}_1^T\mathbf{x}_2}}q_t^T C(\mathbf{x}_1-\mathbf{x}_2) & \frac{1}{2\sqrt{1-(\mathbf{x}_1^T\mathbf{x}_2)^2}}(\mathbf{x}_1+\mathbf{x}_2)^T D(\mathbf{x}_1-\mathbf{x}_2) & \frac{1}{2(1-\mathbf{x}_1^T\mathbf{x}_2)}(\mathbf{x}_1-\mathbf{x}_2)^T D(\mathbf{x}_1-\mathbf{x}_2) \end{bmatrix}$$

$$
=
\begin{bmatrix}
\begin{aligned}
& 2\psi_{1,t}\|q_t\|_2^2(v_t^T\mathbf{x}_1)^2 \\
& + 2\psi_{2,t}\|q_t\|_2^2(v_t^T\mathbf{x}_2)^2 \\
& - \phi_{1,t}v_t^T\mathbf{x}_1 \\
& + \phi_{2,t}v_t^T\mathbf{x}_2
\end{aligned}
&
\begin{aligned}
& \|q_t\|_2\sqrt{1+\mathbf{x}_1^T\mathbf{x}_2} \\
& \cdot \left( \psi_{1,t}\|q_t\|_2^2 v_t^T\mathbf{x}_1 \right. \\
& \quad + \psi_{2,t}\|q_t\|_2^2 v_t^T\mathbf{x}_2 \\
& \quad \left. - \phi_{1,t} + \phi_{2,t} \right)
\end{aligned}
&
\begin{aligned}
& \|q_t\|_2\sqrt{1-\mathbf{x}_1^T\mathbf{x}_2} \\
& \cdot \left( \psi_{1,t}\|q_t\|_2^2 v_t^T\mathbf{x}_1 \right. \\
& \quad - \psi_{2,t}\|q_t\|_2^2 v_t^T\mathbf{x}_2 \\
& \quad \left. - \phi_{1,t} - \phi_{2,t} \right)
\end{aligned}
\\[2em]
\begin{aligned}
& \|q_t\|_2\sqrt{1+\mathbf{x}_1^T\mathbf{x}_2} \\
& \cdot \left( \psi_{1,t}\|q_t\|_2^2 v_t^T\mathbf{x}_1 \right. \\
& \quad + \psi_{2,t}\|q_t\|_2^2 v_t^T\mathbf{x}_2 \\
& \quad \left. - \phi_{1,t} + \phi_{2,t} \right)
\end{aligned}
&
\frac{1}{2}(\psi_{1,t}+\psi_{2,t})\|q_t\|_2^4(1+\mathbf{x}_1^T\mathbf{x}_2)
&
\frac{1}{2}(\psi_{1,t}-\psi_{2,t})\|q_t\|_2^4\sqrt{1-(\mathbf{x}_1^T\mathbf{x}_2)^2}
\\[2em]
\begin{aligned}
& \|q_t\|_2\sqrt{1-\mathbf{x}_1^T\mathbf{x}_2} \\
& \cdot \left( \psi_{1,t}\|q_t\|_2^2 v_t^T\mathbf{x}_1 \right. \\
& \quad - \psi_{2,t}\|q_t\|_2^2 v_t^T\mathbf{x}_2 \\
& \quad \left. - \phi_{1,t} - \phi_{2,t} \right)
\end{aligned}
&
\frac{1}{2}(\psi_{1,t}-\psi_{2,t})\|q_t\|_2^4\sqrt{1-(\mathbf{x}_1^T\mathbf{x}_2)^2}
&
\frac{1}{2}(\psi_{1,t}+\psi_{2,t})\|q_t\|_2^4(1-\mathbf{x}_1^T\mathbf{x}_2)
\end{bmatrix}
$$

With the above $3x3$ matrix, we use Cauchy Interlacing Theorem (Stewart & Sun, 1990) to produce the approximations of the sharpness. For each approximation, we also calculate the error bound using Weyl's inequality.

**Approximation 1 Derivation** We consider the first submatrix formed by removing the first column and the first row.

$$
\begin{bmatrix}
\frac{1}{2}(\psi_{1,t}+\psi_{2,t})\|q_t\|_2^4(1+\mathbf{x}_1^T\mathbf{x}_2) & \frac{1}{2}(\psi_{1,t}-\psi_{2,t})\|q_t\|_2^4\sqrt{1-(\mathbf{x}_1^T\mathbf{x}_2)^2} \\
\frac{1}{2}(\psi_{1,t}-\psi_{2,t})\|q_t\|_2^4\sqrt{1-(\mathbf{x}_1^T\mathbf{x}_2)^2} & \frac{1}{2}(\psi_{1,t}+\psi_{2,t})\|q_t\|_2^4(1-\mathbf{x}_1^T\mathbf{x}_2)
\end{bmatrix}
$$

We show the eigenvalues for this submatrix below.

$$
\lambda = \frac{1}{2}\|q_t\|_2^4 \left( (\psi_{1,t}+\psi_{2,t}) \pm \sqrt{(\psi_{1,t}-\psi_{2,t})^2 + 4\psi_{1,t}\psi_{2,t}\left(\mathbf{x}_1^T\mathbf{x}_2\right)^2} \right)
$$

Now, we use Weyl's inequality to find a bound on this. We write the general $3x3$ matrix using the following decomposition:

$$
\begin{bmatrix}
0 & 0 & 0 \\
0 & \frac{1}{2}(\psi_{1,t}+\psi_{2,t})\|q_t\|_2^4(1+\mathbf{x}_1^T\mathbf{x}_2) & \frac{1}{2}(\psi_{1,t}-\psi_{2,t})\|q_t\|_2^4\sqrt{1-(\mathbf{x}_1^T\mathbf{x}_2)^2} \\
0 & \frac{1}{2}(\psi_{1,t}-\psi_{2,t})\|q_t\|_2^4\sqrt{1-(\mathbf{x}_1^T\mathbf{x}_2)^2} & \frac{1}{2}(\psi_{1,t}+\psi_{2,t})\|q_t\|_2^4(1-\mathbf{x}_1^T\mathbf{x}_2)
\end{bmatrix}
$$

$$+ \begin{bmatrix} \begin{matrix} 2\psi_{1,t}\|q_t\|_2^2(v_t^T\mathbf{x}_1)^2 - \phi_{1,t}v_t^T\mathbf{x}_1 \\ + 2\psi_{2,t}\|q_t\|_2^2(v_t^T\mathbf{x}_2)^2 + \phi_{2,t}v_t^T\mathbf{x}_2 \end{matrix} & \begin{matrix} \|q_t\|_2\sqrt{1+\mathbf{x}_1^T\mathbf{x}_2} \\ \cdot\left(\psi_{1,t}\|q_t\|_2^2v_t^T\mathbf{x}_1 - \phi_{1,t}\right. \\ \left. + \psi_{2,t}\|q_t\|_2^2v_t^T\mathbf{x}_2 + \phi_{2,t}\right) \end{matrix} & \begin{matrix} \|q_t\|_2\sqrt{1-\mathbf{x}_1^T\mathbf{x}_2} \\ \cdot\left(\psi_{1,t}\|q_t\|_2^2v_t^T\mathbf{x}_1 - \phi_{1,t}\right. \\ \left. - \psi_{2,t}\|q_t\|_2^2v_t^T\mathbf{x}_2 - \phi_{2,t}\right) \end{matrix} \\ \\ \begin{matrix} \|q_t\|_2\sqrt{1+\mathbf{x}_1^T\mathbf{x}_2} \\ \cdot\left(\psi_{1,t}\|q_t\|_2^2v_t^T\mathbf{x}_1 - \phi_{1,t}\right. \\ \left. + \psi_{2,t}\|q_t\|_2^2v_t^T\mathbf{x}_2 + \phi_{2,t}\right) \end{matrix} & 0 & 0 \\ \\ \begin{matrix} \|q_t\|_2\sqrt{1-\mathbf{x}_1^T\mathbf{x}_2} \\ \cdot\left(\psi_{1,t}\|q_t\|_2^2v_t^T\mathbf{x}_1 - \phi_{1,t}\right. \\ \left. - \psi_{2,t}\|q_t\|_2^2v_t^T\mathbf{x}_2 - \phi_{2,t}\right) \end{matrix} & 0 & 0 \end{bmatrix}$$

From our calculation, we see that one of the eigenvalues of the second matrix in the decomposition, denoted as our perturbation matrix, is 0. The other two eigenvalues are a symmetric pair that we show below, where the difference between the sharpness and our approximation in this case is upper bounded by the plus direction of the symmetric pair.

$$\lambda = \frac{1}{2}\left(2\psi_{1,t}\|q_t\|_2^2(v_t^T\mathbf{x}_1)^2 - \phi_{1,t}v_t^T\mathbf{x}_1 + 2\psi_{2,t}\|q_t\|_2^2(v_t^T\mathbf{x}_2)^2 + \phi_{2,t}v_t^T\mathbf{x}_2\right)$$

$$\pm \sqrt{\begin{matrix} \frac{1}{4}\left(2\psi_{1,t}\|q_t\|_2^2(v_t^T\mathbf{x}_1)^2 - \phi_{1,t}v_t^T\mathbf{x}_1 + 2\psi_{2,t}\|q_t\|_2^2(v_t^T\mathbf{x}_2)^2 + \phi_{2,t}v_t^T\mathbf{x}_2\right)^2 \\ + 2\|q_t\|_2^2\left(\left(\psi_{1,t}\|q_t\|_2^2v_t^T\mathbf{x}_1 - \phi_{1,t}\right)^2 + \left(\psi_{2,t}\|q_t\|_2^2v_t^T\mathbf{x}_2 + \phi_{2,t}\right)^2\right) \\ + 4\|q_t\|_2^2\mathbf{x}_1^T\mathbf{x}_2\left(\psi_{1,t}\|q_t\|_2^2v_t^T\mathbf{x}_1 - \phi_{1,t}\right)\left(\psi_{2,t}\|q_t\|_2^2v_t^T\mathbf{x}_2 + \phi_{2,t}\right) \end{matrix}}$$

**Approximation 2 Derivation**   Next, we consider the second submatrix formed by removing the second column and second row.

$$\begin{bmatrix} \begin{matrix} 2\psi_{1,t}\|q_t\|_2^2(v_t^T\mathbf{x}_1)^2 \\ + 2\psi_{2,t}\|q_t\|_2^2(v_t^T\mathbf{x}_2)^2 \\ - \phi_{1,t}v_t^T\mathbf{x}_1 \\ + \phi_{2,t}v_t^T\mathbf{x}_2 \end{matrix} & \begin{matrix} \|q_t\|_2\sqrt{1-\mathbf{x}_1^T\mathbf{x}_2} \\ \cdot\left(\psi_{1,t}\|q_t\|_2^2v_t^T\mathbf{x}_1\right. \\ - \psi_{2,t}\|q_t\|_2^2v_t^T\mathbf{x}_2 \\ \left. - \phi_{1,t} - \phi_{2,t}\right) \end{matrix} \\ \\ \begin{matrix} \|q_t\|_2\sqrt{1-\mathbf{x}_1^T\mathbf{x}_2} \\ \cdot\left(\psi_{1,t}\|q_t\|_2^2v_t^T\mathbf{x}_1\right. \\ - \psi_{2,t}\|q_t\|_2^2v_t^T\mathbf{x}_2 \\ \left. - \phi_{1,t} - \phi_{2,t}\right) \end{matrix} & \frac{1}{2}(\psi_{1,t}+\psi_{2,t})\|q_t\|_2^4(1-\mathbf{x}_1^T\mathbf{x}_2) \end{bmatrix}$$

We show the eigenvalues for this submatrix below.

$$\lambda = \frac{1}{2}\left(2\psi_{1,t}\|q_t\|_2^2(v_t^T\mathbf{x}_1)^2 + 2\psi_{2,t}\|q_t\|_2^2(v_t^T\mathbf{x}_2)^2 - \phi_{1,t}v_t^T\mathbf{x}_1 + \phi_{2,t}v_t^T\mathbf{x}_2 + \frac{1}{2}(\psi_{1,t}+\psi_{2,t})\|q_t\|_2^4(1-\mathbf{x}_1^T\mathbf{x}_2)\right)$$

$$\pm \frac{1}{2}\left| \sqrt{ \begin{aligned} &\left(2\psi_{1,t}\|q_t\|_2^2(v_t^T\mathbf{x}_1)^2 + 2\psi_{2,t}\|q_t\|_2^2(v_t^T\mathbf{x}_2)^2 - \phi_{1,t}v_t^T\mathbf{x}_1 + \phi_{2,t}v_t^T\mathbf{x}_2 + \frac{1}{2}(\psi_{1,t}+\psi_{2,t})\|q_t\|_2^4(1-\mathbf{x}_1^T\mathbf{x}_2)\right)^2 \\ &-\left(4\psi_{1,t}\psi_{2,t}\|q_t\|_2^6\left(v_t^T\mathbf{x}_1 + v_t^T\mathbf{x}_2\right)^2 + 6\|q_t\|_2^4\left(\psi_{1,t}v_t^T\mathbf{x}_1 - \psi_{2,t}v_t^T\mathbf{x}_2\right)\left(\phi_{1,t}+\phi_{2,t}\right)\right. \\ &\left.+ 2\left(\phi_{2,t}\psi_{1,t}-\phi_{1,t}\psi_{2,t}\right)\|q_t\|_2^4\left(v_t^T\mathbf{x}_1 + v_t^T\mathbf{x}_2\right) - 4\left(\phi_{1,t}+\phi_{2,t}\right)^2\|q_t\|_2^2\right)\left(1-\mathbf{x}_1^T\mathbf{x}_2\right) \end{aligned} }\right|$$

Next, we solve for the Weyl's inequality bound decomposing the general matrix in the following way.

$$\begin{bmatrix} \begin{aligned} &2\psi_{1,t}\|q_t\|_2^2(v_t^T\mathbf{x}_1)^2 \\ &+ 2\psi_{2,t}\|q_t\|_2^2(v_t^T\mathbf{x}_2)^2 \\ &- \phi_{1,t}v_t^T\mathbf{x}_1 \\ &+ \phi_{2,t}v_t^T\mathbf{x}_2 \end{aligned} & 0 & \begin{aligned} &\|q_t\|_2\sqrt{1-\mathbf{x}_1^T\mathbf{x}_2} \\ &\cdot\left(\psi_{1,t}\|q_t\|_2^2 v_t^T\mathbf{x}_1\right. \\ &\left.-\psi_{2,t}\|q_t\|_2^2 v_t^T\mathbf{x}_2\right. \\ &\left.-\phi_{1,t}-\phi_{2,t}\right) \end{aligned} \\[2em] 0 & 0 & 0 \\[1em] \begin{aligned} &\|q_t\|_2\sqrt{1-\mathbf{x}_1^T\mathbf{x}_2} \\ &\cdot\left(\psi_{1,t}\|q_t\|_2^2 v_t^T\mathbf{x}_1\right. \\ &\left.-\psi_{2,t}\|q_t\|_2^2 v_t^T\mathbf{x}_2\right. \\ &\left.-\phi_{1,t}-\phi_{2,t}\right) \end{aligned} & 0 & \frac{1}{2}(\psi_{1,t}+\psi_{2,t})\|q_t\|_2^4(1-\mathbf{x}_1^T\mathbf{x}_2) \end{bmatrix}$$

$$+ \begin{bmatrix} 0 & \begin{aligned} &\|q_t\|_2\sqrt{1+\mathbf{x}_1^T\mathbf{x}_2} \\ &\cdot\left(\psi_{1,t}\|q_t\|_2^2 v_t^T\mathbf{x}_1\right. \\ &\left.+\psi_{2,t}\|q_t\|_2^2 v_t^T\mathbf{x}_2\right. \\ &\left.-\phi_{1,t}+\phi_{2,t}\right) \end{aligned} & 0 \\[2em] \begin{aligned} &\|q_t\|_2\sqrt{1+\mathbf{x}_1^T\mathbf{x}_2} \\ &\cdot\left(\psi_{1,t}\|q_t\|_2^2 v_t^T\mathbf{x}_1\right. \\ &\left.+\psi_{2,t}\|q_t\|_2^2 v_t^T\mathbf{x}_2\right. \\ &\left.-\phi_{1,t}+\phi_{2,t}\right) \end{aligned} & \frac{1}{2}(\psi_{1,t}+\psi_{2,t})\|q_t\|_2^4(1+\mathbf{x}_1^T\mathbf{x}_2) & \frac{1}{2}(\psi_{1,t}-\psi_{2,t})\|q_t\|_2^4\sqrt{1-(\mathbf{x}_1^T\mathbf{x}_2)^2} \\[2em] 0 & \frac{1}{2}(\psi_{1,t}-\psi_{2,t})\|q_t\|_2^4\sqrt{1-(\mathbf{x}_1^T\mathbf{x}_2)^2} & 0 \end{bmatrix}$$

From our calculations, we find the one of the eigenvalues of our perturbation matrix (the second matrix in the decomposition) is $0$. The other two eigenvalues are a symmetric pair that we show below, where the error bound for this approximation is the plus direction of the symmetric pair.

$$\lambda = \frac{1}{4}(\psi_{1,t} + \psi_{2,t})\|q_t\|_2^4(1 + \mathbf{x}_1^T\mathbf{x}_2)$$

$$\pm \frac{1}{2}\|q_t\|_2\sqrt{1 + \mathbf{x}_1^T\mathbf{x}_2}\sqrt{\begin{array}{l}\frac{1}{4}(\psi_{1,t} + \psi_{2,t})^2\|q_t\|_2^6(1 + \mathbf{x}_1^T\mathbf{x}_2) + (\psi_{1,t} - \psi_{2,t})^2\|q_t\|_2^6\left(1 - \mathbf{x}_1^T\mathbf{x}_2\right) \\ + 4\left(\psi_{1,t}\|q_t\|_2^2 v_t^T\mathbf{x}_1 + \psi_{2,t}\|q_t\|_2^2 v_t^T\mathbf{x}_2 - \phi_{1,t} + \phi_{2,t}\right)^2\end{array}}\Big)$$

**Approximation 3 Derivation**  Next, we consider the third submatrix formed by removing the third row and third column.

$$\begin{bmatrix} \begin{array}{l} 2\psi_{1,t}\|q_t\|_2^2(v_t^T\mathbf{x}_1)^2 \\ + 2\psi_{2,t}\|q_t\|_2^2(v_t^T\mathbf{x}_2)^2 \\ - \phi_{1,t}v_t^T\mathbf{x}_1 \\ + \phi_{2,t}v_t^T\mathbf{x}_2 \end{array} & \begin{array}{l} \|q_t\|_2\sqrt{1 + \mathbf{x}_1^T\mathbf{x}_2} \\ \cdot \Big(\psi_{1,t}\|q_t\|_2^2 v_t^T\mathbf{x}_1 \\ + \psi_{2,t}\|q_t\|_2^2 v_t^T\mathbf{x}_2 \\ - \phi_{1,t} + \phi_{2,t}\Big) \end{array} \\ \\ \begin{array}{l} \|q_t\|_2\sqrt{1 + \mathbf{x}_1^T\mathbf{x}_2} \\ \cdot \Big(\psi_{1,t}\|q_t\|_2^2 v_t^T\mathbf{x}_1 \\ + \psi_{2,t}\|q_t\|_2^2 v_t^T\mathbf{x}_2 \\ - \phi_{1,t} + \phi_{2,t}\Big) \end{array} & \frac{1}{2}(\psi_{1,t} + \psi_{2,t})\|q_t\|_2^4(1 + \mathbf{x}_1^T\mathbf{x}_2) \end{bmatrix}$$

We show the eigenvalues for this submatrix below.

$$\lambda = \frac{1}{2}\left(2\psi_{1,t}\|q_t\|_2^2(v_t^T\mathbf{x}_1)^2 + 2\psi_{2,t}\|q_t\|_2^2(v_t^T\mathbf{x}_2)^2 - \phi_{1,t}v_t^T\mathbf{x}_1 + \phi_{2,t}v_t^T\mathbf{x}_2 + \frac{1}{2}(\psi_{1,t} + \psi_{2,t})\|q_t\|_2^4(1 + \mathbf{x}_1^T\mathbf{x}_2)\right)$$

$$\pm \frac{1}{2}\sqrt{\begin{array}{l}\left(2\psi_{1,t}\|q_t\|_2^2(v_t^T\mathbf{x}_1)^2 + 2\psi_{2,t}\|q_t\|_2^2(v_t^T\mathbf{x}_2)^2 - \phi_{1,t}v_t^T\mathbf{x}_1 + \phi_{2,t}v_t^T\mathbf{x}_2 + \frac{1}{2}(\psi_{1,t} + \psi_{2,t})\|q_t\|_2^4(1 + \mathbf{x}_1^T\mathbf{x}_2)\right)^2 \\ - \left(4\psi_{1,t}\psi_{2,t}\|q_t\|_2^6\left(v_t^T\mathbf{x}_1 - v_t^T\mathbf{x}_2\right)^2 - 2\|q_t\|_2^4\left(\phi_{1,t}\psi_{2,t} + \phi_{2,t}\psi_{1,t}\right)\left(v_t^T\mathbf{x}_1 - v_t^T\mathbf{x}_2\right)\right. \\ \left. + 6\|q_t\|_2^4\left(\psi_{1,t}v_t^T\mathbf{x}_1 + \psi_{2,t}v_t^T\mathbf{x}_2\right)(\phi_{1,t} - \phi_{2,t}) - 4\left(\phi_{1,t} - \phi_{2,t}\right)^2\|q_t\|_2^2\right)\left(1 + \mathbf{x}_1^T\mathbf{x}_2\right)\end{array}}$$

Then, we can decompose the general matrix in the following way for the error bound.

$$
\begin{bmatrix}
\begin{array}{c} 2\psi_{1,t}\|q_t\|_2^2(v_t^T\mathbf{x}_1)^2 - \phi_{1,t}v_t^T\mathbf{x}_1 \\ + 2\psi_{2,t}\|q_t\|_2^2(v_t^T\mathbf{x}_2)^2 + \phi_{2,t}v_t^T\mathbf{x}_2 \end{array} & \begin{array}{c} \|q_t\|_2\sqrt{1+\mathbf{x}_1^T\mathbf{x}_2} \\ \cdot\left(\psi_{1,t}\|q_t\|_2^2 v_t^T\mathbf{x}_1 - \phi_{1,t}\right. \\ \left. + \psi_{2,t}\|q_t\|_2^2 v_t^T\mathbf{x}_2 + \phi_{2,t}\right) \end{array} & 0 \\[3em]
\begin{array}{c} \|q_t\|_2\sqrt{1+\mathbf{x}_1^T\mathbf{x}_2} \\ \cdot\left(\psi_{1,t}\|q_t\|_2^2 v_t^T\mathbf{x}_1 - \phi_{1,t}\right. \\ \left. + \psi_{2,t}\|q_t\|_2^2 v_t^T\mathbf{x}_2 + \phi_{2,t}\right) \end{array} & \frac{1}{2}(\psi_{1,t}+\psi_{2,t})\|q_t\|_2^4(1+\mathbf{x}_1^T\mathbf{x}_2) & 0 \\[3em]
0 & 0 & 0
\end{array}
\end{bmatrix}
$$

$$
+ \begin{bmatrix}
0 & 0 & \begin{array}{c} \|q_t\|_2\sqrt{1-\mathbf{x}_1^T\mathbf{x}_2} \\ \cdot\left(\psi_{1,t}\|q_t\|_2^2 v_t^T\mathbf{x}_1 - \phi_{1,t}\right. \\ \left. - \psi_{2,t}\|q_t\|_2^2 v_t^T\mathbf{x}_2 - \phi_{2,t}\right) \end{array} \\[3em]
0 & 0 & \frac{1}{2}(\psi_{1,t}-\psi_{2,t})\|q_t\|_2^4\sqrt{1-(\mathbf{x}_1^T\mathbf{x}_2)^2} \\[3em]
\begin{array}{c} \|q_t\|_2\sqrt{1-\mathbf{x}_1^T\mathbf{x}_2} \\ \cdot\left(\psi_{1,t}\|q_t\|_2^2 v_t^T\mathbf{x}_1 - \phi_{1,t}\right. \\ \left. - \psi_{2,t}\|q_t\|_2^2 v_t^T\mathbf{x}_2 - \phi_{2,t}\right) \end{array} & \frac{1}{2}(\psi_{1,t}-\psi_{2,t})\|q_t\|_2^4\sqrt{1-(\mathbf{x}_1^T\mathbf{x}_2)^2} & \frac{1}{2}(\psi_{1,t}+\psi_{2,t})\|q_t\|_2^4(1-\mathbf{x}_1^T\mathbf{x}_2)
\end{bmatrix}
$$

For the perturbation matrix (the second matrix in the decomposition), we get that one of the eigenvalues is 0 and the other two form a symmetric pair that we show below, where difference between the sharpness and the approximation from this submatrix is upper bounded by the plus direction of the symmetric pair.

$$
\lambda = \frac{1}{4}\left(\psi_{1,t}+\psi_{2,t}\right)\|q_t\|_2^4\left(1-\mathbf{x}_1^T\mathbf{x}_2\right)
$$

$$
\pm\frac{1}{2}\|q_t\|_2\sqrt{1-\mathbf{x}_1^T\mathbf{x}_2}\sqrt{\begin{array}{l} \frac{1}{4}\left(\psi_{1,t}+\psi_{2,t}\right)^2\|q_t\|_2^6\left(1-\mathbf{x}_1^T\mathbf{x}_2\right) \\ + \left(\psi_{1,t}-\psi_{2,t}\right)^2\|q_t\|_2^6\left(1+\mathbf{x}_1^T\mathbf{x}_2\right) \\ + 4\left(\psi_{1,t}\|q_t\|_2^2 v_t^T\mathbf{x}_1 - \phi_{1,t} - \psi_{2,t}\|q_t\|_2^2 v_t^T\mathbf{x}_2 - \phi_{2,t}\right)^2 \end{array}}
$$

**Approximation 4 Derivation** In this last approximation, we consider the middle element in the $\Lambda^{-\frac{1}{2}}QKQ\Lambda^{-\frac{1}{2}}$ matrix, $\frac{1}{2}(\psi_{1,t}+\psi_{2,t})\|q_t\|_2^4(1+\mathbf{x}_1^T\mathbf{x}_2)$, which is the second $1x1$ principal submatrix. Our motivation for focusing on this principal submatrix is from our analysis of the $\Lambda^{-\frac{1}{2}}QKQ\Lambda^{-\frac{1}{2}}$ matrix under the assumption that $c_t = 0$ (i.e., $v_t^T\mathbf{x}_1 = -v_t^T\mathbf{x}_2$), which causes $\psi_{1,t} = \psi_{2,t}$ and

$\phi_{1,t} = \phi_{2,t}$. Then, the $3x3$ matrix reduces to the following.

$$
\begin{bmatrix}
\begin{array}{l} 4\psi_{1,t}\|q_t\|_2^2(v_t^T\mathbf{x}_1)^2 \\ - 2\phi_{1,t}v_t^T\mathbf{x}_1 \end{array} & 0 & \begin{array}{l} 2\|q_t\|_2\sqrt{1 - \mathbf{x}_1^T\mathbf{x}_2} \\ \cdot\left(\psi_{1,t}\|q_t\|_2^2 v_t^T\mathbf{x}_1 - \phi_{1,t}\right) \end{array} \\[2em]
0 & \psi_{1,t}\|q_t\|_2^4(1 + \mathbf{x}_1^T\mathbf{x}_2) & 0 \\[2em]
\begin{array}{l} 2\|q_t\|_2\sqrt{1 - \mathbf{x}_1^T\mathbf{x}_2} \\ \cdot\left(\psi_{1,t}\|q_t\|_2^2 v_t^T\mathbf{x}_1 - \phi_{1,t}\right) \end{array} & 0 & \psi_{1,t}\|q_t\|_2^4(1 - \mathbf{x}_1^T\mathbf{x}_2)
\end{array}
\end{bmatrix}
$$

Given the sparsity of the above matrix, we can directly compute the eigenvalues. We know that one eigenvalue is $\psi_{1,t}\|q_t\|_2^4(1 + \mathbf{x}_1^T\mathbf{x}_2)$. We show the symmetric pair of eigenvalues below.

$$
2\psi_{1,t}\|q_t\|_2^2(v_t^T\mathbf{x}_1)^2 - \phi_{1,t}v_t^T\mathbf{x}_1 + \frac{1}{2}\psi_{1,t}\|q_t\|_2^4(1 - \mathbf{x}_1^T\mathbf{x}_2)
$$

$$
\pm \sqrt{\left(2\psi_{1,t}\|q_t\|_2^2(v_t^T\mathbf{x}_1)^2 - \phi_{1,t}v_t^T\mathbf{x}_1 + \frac{1}{2}\psi_{1,t}\|q_t\|_2^4(1 - \mathbf{x}_1^T\mathbf{x}_2)\right)^2 - 6\phi_{1,t}\psi_{1,t}\|q_t\|_2^4 v_t^T\mathbf{x}_1(1 - \mathbf{x}_1^T\mathbf{x}_2) - 4\phi_{1,t}^2\|q_t\|_2^2(1 - \mathbf{x}_1^T\mathbf{x}_2)}
$$

From experimental evaluation, we compare the $\psi_{1,t}\|q_t\|_2^4(1 + \mathbf{x}_1^T\mathbf{x}_2)$ eigenvalue with the positive direction of the symmetric pair of eigenvalues above and find that the former is the exact eigenvalue in our setting under the $c_t = 0$ assumption, shown in Figures 7 and 8. For conciseness, we refer to $\psi_{1,t}\|q_t\|_2^4(1 + \mathbf{x}_1^T\mathbf{x}_2)$ as $eig\_g1$ and the other eigenvalue as $eig\_g2$ in the figures.

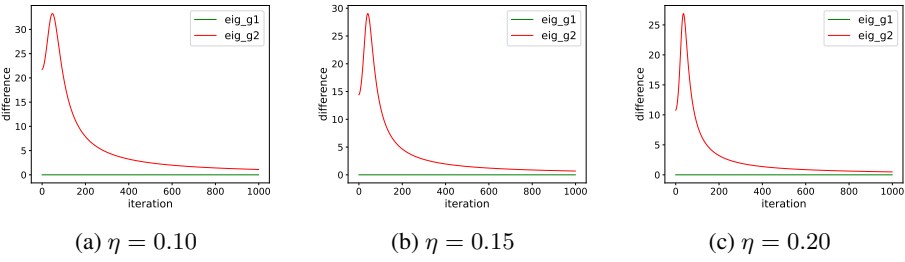

(a) $\eta = 0.10$      (b) $\eta = 0.15$      (c) $\eta = 0.20$

Figure 7: We show the differences between the sharpness and both eigenvalues we calculated for the $c_0 = 0$ case, where $\eta \in \{0.10, 0.15, 0.20\}$, $\mathbf{x}_1^T\mathbf{x}_2 = 0.99$, $\|q_0\|_2^2 = 1.05\sqrt{\frac{8}{0.13 \cdot 1.99}}$, and $m_0 = 0.01$.

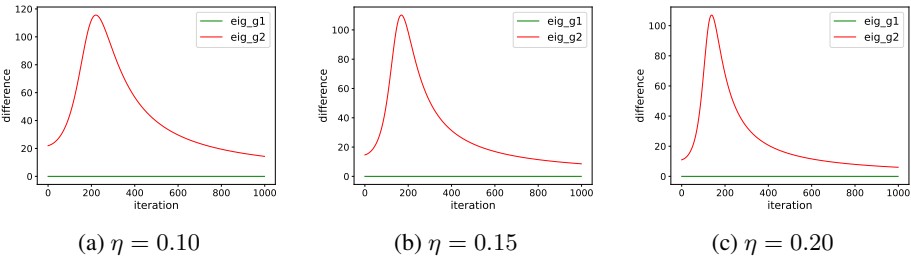

(a) $\eta = 0.10$      (b) $\eta = 0.15$      (c) $\eta = 0.20$

Figure 8: We show the differences between the sharpness and both eigenvalues we calculated for the $c_0 = 0$ case, where $\eta \in \{0.10, 0.15, 0.20\}$, $\mathbf{x}_1^T\mathbf{x}_2 = 0.9991$, $\|q_0\|_2^2 = 1.05\sqrt{\frac{8}{0.13 \cdot 1.99}}$, and $m_0 = 0.01$.

Since $\frac{1}{2}(\psi_{1,t} + \psi_{2,t})\|q_t\|_2^4(1 + \mathbf{x}_1^T\mathbf{x}_2)$ reduces to $\psi_{1,t}\|q_t\|_2^4(1 + \mathbf{x}_1^T\mathbf{x}_2)$ under the $c_t = 0$ assumption and is easier to interpret than the prior approximations, we give it consideration.

Now, we calculate Weyl's Inequality, starting from the following decomposition.

$$\begin{bmatrix} 0 & 0 & 0 \\ 0 & \frac{1}{2}(\psi_{1,t} + \psi_{2,t})\|q_t\|_2^4(1 + \mathbf{x}_1^T\mathbf{x}_2) & 0 \\ 0 & 0 & 0 \end{bmatrix}$$

$$+ \begin{bmatrix} \begin{matrix} 2\psi_{1,t}\|q_t\|_2^2(v_t^T\mathbf{x}_1)^2 \\ + 2\psi_{2,t}\|q_t\|_2^2(v_t^T\mathbf{x}_2)^2 \\ - \phi_{1,t}v_t^T\mathbf{x}_1 \\ + \phi_{2,t}v_t^T\mathbf{x}_2 \end{matrix} & \begin{matrix} \|q_t\|_2\sqrt{1 + \mathbf{x}_1^T\mathbf{x}_2} \\ \cdot \left( \psi_{1,t}\|q_t\|_2^2 v_t^T\mathbf{x}_1 \right. \\ + \psi_{2,t}\|q_t\|_2^2 v_t^T\mathbf{x}_2 \\ \left. - \phi_{1,t} + \phi_{2,t} \right) \end{matrix} & \begin{matrix} \|q_t\|_2\sqrt{1 - \mathbf{x}_1^T\mathbf{x}_2} \\ \cdot \left( \psi_{1,t}\|q_t\|_2^2 v_t^T\mathbf{x}_1 \right. \\ - \psi_{2,t}\|q_t\|_2^2 v_t^T\mathbf{x}_2 \\ \left. - \phi_{1,t} - \phi_{2,t} \right) \end{matrix} \\ \begin{matrix} \|q_t\|_2\sqrt{1 + \mathbf{x}_1^T\mathbf{x}_2} \\ \cdot \left( \psi_{1,t}\|q_t\|_2^2 v_t^T\mathbf{x}_1 \right. \\ + \psi_{2,t}\|q_t\|_2^2 v_t^T\mathbf{x}_2 \\ \left. - \phi_{1,t} + \phi_{2,t} \right) \end{matrix} & 0 & \frac{1}{2}(\psi_{1,t} - \psi_{2,t})\|q_t\|_2^4\sqrt{1 - (\mathbf{x}_1^T\mathbf{x}_2)^2} \\ \begin{matrix} \|q_t\|_2\sqrt{1 - \mathbf{x}_1^T\mathbf{x}_2} \\ \cdot \left( \psi_{1,t}\|q_t\|_2^2 v_t^T\mathbf{x}_1 \right. \\ - \psi_{2,t}\|q_t\|_2^2 v_t^T\mathbf{x}_2 \\ \left. - \phi_{1,t} - \phi_{2,t} \right) \end{matrix} & \frac{1}{2}(\psi_{1,t} - \psi_{2,t})\|q_t\|_2^4\sqrt{1 - (\mathbf{x}_1^T\mathbf{x}_2)^2} & \frac{1}{2}(\psi_{1,t} + \psi_{2,t})\|q_t\|_2^4(1 - \mathbf{x}_1^T\mathbf{x}_2) \end{bmatrix}$$

Since the perturbation matrix (second matrix in the decomposition) is essentially as complex as the original $3x3$ matrix, we calculate the Frobenius norm of the perturbation matrix instead as our error bound.

$$\sqrt{\begin{matrix} \left(2\psi_{1,t}\|q_t\|_2^2(v_t^T\mathbf{x}_1)^2 + 2\psi_{2,t}\|q_t\|_2^2(v_t^T\mathbf{x}_2)^2 - \phi_{1,t}v_t^T\mathbf{x}_1 + \phi_{2,t}v_t^T\mathbf{x}_2\right)^2 \\ + 2\|q_t\|_2^2\left(1 + \mathbf{x}_1^T\mathbf{x}_2\right)\left(\psi_{1,t}\|q_t\|_2^2 v_t^T\mathbf{x}_1 + \psi_{2,t}\|q_t\|_2^2 v_t^T\mathbf{x}_2 - \phi_{1,t} + \phi_{2,t}\right)^2 \\ + 2\|q_t\|_2^2\left(1 - \mathbf{x}_1^T\mathbf{x}_2\right)\left(\psi_{1,t}\|q_t\|_2^2 v_t^T\mathbf{x}_1 - \psi_{2,t}\|q_t\|_2^2 v_t^T\mathbf{x}_2 - \phi_{1,t} - \phi_{2,t}\right)^2 \\ + \frac{1}{2}\left(\psi_{1,t} - \psi_{2,t}\right)^2\|q_t\|_2^8\left(1 - (\mathbf{x}_1^T\mathbf{x}_2)^2\right) + \frac{1}{4}\left(\psi_{1,t} + \psi_{2,t}\right)^2\|q_t\|_2^8\left(1 - \mathbf{x}_1^T\mathbf{x}_2\right)^2 \end{matrix}}$$

Next, we analyze the error bounds for our different approximations and provide some empirical results to support our claim that Approximation 3 is a low-error approximation for the sharpness in our setting.

**Analysis of the Approximations** From the approximations we derived, we note that approximation 3 appears to the most accurate in our setting. In comparison to the error bound that we calculated for approximation 2, we note that it is quite similar in form to the error bound for approximation 3. We note that difference in whether $1 + \mathbf{x}_1^T\mathbf{x}_2$ or $1 - \mathbf{x}_1^T\mathbf{x}_2$ are prominent factors in the error bound. Given that we assume $0.99 \leq \mathbf{x}_1^T\mathbf{x}_2 < 1$, the error bounds suggests that approximation 3 is more accurate than approximation 2 in our setting. As for approximations 1 and 4, we determine that approximation 3 is more accurate than them by experimental evaluation, shown in Figures 9 and 10.

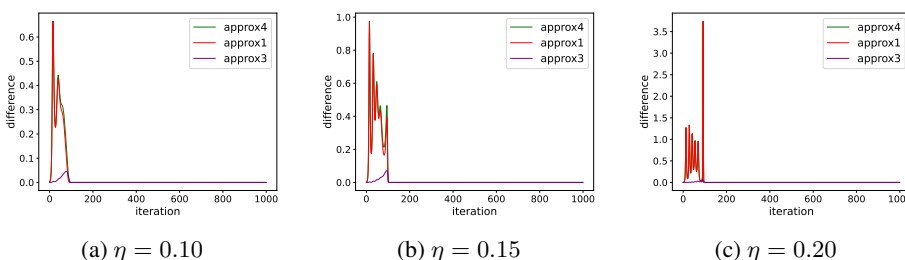

(a) $\eta = 0.10$    (b) $\eta = 0.15$    (c) $\eta = 0.20$

Figure 9: We show the differences between the sharpness and approximations 1, 3, and 4 for $\eta \in \{0.10, 0.15, 0.20\}$, $\mathbf{x}_1^T x2 = 0.99$, $\|q_0\|_2^2 = 1.05\sqrt{\frac{8}{0.13 \cdot 1.99}}$, $c_0 = 0.02$, and $m_0 = 0.01$.

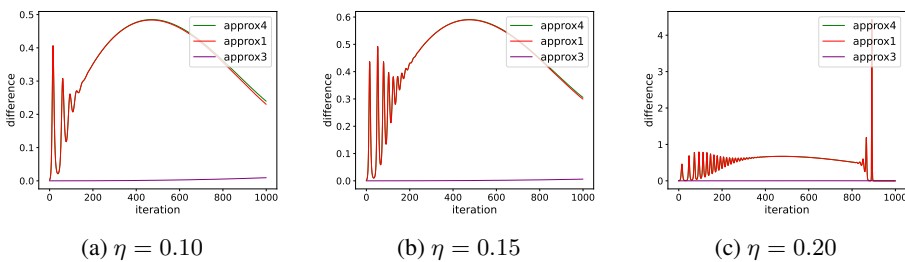

(a) $\eta = 0.10$    (b) $\eta = 0.15$    (c) $\eta = 0.20$

Figure 10: We show the differences between the sharpness and approximations 1, 3, and 4 for $\eta \in \{0.10, 0.15, 0.20\}$, $\mathbf{x}_1^T \mathbf{x}_2 = 0.99$, $\|q_0\|_2^2 = 1.05\sqrt{\frac{8}{0.13 \cdot 1.9991}}$, $c_0 = 0.02$, and $m_0 = 0.01$.

Furthermore, we also note that the differences of approximations 1 and 4 to the true sharpness are quite similar.

From this, we denote approximation 3 as our low-error sharpness approximation and approximation 4 as our interpretable sharpness approximation. We further note that our interpretable approximation supports the idea proposed by Wang et al. (2022) about the correlation between the sharpness and the norm of the output layer of a two-layer linear model. However, we see that the correlation is not direct and more subtle.

**Additional Eigenvalues of Loss Hessian**   In addition to our sharpness approximations, we provide closed-forms for all but the top three largest (in algebraic value) eigenvalues of the Loss Hessian.

We start with

$$\nabla_{\theta_t} L(q_t, u_t, v_t) - \lambda I_{2h+d}$$
$$= \begin{bmatrix} A & B & C \\ B & A & C \\ C^T & C^T & D \end{bmatrix}$$

where

$$A = \left[\psi_{1,t}\left(v_t^T \mathbf{x}_1\right)^2 + \psi_{2,t}\left(v_t^T \mathbf{x}_2\right)^2\right] q_t q_t^T - \lambda I_h$$
$$B = -\left[\phi_{1,t} v_t^T \mathbf{x}_1 - \phi_{2,t} v_t^T \mathbf{x}_2\right] I_h + \left[\psi_{1,t}\left(v_t^T \mathbf{x}_1\right)^2 + \psi_{2,t}\left(v_t^T \mathbf{x}_2\right)^2\right] q_t q_t^T$$
$$C = -q_t \left[\phi_{1,t}\mathbf{x}_1^T - \phi_{2,t}\mathbf{x}_2^T\right] + \|q_t\|_2^2 q_t \left[\psi_{1,t} v_t^T \mathbf{x}_1 \mathbf{x}_1^T + \psi_{2,t} v_t^T \mathbf{x}_2 \mathbf{x}_2^T\right]$$
$$D = \|q_t\|_2^4 \left[\psi_{1,t}\mathbf{x}_1\mathbf{x}_1^T + \psi_{2,t}\mathbf{x}_2\mathbf{x}_2^T\right] - \lambda I_d.$$

From here, we follow a similar calculation as in the single datapoint case, except we use Woodbury Matrix Inversion instead of the Sherman-Morrison Formula to handle the $D$ matrix. From those calculations, we find a closed-form for all but the top three eigenvalues that are analogous to those

in the single datapoint case. There are $h$ copies of $\phi_{1,t} v_t^T \mathbf{x}_1 - \phi_{2,t} v_t^T \mathbf{x}_2$, $h-1$ copies of $-\phi_{1,t} v_t^T \mathbf{x}_1 + \phi_{2,t} v_t^T \mathbf{x}_2$, and $d-2$ copies of 0.

### C.2.3 EXPLAINING SHARPNESS TRAJECTORY

In this section, we revisit the training trajectory that we studied in Section 4, shown in Figure 11.

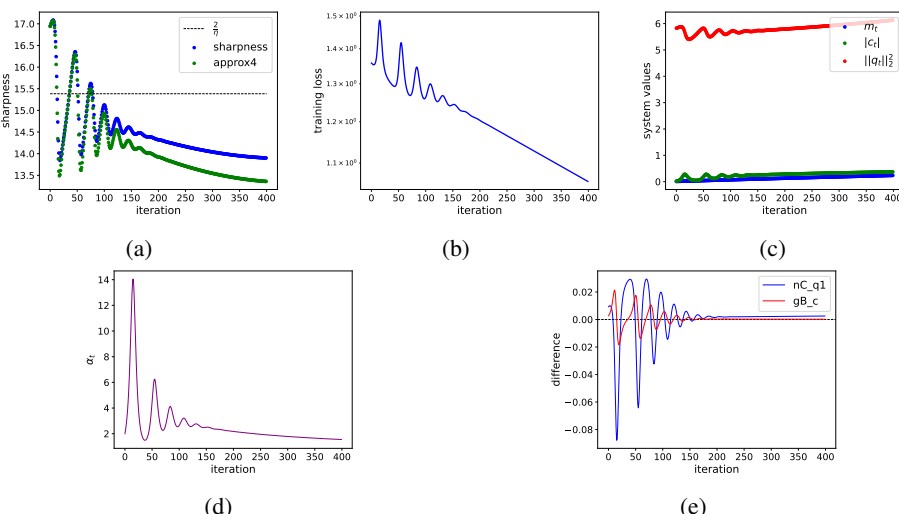

Figure 11: We show the sharpness (a), training loss (b), system values (c), $\alpha$ (i.e., $\frac{|c_t|}{m_t}$) (d), and "location" of the system relative to $\|q_t\|\_2^2$ to nullcline 1 ($nC\_q1 = 0$) and $|c_t|$ the growth boundary ($gB\_c = 0$) (e) for $\eta = 0.13$, $\|q_0\|_2^2 = 1.05\sqrt{\frac{8}{0.13 \cdot 1.9991}}$, $c_0 = 0.02$, and $m_0 = 0.01$. We also show the similarity between our sharpness approximation and the true sharpness.

Recall from Section 4 that we defined $\|q_t\|_2^2$ nullcline 1 as

$$\left(1 + \exp\left(\frac{1}{2}\|q_t\|_2^2 (m_t + c_t)\right)\right)^{-1} (m_t + c_t) + \left(1 + \exp\left(\frac{1}{2}\|q_t\|_2^2 (m_t - c_t)\right)\right)^{-1} (m_t - c_t) = 0,$$

the $|c_t|$ growth boundary as

$$\left(1 + \exp\left(\frac{1}{2}\|q_t\|_2^2 (m_t - |c_t|)\right)\right)^{-1} - \left(1 + \exp\left(\frac{1}{2}\|q_t\|_2^2 (m_t + |c_t|)\right)\right)^{-1} = \frac{2|c_t|}{\eta\|q_t\|_2^2 \left(1 + \mathbf{x}_1^T \mathbf{x}_2\right)},$$

and split the initial trajectory into four phases based on the sign of $nC\_q1$ and $gB\_c$ in Figure 11(e), defined below.

In the initial part of the trajectory, we see that the system is inside $\|q_t\|_2^2$ nullcline 1 and the $|c_t|$ growth boundary ($nC\_q1 > 0, gB\_c > 0$), which we denote as phase 1. Then, the system is outside $\|q_t\|_2^2$ nullcline 1, but still inside the $|c_t|$ growth boundary ($nC\_q1 < 0, gB\_c > 0$), which we denote as phase 2. In phase 3, we see that the system is outside both $\|q_t\|_2^2$ nullcline 1 and the $|c_t|$ growth boundary ($nC\_q1 < 0, gB\_c < 0$). Lastly, in phase 4, the system is inside $\|q_t\|_2^2$ nullcline 1, but still outside the $|c_t|$ growth boundary ($nC\_q1 > 0, gB\_c < 0$). To explain the sharpness trajectory, we use the interpretable sharpness approximation we derived in Appendix C.2.2, shown below, to explain the first oscillation in sharpness.

$$\frac{1}{2}(\psi_{1,t} + \psi_{2,t})\|q_t\|_2^4(1 + \mathbf{x}_1^T \mathbf{x}_2),$$

where

$$\psi_{1,t} = \left(1 + \exp\left(\frac{1}{2}\|q_t\|_2^2(m_t + c_t)\right)\right)^{-2} \exp\left(\frac{1}{2}\|q_t\|_2^2(m_t + c_t)\right)$$

$$\psi_{2,t} = \left(1 + \exp\left(\frac{1}{2}\|q_t\|_2^2(m_t - c_t)\right)\right)^{-2} \exp\left(\frac{1}{2}\|q_t\|_2^2(m_t - c_t)\right).$$

By rewriting the sharpness in terms of $\alpha_t$, we note an initial increase in the sharpness as the norm of $\|q_t\|_2^2$ and $\alpha_t$ grow in phase 1. However, the sigmoid second derivative terms in the approximation start to dominate the growth of $\|q_t\|_2^2$ due to the fast growth of $\alpha_t$, which causes the sharpness to begin to shrink for the remainder of phase 1. In phase 2, the sharpness continues to shrink $\alpha_t$ still grows quite fast, causing the sigmoid second derivative terms to continue to dominate even as $\|q_t\|_2^2$ shrinks. Initially in phase 3, we find that the sharpness continues to decrease as $\alpha_t$ initially shrinks slowly. However, as $\alpha_t$ shrinks faster, we begin to observe an increase in sharpness. We can think of this as the sigmoid second derivative terms allowing more of $\|q_t\|_2^2$ to be "present" in the sharpness, where the "presence" $\|q_t\|_2^2$ increases over time from some small initial amount. In phase 4, the shrinkage in $\alpha_t$ and growth in $\|q_t\|_2^2$ continues to cause the sharpness to increase, as $\|q_t\|_2^2$ dominates the sigmoid second derivative terms. As seen from Figure 11(e), the system can enter phase 1, and the cycle repeats. As mentioned in Section 4, we note that the dampening in the oscillations for the sharpness is likely the result of the growth of $m_t$, which constrains the magnitude of $\alpha_t$ (see Figure 11(d)).

### C.2.4 ADDITIONAL EXPERIMENTS FOR EDGE OF STABILITY

In this section, we provide additional experiments to show Edge of Stability behavior in our two datapoint setting. Our experiments consider $\mathbf{x}_1^T\mathbf{x}_2 \in \{0.99, 0.9991\}$, $\eta \in \{0.10, 0.12, 0.15, 0.17, 0.20\}$, $\|q_0\|_2^2 = 1.05\sqrt{\frac{8}{\eta(1+\mathbf{x}_1^T x2)}}$, $c_0 = 0.02$, and $m = 0.01$, where each model is trained for 1000 iterations with hidden width 1024. In each of the figures below, we show the sharpness, loss, and system trajectories.

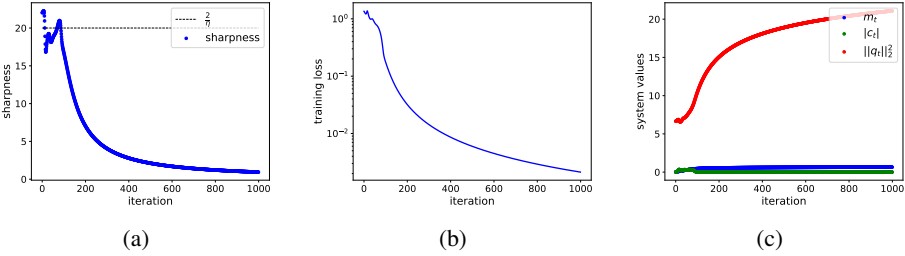

|     |     |     |
|:---:|:---:|:---:|
| (a) | (b) | (c) |

Figure 12: We show the sharpness (a), training loss (b), and system trajectory (c) for $\mathbf{x}_1^T\mathbf{x}_2 = 0.99$ and $\eta = 0.10$.

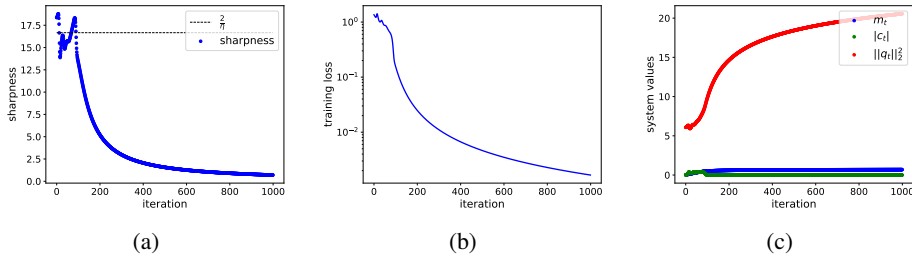

|     |     |     |
|:---:|:---:|:---:|
| (a) | (b) | (c) |

Figure 13: We show the sharpness (a), training loss (b), and system trajectory (c) for $\mathbf{x}_1^T\mathbf{x}_2 = 0.99$ and $\eta = 0.12$.

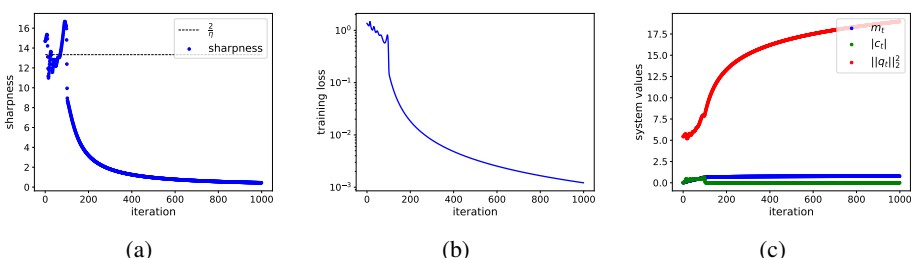

Figure 14: We show the sharpness (a), training loss (b), and system trajectory (c) for $\mathbf{x}_1^T \mathbf{x}_2 = 0.99$ and $\eta = 0.15$.

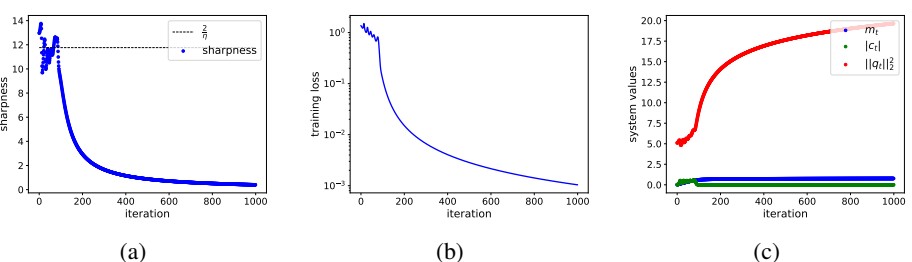

Figure 15: We show the sharpness (a), training loss (b), and system trajectory (c) for $\mathbf{x}_1^T \mathbf{x}_2 = 0.99$ and $\eta = 0.17$.

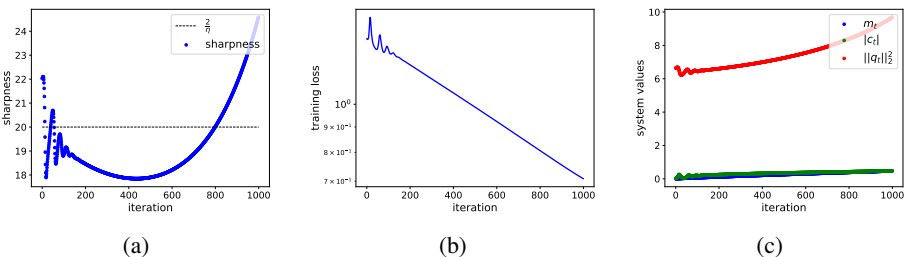

Figure 16: We show the sharpness (a), training loss (b), and system trajectory (c) for $\mathbf{x}_1^T \mathbf{x}_2 = 0.99$ and $\eta = 0.20$.

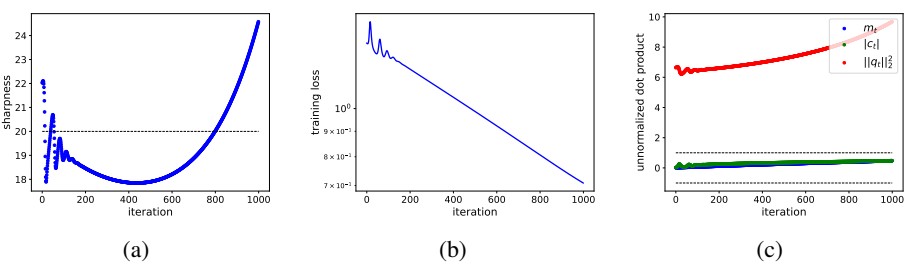

Figure 17: We show the sharpness (a), training loss (b), and system trajectory (c) for $\mathbf{x}_1^T \mathbf{x}_2 = 0.9991$ and $\eta = 0.10$.

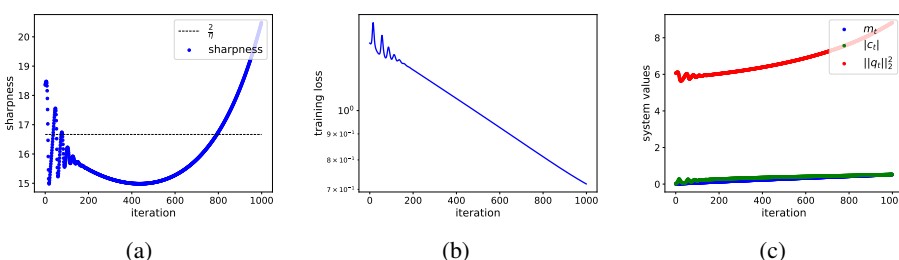

Figure 18: We show the sharpness (a), training loss (b), and system trajectory (c) for $\mathbf{x}_1^T\mathbf{x}_2 = 0.9991$ and $\eta = 0.12$.

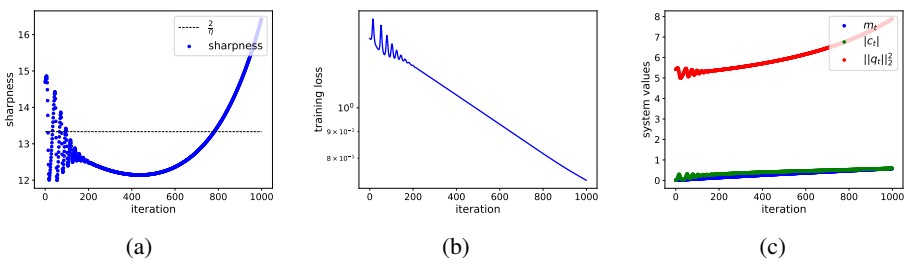

Figure 19: We show the sharpness (a), training loss (b), and system trajectory (c) for $\mathbf{x}_1^T\mathbf{x}_2 = 0.9991$ and $\eta = 0.15$.

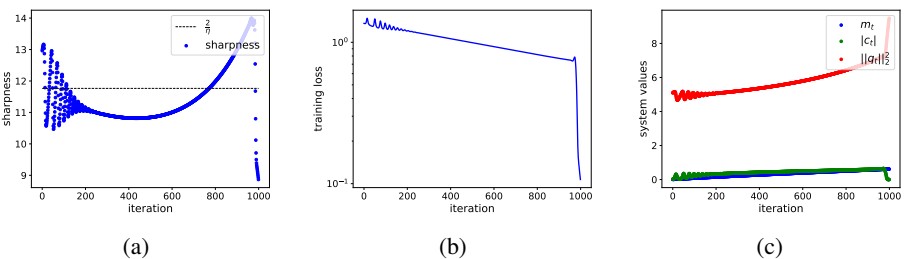

Figure 20: We show the sharpness (a), training loss (b), and system trajectory (c) for $\mathbf{x}_1^T\mathbf{x}_2 = 0.9991$ and $\eta = 0.17$.

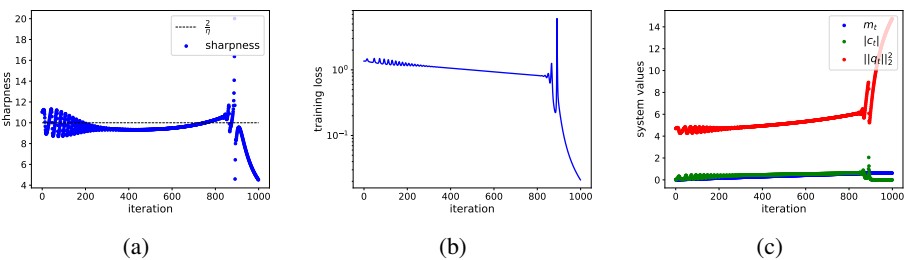

Figure 21: We show the sharpness (a), training loss (b), and system trajectory (c) for $\mathbf{x}_1^T\mathbf{x}_2 = 0.9991$ and $\eta = 0.20$.

### C.2.5 ADDITIONAL SIMULATIONS OF ASYMPTOTIC SETTING

In this section, we provide additional simulations of our model in the asymptotic case. Recall from Section 4 that our system reduces to $(\|q_t\|_2^2, c_t)$ and $m_t$ becomes constant in the asymptotic setting, where we consider $\mathbf{x}_1^T \mathbf{x}_2 \to 1$. Our experiments consist of additional cases where the system $(\|q_t\|_2^2, c_t)$ jumps between different values, exhibiting a "band" in the trajectory. For each case, we show the loss, sharpness, and system trajectories, focusing on $|c_t|$ instead of $c_t$, as we find magnitude to be most relevant for loss and sharpness. We also provide zoomed-in versions of the plots to make values that the system jumps between more clear.

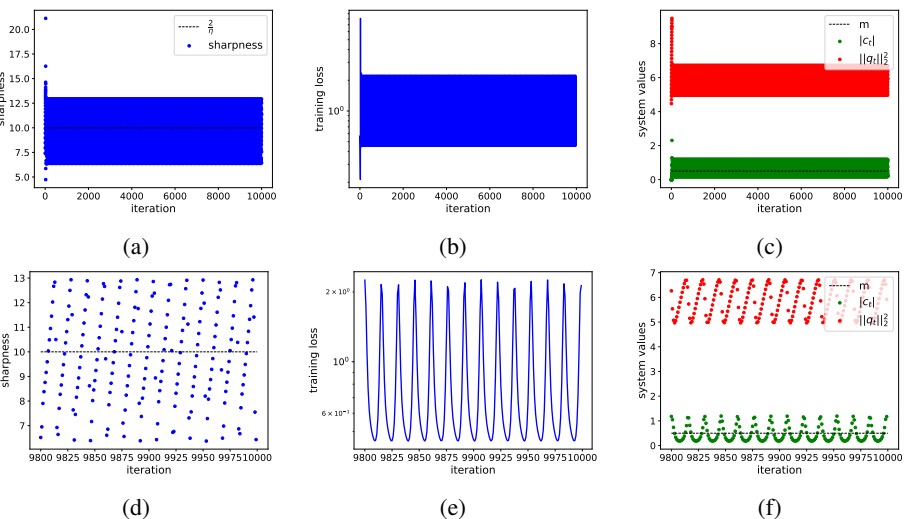

Figure 22: We show the sharpness (a), training loss (b), system iterate evolution (c) for $\eta = 0.20$, m=0.5, $\|q_0\|_2^2 = \frac{2}{\sqrt{0.2}}$, and $c_0 = 10^{-3}$. We also provide zoomed-in versions of the sharpness (d), training loss (e), and system iterate evolution (f).

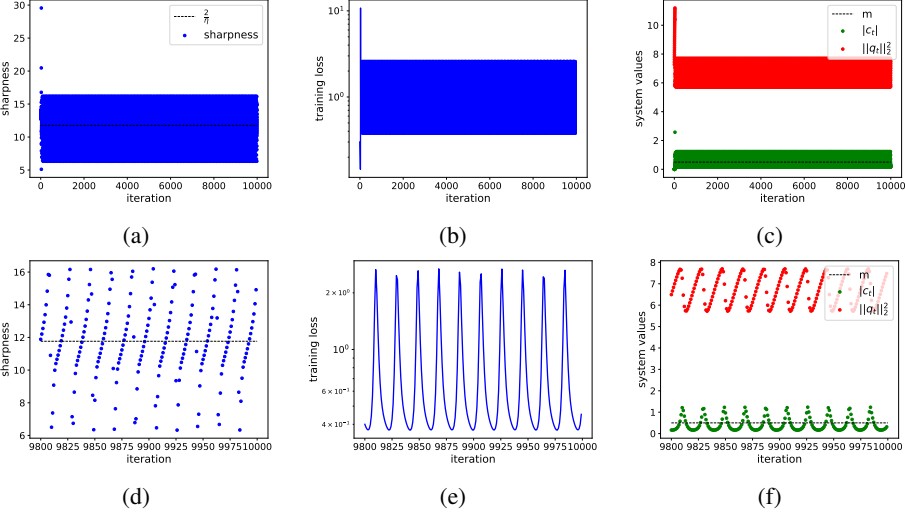

Figure 23: We show the sharpness (a), training loss (b), system iterate evolution (c) for $\eta = 0.17$, m=0.5, $\|q_0\|_2^2 = \frac{3}{\sqrt{0.17}}$, and $c_0 = 10^{-4}$. We also provide zoomed-in versions of the sharpness (d), training loss (e), and system iterate evolution (f).

Recall from Section 4 that we defined $\|q_t\|_2^2$ nullcline 1 as

$$\left(1 + \exp\left(\frac{1}{2}\|q_t\|_2^2 (m_t + c_t)\right)\right)^{-1} (m_t + c_t) + \left(1 + \exp\left(\frac{1}{2}\|q_t\|_2^2 (m_t - c_t)\right)\right)^{-1} (m_t - c_t) = 0$$

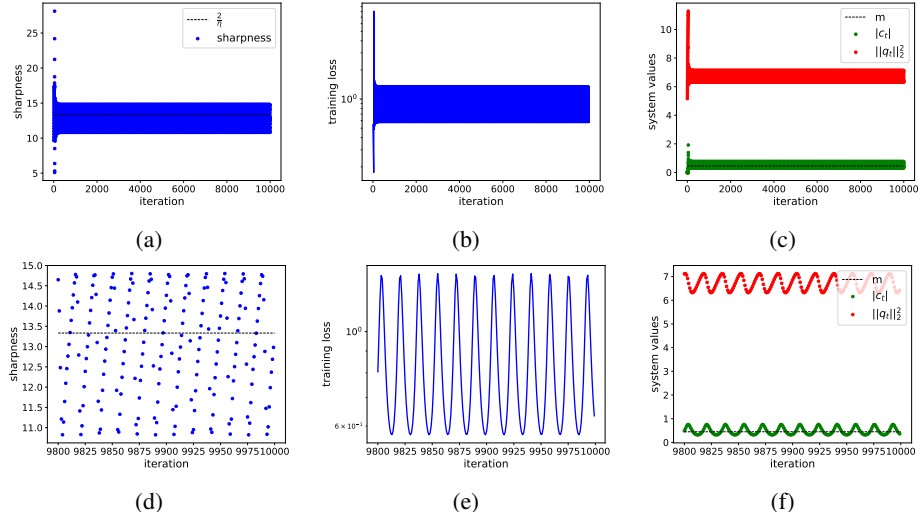

(a)                 (b)                 (c)

(d)                 (e)                 (f)

Figure 24: We show the sharpness (a), training loss (b), system iterate evolution (c) for $\eta = 0.15$, m=0.45, $||q_0||_2^2 = \frac{2}{\sqrt{0.15}}$, and $c_0 = 10^{-4}$. We also provide zoomed-in versions of the sharpness (d), training loss (e), and system iterate evolution (f).

and the $|c_t|$ growth boundary as

$$\left(1 + \exp\left(\frac{1}{2}\|q_t\|_2^2 (m_t - |c_t|)\right)\right)^{-1} - \left(1 + \exp\left(\frac{1}{2}\|q_t\|_2^2 (m_t + |c_t|)\right)\right)^{-1} = \frac{2|c_t|}{\eta\|q_t\|_2^2 \left(1 + \mathbf{x}_1^T \mathbf{x}_2\right)}.$$

We also show cases for $m = 0.45, \eta = 0.20$, where the model appears to have entered a stable orbit around the intersection of $\|q_t\|_2^2$ nullcline 1 and $|c_t|$ growth boundary. For each case, we show the loss, sharpness, and system trajectories, focusing with $|c_t|$ instead of $c_t$, as we find magnitude to be most relevant for loss and sharpness. We also provide zoomed-in versions of the plots, since the scale of the original plots squishes the oscillations in the tail ends of each trajectory.

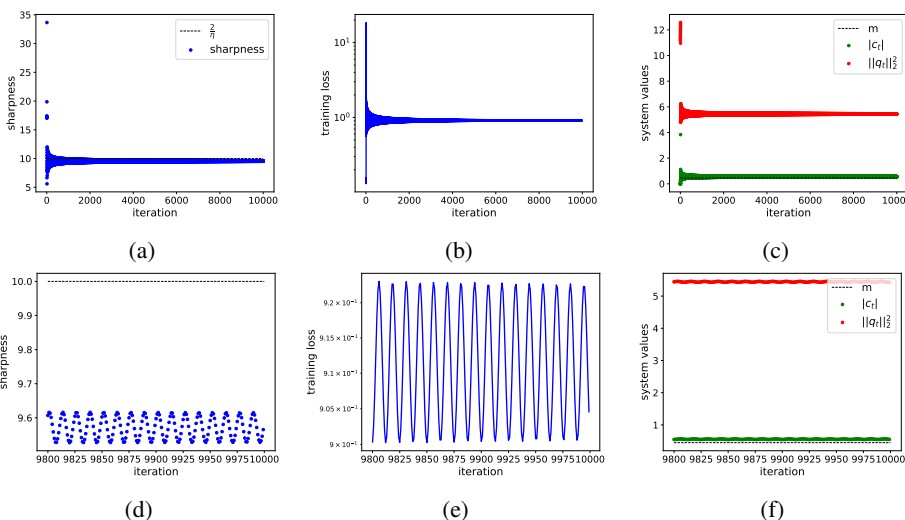

(a)                 (b)                 (c)

(d)                 (e)                 (f)

Figure 25: We show the sharpness (a), training loss (b), system trajectory (c) for $||q_0||_2^2 = \frac{5}{\sqrt{0.2}}$, and $c_0 = 10^{-4}$. We also provide zoomed-in versions of the sharpness (d), training loss (e), and system trajectory (f).

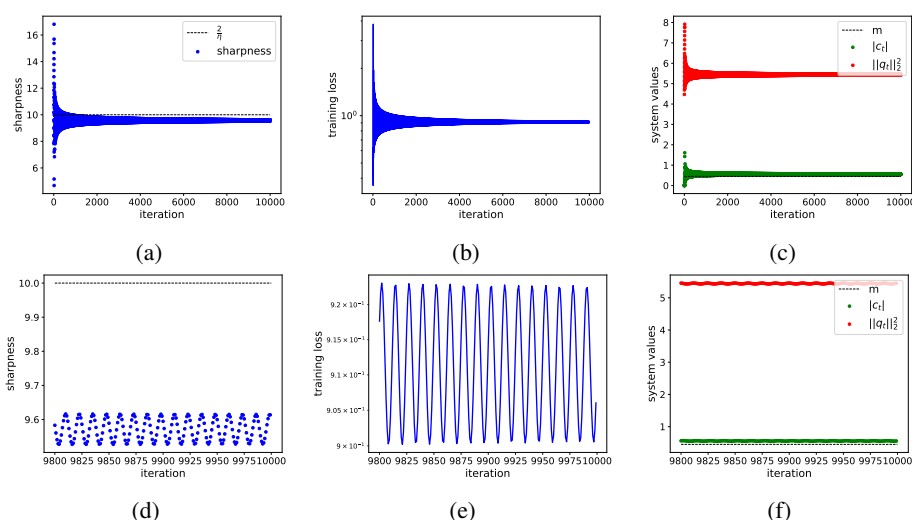

Figure 26: We show the sharpness (a), training loss (b), system trajectory (c) for $||q_0||_2^2 = \frac{2}{\sqrt{0.2}}$, and $c_0 = 10^{-2}$. We also provide zoomed-in versions of the sharpness (d), training loss (e), and system trajectory (f).

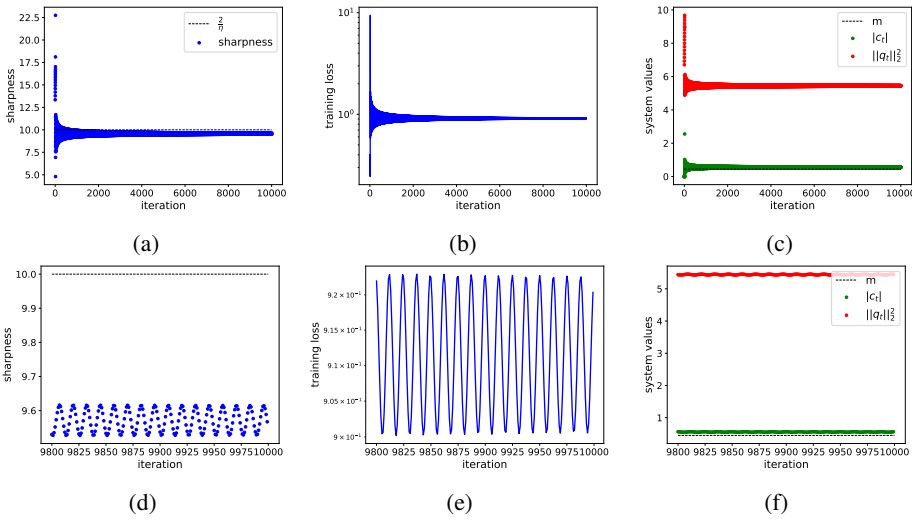

Figure 27: We show the sharpness, training loss, system trajectory for $||q_0||_2^2 = \frac{3}{\sqrt{0.2}}$, and $c_0 = 10^{-5}$. We also provide zoomed-in versions of the sharpness (d), training loss (e), and system trajectory (f).

