# OpenReview forum: "Understanding Edge of Stability in Rank-1 Linear Models for Binary Classification"
_ICLR.cc/2026/Conference — Submitted to ICLR 2026_

### Official Review · Reviewer_NUXe · 2025-10-28

**Soundness:** 2
**Presentation:** 2
**Contribution:** 2
**Rating:** 4
**Confidence:** 4

**Summary:**

This paper studies the Edge of Stability (EoS) phenomenon—where training sharpness oscillates around the stability threshold—in a minimal setting: a two-layer rank-1 linear model trained with logistic loss. The authors rigorously prove that EoS cannot occur in the single-datapoint case and empirically show that it can appear when two datapoints are used. They further provide an analytical approximation of the Hessian’s top eigenvalue (sharpness), interpret the oscillatory behavior through phase-space analysis, and explore an asymptotic case as the data margin approaches zero, where perpetual oscillation may emerge.

**Strengths:**

1. Theorem 4.2 rigorously demonstrates that with a single datapoint, the loss can have at most one local maximum, thus ruling out EoS.
2. The partitioning into nullcline regions and use of fixed-point analysis make the dynamics interpretable.

**Weaknesses:**

1. While the two-point empirical results show oscillations reminiscent of EoS, the theoretical results are limited. The analysis does not establish a rigorous proof that EoS truly occurs even in the two-point setting.

2. While the paper discusses oscillations in sharpness qualitatively reminiscent of the EoS threshold 2/η, it does not theoretically establish or derive this condition from the model’s dynamics.

**Questions:**

See Weakness.

---

### Official Review · Reviewer_Daug · 2025-10-30

**Soundness:** 3
**Presentation:** 3
**Contribution:** 2
**Rating:** 4
**Confidence:** 4

**Summary:**

This paper studies edge of stability dynamics in a specific setup: the model is a two-layer linear net where the first layer's weight matrix is rank-1, and the loss function is the logistic loss.  The paper proves that with one datapoint, edge of stability (which they define as multiple local maxima in the train loss curve) is impossible.  That is, the loss can go up and then down ("catapult"), but it can't then go back up again.  The intuition for the proof can be seen from drawing a 2d diagram of the possible states of the system and reasoning about which transitions are possible.  The result stands in contrast to MSE loss, where it has been previously shown that EOS does happen with one data point.  The paper then considers the case of two datapoints, shows experimentally that edge of stability is possible, and gives some reasoning for why that is.  Finally, the paper conjectures that in a certain asymptotic setting, where the correlation between the two input data points goes to 1, the dynamics enter a stable orbit.

**Strengths:**

I enjoyed the proof (which has good visual intuition) that edge of stability is impossible with logistic loss and 1 datapoint.

**Weaknesses:**

The main weaknesses are that the paper studies a very specific setting (2 layer linear network with rank-1 first layer, and trained on either 1 or 2 datapoints), and even within that setting the results are limited (proving that EOS cannot happen for 1 datapoint, and then arguing informally why EOS can happen for two datapoints).

**Questions:**

Do you think it is possible to generalize this style of analysis to an arbitrary number of input datpoints?

---

### Official Review · Reviewer_PugV · 2025-10-31

**Soundness:** 3
**Presentation:** 2
**Contribution:** 2
**Rating:** 4
**Confidence:** 3

**Summary:**

The paper explores the Edge of Stability (EoS) phenomenon in a minimal yet analytically tractable setting—a two-layer rank-1 linear model trained with gradient descent on linearly separable data under logistic loss. By reducing the training dynamics to a low-dimensional system, the authors rigorously prove that EoS cannot arise in the single-datapoint case, while empirical analyses reveal that EoS behavior can emerge when two datapoints are present. They identify the coupling mechanism responsible for oscillations in both the top Hessian eigenvalue ($\lambda_{H,1}$) and the non-monotonic loss trajectory, and further derive new analytical approximations for $\lambda_{H,1}$. In an asymptotic regime where the margin tends to zero, the study presents evidence of stable, long-term oscillations in loss and sharpness. The paper concludes by framing an open challenge—to establish a rigorous theoretical proof of EoS in the two-datapoint case and to generalize the analysis to more complex, multi-datapoint settings.

**Strengths:**

- The paper presents a clean and tractable model that isolates the EoS phenomenon in the simplest possible classification setting. This theoretical minimalism allows the dynamics to be analyzed rigorously and transparently.
- The proof that EoS cannot occur in the single-datapoint logistic-loss regime is an important negative result. It helps delineate the boundary conditions under which oscillatory training dynamics cannot arise, complementing prior works that mainly focus on positive evidence of EoS.

**Weaknesses:**

- The negative result—“no EoS” in the single-datapoint, two-layer rank-1 linear, logistic-loss setting with additional structural assumptions (including symmetric initialization)—rests on an extremely simplified regime. This departs substantially from realistic training (multiple datapoints, nonlinearities, SGD noise, asymmetric initialization, heterogeneous data). How does this help in understanding actual network training and the EOS phenomenon?
- The paper states that its analysis—together with the sharpness approximation in Appendix C.2.2—serves as a starting point for understanding EoS with logistic losses in more complicated models. As it stands, the evidence is drawn from highly simplified dynamics and empirical observations; there remains a substantial gap to robust mechanisms in multi-sample / higher-rank / noisy-optimization / weakly nonlinear regimes.
- The formal definition of EoS is based solely on multiple local maxima in loss within a time window. Without an explicit condition on the sharpness threshold ($\lambda_{H,1} \approx 2/\eta$), the definition risks conflating general loss oscillations with true stability-edge phenomena.

**Questions:**

1. The experiments show EoS only for specific $(\eta, x_1^\top x_2)$ pairs. Could the authors characterize this dependence systematically, perhaps via a bifurcation diagram or parameter sweep?
2. How does the author explain the loss spike behavior we observe in practice, that is, sometimes it does not manifest as oscillations but as sharp loss spikes? How does the author's analysis reflect the subtle differences between them?
3. See the weaknesses.

---

### Official Review · Reviewer_Ndx7 · 2025-11-04

**Soundness:** 2
**Presentation:** 3
**Contribution:** 1
**Rating:** 2
**Confidence:** 4

**Summary:**

This paper investigates the Edge of Stability (EoS) phenomenon, where the top Hessian eigenvalue $ \\lambda\_{\\max} $ oscillates around the stability threshold $ {2}/{\\eta} $ during gradient descent training. Focusing on a two-layer rank-1 linear model with logistic loss for binary classification, the authors analytically show that EoS cannot occur in the single-datapoint case but can emerge with two datapoints. They identify the source of oscillations in $ \\lambda\_{\\max} $ and the non-monotonic training loss, propose new approximations for $\\lambda\_{\\max} $, and provide empirical evidence suggesting that, asymptotically, the loss and sharpness may exhibit sustained oscillations.

**Strengths:**

This paper explores an intriguing and not yet fully understood phenomenon, the Edge of Stability (EoS). The problem setting and assumptions are clearly presented, and the authors support their analysis with illustrative examples and well-designed figures. Overall, the paper is clearly written and effectively communicates its main ideas.

**Weaknesses:**

**Definition:**
The Edge of Stability (EoS) typically refers to a training regime in which the sharpness initially increases—a phase known as *progressive sharpening*—and subsequently hovers just above $ 2/\\eta $, accompanied by oscillations in the training loss. However, this paper focuses solely on one aspect of the phenomenon: the loss oscillations. This narrow focus overlooks the defining feature of EoS, namely the sharpness stabilizing slightly above $ 2/\eta $. Moreover, oscillatory behavior alone is not unique to EoS. For instance, consider applying gradient descent to $ f(x) = \\tfrac{1}{2}x\^2 + x\^3 $, where the stability threshold at the minimum $ x = 0 $ is $ 2 / f''(0) = 2 $. The resulting dynamics resemble the logistic map, exhibiting period-doubling bifurcations. In this case, one can use step sizes significantly larger than 2 and still observe stable oscillatory patterns with local maxima, which, under the paper’s definition (Def. 3), might be misclassified as EoS, even though they arise from a different mechanism.


**Network architecture:**
The authors cite [R1] as motivation for studying rank-1 networks. However, the main focus of [R1] is to demonstrate that, under gradient flow (GF), deep linear networks trained on binary classification tasks converge in direction to rank-1 solutions. This result, however, applies specifically to the GF setting, where the effects of dynamical stability induced by non-vanishing step sizes are absent. This distinction contrasts with the primary objective of the present paper, which is to investigate dynamical stability phenomena.


**Setting:**
Apart from a rank-1 linear shallow network, the setting of the analysis is unrealistic and very limited. For example:
1. Only two data points at max
2. The two datapoints are extremely correlated (over 0.99 cosine similarity)
3. Same initialization for different parts of different layers
4. The rigorous analysis applies only to datasets with one point, where the authors exclude EoS. I.e., there is no analysis of proving EoS.
5. Small interval of learning rates


**Missing citation:**\
A Minimalist Example of Edge-of-Stability and Progressive Sharpening


**References:**\
[R1] - Gradient descent aligns the layers of deep linear networks

**Questions:**

1. In the paper, you mentioned analyzing fixed points of the iterations (Sec. 3.4). What are the stable points of the dynamics? Kindly note that in this setting, there are no minima, only infima. Specifically, the weights converge in direction.

---

### Meta-Review · Area_Chair_o3Am · 2026-01-07

**Summary:**

As no rebuttal was submitted, the paper is rejected.

**Reviewer Concerns:**

No rebuttal

**Reviewer Scores:**

No rebuttal

---

### Decision · Program_Chairs · 2026-01-26

Reject